

# Excess nitrogen as a marker of intense dinitrogen fixation in the Western Tropical South Pacific Ocean: impact on the thermocline waters of the South Pacific

Alain Fumenia[1], Thierry Moutin[1], Sophie Bonnet[2], Mar Benavides[1,4], Anne Petrenko[1], Sandra Helias Nunige[1], and Christophe Maes[3]

[1]Aix Marseille Univ, Universite de Toulon, CNRS, IRD, OSU PYTHEAS, Mediterranean Institute of Oceanography (MIO), UM 110, 13288, Marseille, Cedex 09, France
[2]Aix Marseille Université, CNRS, Université de Toulon, IRD, OSU Pythéas, Mediterranean Institute of Oceanography (MIO), UM 110, 98848, Nouméa, New Caledonia
[3]Laboratoire d'Oceanographie Physique et Spatiale, CNRS, Ifremer, IRD, UBO, Brest, France
[4]Marine Biology Section, Department of Biology, University of Copenhagen, 3000 Helsingør, Denmark

*Correspondence to*: Alain Fumenia (alain.fumenia@mio.osupytheas.fr)

**Abstract**. As part of the Oligotrophy to UlTra-oligotrophy PACific Experiment cruise, which took place in the Western Tropical South Pacific during the austral summer (March-April 2015), we present data on nitrate, phosphate and on particulate and dissolved organic matter. The stoichiometric nitrogen-to-phosphorus ratios of the inorganic and organic material and the tracer N* are described. N* allows to trace changes in the proportion of fixed nitrogen due to diazotrophy and/or denitrification. Our results showed that the Melanesian archipelago waters between 160° E and 170° W are characterized by a deficit of nitrate and phosphate in the productive layer, significant dinitrogen fixation rates and an excess of particulate organic nitrogen compared to the canonical ratio of Redfield. A positive N* anomaly was observed in the productive layer reflecting the combined effect of phosphate uptake by diazotrophic organisms and remineralization of excess particulate organic nitrogen. The South Pacific Gyre waters between 170° W and 160° W were depleted in nitrate but rich in phosphate. Surface waters exhibited very low dinitrogen fixation rates, an absence of excess particulate organic nitrogen and a N* signal close to zero. The higher iron availability coupled with an absence of nitrate in the suface water of the Melanesian archipelago could stimulate the diazotrophic activity, which in turn will introduce excess nitrogen, deplete the surface waters in phosphate and be the explanation for the positive N* anomaly in the Melanesian archipelago waters. In the thermocline waters, the N* tracer revealed its full complexity, with notably the cumulative effect of the remineralization of particulate organic nitrogen and the effects of the mixing of water masses. At the global ocean scale, calculation of N* signal from the new Global Ocean Data Analysis Project version 2 database showed a strong spatial decoupling between the thermocline waters of the Eastern Tropical South Pacific and those of the Western Tropical South Pacific. A strongly positive N* anomaly was observed in the thermocline waters of the Western Tropical South Pacific in the Coral/Tasman Seas and in the southern part of the subtropical gyre between latitude 23° S and 32° S. A strong negative N* signal was observed in the waters of the Eestern Tropical South Pacific between latitude 5°S and 20°S-23°S. We hypothesise that the nitrogen excess observed in the thermocline waters of the Western Tropical South Pacific is transported eastward and then northward by the circulation of the South Pacific subtropical gyre and could influence positively the thermocline waters of the South Pacific being thus at the origin of the westward increase of the strongly negative N* signal transported by the South Equatorial Current.

## 1 Introduction

It is commonly accepted that net biological activity is supported by the supply of new nitrogen into the surface productive layer (Dugdale and Goering, 1967; Capone et al., 2005). At the global ocean scale, Redfield et al. (1963) reported similarity



between nitrogen-to-phosphorus (N:P) ratios in surface ocean particulate organic matter (POM) and in deep water inorganic nutrients, and since then the canonical Redfield ratio (RR) of 16:1 has become a fundamental tenet in marine biogeochemistry (Deutsch and Weber, 2012). Deviations from this ratio have been used by marine geochemists to provide insights into nutrient limitation of primary production, efficiency of the biological carbon sequestration and dinitrogen ($N_2$) fixation (Moore et al.,

2013). The canonical RR of the organic and mineral matter is strongly variable at a regional scale (Moore et al., 2013) and mostly depends on nitrogen sources and sinks. The supply of nitrogen to the productive layer comes from turbulent mixing of water masses and diffusion (Oschlies and Garcon, 1998; Capone et al., 2005), atmospheric and river inputs (Hansell et al., 2004; Jickells et al., 2017) and $N_2$ fixation, the conversion of atmospheric $N_2$ into ammonia by diazotrophic organisms (Karl et al., 1992). These sources are continuously balanced by N loss processes, mainly driven by denitrification which converts

nitrate ($NO_3^-$) into $N_2$ in Oxygen Minimum Zones (OMZ) (Codispoti and Richards, 1976). At the global ocean scale, the nitrogen budget is controlled by a balance between the main sources ($N_2$ fixation), and sinks (denitrification and anammox) (Gruber and Sarmiento, 1997; Gruber, 2016), both mediates by microbial processes.

The spatial and temporal variability of the ocean nitrogen budget remains difficult to quantify from in situ measurements. Alternatively, the use of geochemical tracers based on nutrients ratios allows to visualize and quantify the excess of nitrogen

over large time and space scales contrary to the biological estimates, which are often dedicated to process studies at the local scale and on short time scales. Broecker and Peng (1982) were the first to use a tracer to quantify the $NO_3^-$ deficit in the Indian Ocean and in the Bering Sea by multiplying the observed phosphorus concentrations by the stoichiometric ratio of 15 and by subtracting the result obtained by the observed $NO_3^-$ concentrations (Gruber and Sarmiento, 1997). Broecker (1974) and then Naqvi and Sen Gupta (1985) used the NO tracer to estimate the $NO_3^-$ deficiency observed in the Arabian Sea. Michaels et al.

(1996) and Gruber and Sarmiento (1997) proposed the N* as a tracer to determine the proportion of changes in the fixed nitrogen stock (all nitrogenous forms except atmospheric $N_2$) resulting from the combined effect of denitrification and/or of diazotrophy. Despite the apparent simplicity of the proposed formula (Eq. 3), there is nonetheless a certain complexity, especially in the deconvolution of the different processes influencing the intensity of this tracer. Factors may be biogeochemical, such as phosphate ($PO_4^{3-}$) uptake by diazotrophic organisms in the productive layer that will induce positive

N* variations (Fig. 1). The remineralization of organic matter from diazotrophic organisms, being naturally enriched in nitrogen (Redfield et al., 1963; Anderson and Sarmiento, 1995; Hansell et al., 2004), will supply nitrogen to the system and will positively influence the N* (Fig. 1). On the opposite, denitrification will cause negative variations of N* (Fig. 1), firstly because of the direct use of $NO_3^-$ as an oxidant by heterotrophic bacteria, and secondly because of the conversion of organic nitrogen to $N_2$. As a result, net positive changes in N* will trace a source of nitrogen relative to phosphorus, and negative net

changes will indicate nitrogen loss in the system through the denitrification process.

From the World Ocean Circulation Experiment (WOCE) cruises, Deutsch et al. (2001) showed that the N* signal in the South Pacific presented maximum values at the surface, followed by a decrease with depth. The highest negative N* signal (<10 $\mu$mol kg$^{-1}$) for the entire South Pacific were found in the thermocline waters of the East Tropical South Pacific (ETSP) extending horizontally westwards from the South American coast (15° S), which was later confirmed by Rafter et al. (2012)

and Yoshikawa et al. (2015). The OMZ of the ETSP is one of the three regions of the world ocean where strong denitrification occurs (Codispoti and Richards, 1976). Rafter et al. (2012) and Yoshikawa et al. (2015) suggested that the strongly negative N* signal in the ETSP is likely to be induced by the mixing of sub-Antarctic Modal Waters (SAMW) from the Southern Ocean and waters in relation with the OMZ of the ETSP. On the opposite, the highest N* values (>0) were found in surface waters of the western subtropical waters near Australia (Deutsch et al., 2001), which is consistent with the high densities of

diazotrophic organisms, including cyanobacteria *Trichodesmium* sp., found in the warm oligotrophic waters of the Western Tropical South Pacific (WTSP) (Moutin et al., 2005; Bonnet et al., 2009; Messer et al., 2015). This region has recently been identified as a globally important hot spot of $N_2$ fixation with an average aera rates of 570 $\mu$mol m$^{-2}$ d$^{-1}$ (Bonnet et al., 2017),



but the effect of intense $N_2$ fixation on geochemical properties of the waters masses in the WTSP and the whole South Pacific has never been investigated.

The aim of this study is to is to characterize in detail the N* along trophic and $N_2$ fixation gradients in the WTSP, and to assess the potential effect of intense $N_2$ fixation on the geochemical properties of thermocline waters in the South Pacific. After a brief description of the main characteristics of water masses encountered during the Oligotrophy to UlTra-oligotrophy PACific Experiment (OUTPACE) cruise, the N and P pools, nutrient ($NO_3^-$, $PO_4^{3-}$), POM and dissolved organic matter (DOM) data, are presented emphasizing their stoichiometric ratios. Then, the different processes known to influence the N* distribution are

approached. Finally, the N* distribution in the thermocline waters of the whole South Pacific will be presented and the influence of global circulation on local N* signal discussed.

## 2 Materials and methods

### 2.1 Station location and sampling

The OUTPACE cruise took place between 18 February to 3 April 2015, between longitude 159.54° E and 160° W and latitude

18° S and 22° S from Noumea (New Caledonia) to Papeete (French Polynesia) (Fig. 2a). A total of 18 stations were sampled. Station LD B is located at the geographical boundary between the Melanesian Archipelago (MA) and the waters of the South Pacific subtropical gyre (GY) (Fig. 2a). This station had high concentrations of chlorophyll-a at the surface induced by a common sub-mesoscale complex allowing the advection of surface waters from the MA towards the east (De Verneil et al., 2017). Stations SD 1 to LD B will in the remainder of this study be referred to as MA waters and will be color coded red, and

stations SD 13 to SD 15 will be referred to as the waters of the GY and will be color coded blue (Fig. 2b).

All samples were collected with a CTD Sea-Bird 911+ combined with a carousel equipped with a 24 bottles rosette Niskin® 12 liters. The entire cruise program and sampling strategy is described in Moutin et al. (2017). Measurements of the oxygen sensors were calibrated on board using a chemical assay by the Winkler (1888) method modified by Carpenter (1965) and Carritt and Carpenter (1966) and according to the protocol described in Culberson (1991) and Dickson (1996). The

determination of the oxygen concentrations was carried out partly according to the photometric method (Williams and Jenkinson, 1982) and partly according to the potentiometric method (Titrino 716 DMS Metrohm®). The saturation oxygen concentration ($[O_2]$ sat) was calculated for all stations sampled through the GSW Oceanographic Toolbox (McDougall and Barker, 2011). The Apparent Use of Oxygen (AOU) was calculated according to the following formula:

$$AOU = [O_2] \, sat - [O_2] \, mesured \; = \; -\Delta O_2 \, (remin) \tag{1}$$

$\Delta O_2$ (remin) is the change in oxygen due to the remineralization process, assuming that oxygen is at saturation when waters are in contact with the atmosphere.

### 2.2 Inorganic and organic pools

The samples used for the analysis of the inorganic nutrient were collected for each depth in a polypropylene scintillation bottle closed with a HDPE cap fixed with 50 μL of $HgCl_2$ (20 mg $L^{-1}$), and stored at 4 °C before analysis in the laboratory. $PO_4^{3-}$ and

$NO_3^-$ + nitrites ($NO_2^-$) concentrations were determined automatically using the continuous flow method (Aminot and Kérouel, 2007) using a SEAL Analitical AA3HR auto analyzer. $NO_2^-$ is obtained according to a method based on the Griess reaction, adapted for seawater by Benschneider and Robinson (1952). The $NO_3^-$ is obtained according to a technique developed by Wood et al. (1976). $PO_4^{3-}$ is obtained according to the method adapted for seawater by Murphy and Riley (1962). The quantification limit (QL) for all nutrient is 0.05 μmol $L^{-1}$. The data set of the SD 8 station is not taken into account in the results

of this study due to an analytical problem.



The samples used for the analysis of nitrogen and particulate organic phosphate (respectively PON and POP) were collected for each depth in polycarbonate bottles and 1.2 L were filtered through a precombusted (24h, 450 °C) glass fiber filter (Whatman GF/F, 25 mm). PON and POP concentrations were quantified using the wet oxidation method based on persulfate digestion at 120 °C (Pujo-Pay and Raimbault, 1994). $NO_3^-$ and $PO_4^{3-}$ formed by oxidation were analyzed using the auto analyzer

SEAL Analytical AA3HR as described above. The QL of the POP was 0.001 µmol $L^{-1}$ and of PON 0.02 µmol $L^{-1}$.

The dissolve organic nitrogen (DON) and phosphorus (DOP) samples were collected in 100 mL combusted glass bottles from each depth and immediately filtered through 2 precombusted (24 h, 450 °C) glass fiber filters (Whatman GF/F, 25 mm). Filtered samples were collected in teflon vials adjusted to 20 ml for wet oxidation. DON and DOP concentrations were obtained by the wet oxidation procedure according to Pujo-Pay and Raimbault (1994). Just after collection, 2.5 mL of the oxidative

reagent were added to the samples that were then heated to 120 °C for 30 min. After cooling, the concentration of Total Dissolved Nitrogen (TDN) and Total Dissolved Phosphorus (TDP) was determined using continuous flow analysis (AAIII HR). DON is obtained by the difference between TDN and Inorganic Nitrogen species ($NO_3^- + NO_2^- + NH_4^+$). DOP was obtained by the difference between TDP and Dissolved Inorganic Phosphorus ($PO_4^{3-}$). The QL of POD was fixed at 0.05 µmol $L^{-1}$ and of DON 0.50 µmol $L^{-1}$. All DON and DOP data from the SD 3 and SD 15 stations are not taken into account in the results of

this study due to an analytical problem.

**2.3 $N_2$ fixation rates**

$N_2$ fixation rates were measured in triplicate at all stations (except SD13) using the $^{15}N_2$ isotopic tracer technique (adapted from Montoya et al., 1996). Breifly, seawater samples were collected in HCl-washed, sample-rinsed (3 times) light-transparent polycarbonate 2.3 L bottles from 6 depths (75 %, 50 %, 20 %, 10 %, 1 %, and 0.1 % surface irradiance levels), sealed with

caps fitted with silicon septa and amended with 2 mL of 98.9 atom% $^{15}N_2$ (Cambridge isotopes). Incubation bottles were incubated in on-deck incubators equipped with circulating seawater at the specified irradiances using blue screening. Incubations were stopped by filtration of the entire sample onto precombusted glass fiber (Whatman GF/F, 25 mm, 0.7 µm nominal pore size) filters, which were then analyzed for $^{15}N/^{14}N$ ratios and PON concentrations using an elemental analyzer coupled to a mass spectrometer (EA-IRMS, Integra CN, SerCon Ltd) as described in Bonnet et al., (This issue).

**2.4 Calculation of PON excess**

The excess of PON ($PON_{excess}$) represents the difference between the measured PON concentrations and the PON concentrations calculated based on the RR (PON: POP = 16: 1). It was calculated as follows:

$$PON_{excess} = PON_{mesured} - 16 \times POP_{mesured} \ (\mu mol \ L^{-1}) \tag{2}$$

**2.5 Calculation of N***

The N* signal was calculated from the definition of Deutsch et al (2001).

$$N^* = NO_3^- - r_{nitr}^{\frac{N}{P}} \times PO_4^{3-} + 2.90 \ (\mu mol \ L^{-1}) \tag{3}$$

$PO_4^{3-}$ is the concentration of "soluble reactive phosphate" (µmol $L^{-1}$). The concentrations of $NO_3^-$ and $PO_4^{3-}$ are linked together by the stoichiometric ratio $r^{N/P}_{nitr} = 16$ determined by Redfield et al. (1963) and Takahashi et al. (1985). Equation (3) assumes that the effect of the biological pump on the absolute value of N* is zero, that is the difference between $NO_3^-$ and remineralized

$PO_4^{3-}$ is constant (Gruber and Sarmiento, 1997). The constant 2.90 makes it possible to force to zero the overall average value of the N* calculated from GEOSECS data (Geochemical Ocean Sections Study, 1972-1978) (Deutsch et al., 2001). This constant has been included in this study in order to compare the observations with the literature data.




In surface waters, above isopycnal 23.5 (σ23.5), the $NO_3^-$ concentrations were not measurable and did not allow direct calculation of the N* anomaly. Considering only $PO_4^{3-}$ data above QL in surface waters, an N* estimate ($N^*_{surf}$) was carried out considering that below QL, $NO_3^-$ concentrations were between 0 and 0.05 µmol $L^{-1}$:

$$-16 \times PO_4^{3-} + 2.9 \leq N^*_{surf} \leq 0.05 - 16 \times PO_4^{3-} + 2.9 \tag{4}$$

In the WTSP, the upper thermocline waters are characterized by maximum subsurface salinity (Tomczak and Hao 1989; Donguy 1994; Donguy and Henin 1997) and the underlying waters of the permanent thermocline are marked by wide range of temperature and salinity (Emery and Meincke, 1986; Sprintall and Tomczak, 1992; Tomczak and Godfrey, 1994). To visualize the variations of N* in the thermocline waters of the OUTPACE section, the N* anomaly has been represented on isopycnal surfaces 24.70 and 26.30, which respectively correspond to the cores of the upper and lower thermoclines observed on the Θ-S diagram (Fig. 4a). The density, $NO_3^-$, $PO_4^{3-}$, and N* anomaly data were interpolated linearly between σ24.65 and σ24.75 and between σ26.25 and σ26.35. The mean and standard deviation of interpolated N* values was calculated for all stations on σ24.70 ±0.05 and σ26.30 ±0.05.

### 2.6 Variability of the $PO_4^{3-}$ pool

The difference in average concentration observed in $PO_4^{3-}$ between the MA and the GY waters ($\Delta PO_4^{3-}_{observed}$) was calculated from means (±sd) observed in surface waters (< σ23.5) and is equal to:

$$\Delta PO_4^{3-}_{observed} = PO_4^{3-}_{observed (MA)} - PO_4^{3-}_{observed (GY)} \tag{5}$$

From the estimate of $N^*_{surf}$ in surface waters (< σ23.5), the difference between the mean concentrations observed in N* (ΔN*) in the surface MA and surface GY waters is equal to:

$$\Delta N^* = N^*_{mean (MA)} - N^*_{mean (GY)} \tag{6}$$

Assuming that the formation of the organic matter is consistent with the RR, the variation of $PO_4^{3-}$ concentrations induced by the excess nitrogen supply in the MA ($\Delta PO_4^{3-}_{estimated}$) is equal to:

$$\Delta PO_4^{3-}_{estimated} = -\Delta N^*/16 \tag{7}$$

### 2.7 GLODAPv2 database

The Global Ocean Data Analysis Project version 2 (GLODAPv2) is an international effort to consolidate all data from ocean bottle samples collected as part of many oceanic cruises (Olsen et al., 2016). Previous databases, WOCE/JGOFS combined in GLODAPv1 in 2004 (Sabine et al., 2005; Key et al., 2004), CARINA (CARbon IN the Atlantic) in 2009/2010 (Key et al., 2010; Tanhua et al., 2009), and PACIFICA (PACIFIc ocean Interior CArbon) in 2013 (Suzuki et al., 2013) as well as 168 additional cruises were grouped together and have undergone one and only high-quality control, based on two steps (primary and secondary QC) applied to each data. The GLODAPv2 database is available at http://cdiac.ornl.gov/oceans/GLODAPv2/ and the details of the overall production strategy are explained in Olsen et al. (2016). The GLODAPv2 database includes samples of core variables such as salinity, oxygen, macronutrients, and seawater $CO_2$ chemistry from 724 oceanic cruises. In this study, we focused on two variables: $NO_3^-$ and $PO_4^{3-}$ and we calculated N* according to the formula of Deutsch et al. (2001) (Eq. 3) for all available data in the South Pacific. The details of the available cruises are shown in figure 3 and in Table S1. Similar to OUTPACE data processing, the density, $NO_3^-$, $PO_4^{3-}$, and N* anomaly data were interpolated linearly between σ24.65 and σ24.75 and between σ26.25 and σ26.35.



## 3 Results


In order to replace the OUTPACE cruise within the context of the general circulation in the South Pacific we start by a global survey (Fig. 2b). The South Equatorial Current (SEC) represents the northern branch of the southern subtropical anticyclonic gyre from the eastern boundary Peru-Chile Current (PCC) to the Coral Sea and the southwestern Pacific (Wyrtki, 1975; Solokov and Rintoul, 2000; Talley et al., 2011; Kessler and Cravatte, 2013). At depth, the SEC transports thermocline waters

which are salty (larger than 35), and thus makes it possible to spatially couple the ETSP and the WTSP. Arriving to the MA, the SEC is divided into two branches: The North Vanuatu Jet (NVJ) which brings salty, warm and relatively low oxygenated waters (Wyrtki, 1962b; Solokov and Rintoul, 2000; Webb, 2000) and the North Caledonian Jet (NCJ) which flows near 18° S between the surface and 1500 m and transports more oxygenated waters (Gourdeau et al., 2008; Kessler and Cravatte, 2013; Gasparin et al., 2014). At its arrival to the Australian coasts, the SEC bifurcates and becomes the East Australian Current

(EAC) to the south and the North Queensland Current (NQC) toward the north and the Solomon Sea. Near 30° S, the EAC breaks down into several branches: one continuing south of 40° S, another one returning to the north through whirlwinds and meanders (Church 1987; Solokov and Rintoul 2000; Marchesiello and Middleton, 2000). The last part flows eastward along the Tasman Front (TF) (40°-45° S) through the north of New Zealand to become East Auckland Current (EAUC). It then joins back into and the circulation of the anticyclonic gyre through the South Pacific Current (SPC) which mostly circulates nearby

the latitude of 30° S (Stramma et al., 1995; Talley et al., 2011). The CTD profiles collected during the OUTPACE cruise take place in the southwestern Pacific region when the SEC encounters the Fiji and New Caledonia (Fig, 2a; 2b).

### 3.1 Water masses and general biogeochemical trends

Figure 4a shows the superimposition of $\Theta$-S diagrams (0-2000 dbar) for all 18 stations sampled during the cruise. The temperature range observed on the OUTPACE section varied from 29.92 °C at the surface to 1.99 °C at 2000 dbar depth (Fig.

4a). Salinity varied from 34.32 to 36.11 (Fig. 4a). The surface waters (SW) were in the range between $\sigma 21.7$ and $\sigma 23.5$ (4.12-71.30 dbar, Table 1) and had a temperature range of 25.60 °C to 29.93 °C and were marked by small longitudinal variations (Fig. 4a). Surface salinity values ranged from 34.89 for the most westerly stations (Fig. 4a) to 35.67 for the most easterly stations (Fig. 4a). The concentrations of AOU were close to zero or slightly negative in the SW for all the stations (Fig. 4b). A minimum of AOU concentrations of about -14 µmol kg$^{-1}$ was visible between $\sigma 23$ and $\sigma 23.5$ (Fig. 4b). In ths SW of the

MA, N$_2$ fixation rates ranged from 1.41 to 42.25 nmol N L$^{-1}$ d$^{-1}$ with an average rate of $12.38 \pm 9.80$ nmol L$^{-1}$ N d$^{-1}$ (Fig. 4d, Table 1). Rates decreased drastically towards the GY with values ranging 0.01 and 2.29 nmol N L$^{-1}$ d$^{-1}$ and an average rate of $0.75 \pm 0.90$ nmol N L$^{-1}$ d$^{-1}$ (Fig. 4d, Table 1). Between $\sigma 23.5$ and $\sigma 25.4$, the Upper Thermocline Waters (UTW) (35.42-251.52 dbar, Table 2) were characterized by a temperature and salinity range between 19.26 °C and 26.55 °C and 35.58 and 35.90, respectively (Fig. 4a). A salinity maximum (S$_{max}$) was observed between $\sigma 24$ and $\sigma 25.4$, centered on $\sigma 24.7$ (Fig. 4a). The S$_{max}$

had maximum values in the UTW of the GY (S≈36.11) and tended to decrease to reach a maximum of 35.6-35.7 in the UTW of the MA (Fig. 4a). An increase in Chl-a concentrations was observed in the upper part of the UTW (Fig. 4c). N$_2$ fixation rates decreased sharply between $\sigma 23.5$ and $\sigma 24.7$ and reached homogeneous values between 0.01 and 1.73 nmol N L$^{-1}$ d$^{-1}$ in the MA waters and between 0.01 and 1.18 nmol N L$^{-1}$ d$^{-1}$ in the GY waters (Fig. 4d). The AOU concentrations increased for all the stations with nevertheless a significant variability of the observed values which varied from -10.10 to 55.70 µmol kg$^{-1}$

in the UTW of the MA and between -7.40 and 40.90 µmol kg$^{-1}$ in the UTW of the GY (Fig.4b). Between $\sigma 25.4$ and $\sigma 26.7$ (249.14-501.21 dbar, Table 3), a strong decreasing gradient of temperature and associated salinity marked the presence of Lower Thermocline Waters (LTW). The values of temperature and salinity varied little from east to west on $\sigma 25.40$ and were $19.34 \pm 0.11$ °C and $35.61 \pm 0.03$ (Fig. 4a), respectively. In the LTW, the temperature and salinity ranged from 19.26 °C to 9.34 °C and from 35.72 to 34.54, respectively (Fig. 4a). The core of the LTW ($\sigma 26.3$) was colder (T=12.93 ±0.09 °C), less

salty (S=34.87 ±0.02) in the GY than in the MA (T=13.68 ± 0.26 °C, S=35.07 ± 0.06) (Fig. 4a). The AOU concentrations





increased almost linearly in the LTW and ranged from 38.20 to 93.20 µmol kg$^{-1}$ in the MA and between 54.70 and 130.50 µmol kg$^{-1}$ in the GY (Fig. 4b). Below σ25.4, the Chl-a concentrations were close to zero for all the stations (Fig. 4c) and the N$_2$ fixation rates were undetectable (Fig. 4d). Subantarctic Modal Water (SAMW) and Intermediate Antarctic waters (AAIW) were considered in this study as a single water mass that flowed between σ26.7 and σ27.3 (496.91-1008 dbar, Table 4) and

was characterized by a minimum of salinity (S=34.37 ± 0.02) centered on σ27.1 (Fig. 4a). Deep waters (σ27.30-σ27.65) (Upper Circumpolar Deep Water, UCDW) were characterized by an increase in salinity and a decrease in temperature. On σ27.65 the salinity and temperature, respectively, reached 34.62 ± 0.01 and 2.25 ± 0.04 °C (Fig. 4a).

### 3.2 Inorganic nutrient pools

The properties observed in the MA and GY waters differed greatly. The MA waters showed in the productive layer (≈0-150

dbar) NO$_3^-$ concentrations below the QL, PO$_4^{3-}$ concentrations below or close QL, and high concentrations of PON and POP. (Fig. 5). PO$_4^{3-}$ concentrations increased in the productive layer of the GY waters, along with a decrease in PON and POP concentrations (Fig. 5).

The concentrations of NO$_3^-$ were below the QL (0.05 µmol L$^{-1}$) in the SW for all stations (Fig. 6a). The concentrations of PO$_4^{3-}$ were low in the SW of the MA and ranged between 0.05 and 0.17 µmol L$^{-1}$ (mean=0.08 ± 0.03) (Fig. 6b, Table 1). An increase

in PO$_4^{3-}$ concentrations was observed in the SW of the GY with values ranging between 0.11 and 0.17 µmol L$^{-1}$ (mean=0.15 ± 0.02) (Fig. 6b, Table 1). The NO$_3^-$ concentrations became quantifiable from σ23.5 and increased gradually. In the UTW, the mean NO$_3^-$ concentration was 2.35 ± 1.59 µmol L$^{-1}$ in the MA and 2.19 ± 1.90 µmol L$^{-1}$ in the GY (Fig. 6a, Table 2). The PO$_4^{3-}$ concentrations were 0.24 ± 0.11 µmol L$^{-1}$ in the MA and 0.28 ± 0.15 µmol L$^{-1}$ in the GY (Fig. 6b, Table 2). In the LTW, the concentrations increased with depth. The NO$_3^-$ concentrations ranged between 4.13 and 18.35 µmol L$^{-1}$ (mean=10.07 ± 4.53)

in the MA and between 6.42 and 18.69 µmol L$^{-1}$ (mean=11.33 ± 4.46) in the GY (Fig. 6a, Table 3). The PO$_4^{3-}$ concentrations ranged between 0.35 and 1.31 µmol L$^{-1}$ (mean=0.77 ± 0.31) in the MA and between 0.57 and 1.44 µmol L$^{-1}$ (mean=0.92 ± 0.29) in the GY (Fig. 6b, Table 3). In the SAMW/AAIW, the NO$_3^-$ concentrations increased from 19.62 to 33.81 µmol L$^{-1}$ (mean=26.91 ± 4.32) in the MA and from 23.60 to 34.26 µmol L$^{-1}$ (mean=29.16 ± 4.17) in the GY (Fig. 6a, Table 4). The PO$_4^{3-}$ concentrations increased from 1.37 to 2.32 µmol L$^{-1}$ (mean=1.88 ± 0.29) in the MA and from 1.67 to 2.35 µmol L$^{-1}$

(mean=2.03 ± 0.25) in the GY (Fig. 6b, Table 4). The bottom waters (>σ27.5, 1000-2000 dbar) had relatively homogeneous maximum concentrations with a range of values in NO$_3^-$ and PO$_4^{3-}$ between 35.10 and 36.50 µmol L$^{-1}$ and between 2.40 and 2.55 µmol L$^{-1}$, respectively (Fig. 6a, Fig.6b).

The NO$_3^-$: PO$_4^{3-}$ ratio increased sharply in the UTW. In the UTW of the MA, the ratio ranged between 0.36 and 13.04 (mean=8.09 ± 3.65) and between 0.36 and 9.77 (mean=5.49 ± 3.62) in the UTW of the GY (Fig. 6c, Table 2). In the LTW, the

NO$_3^-$: PO$_4^{3-}$ ratio increased with a lower slope and rose from 11.33 to 14.33 (mean=12.92 ± 0.82) in the MA and from 10.64 to 14.00 (mean=12.06 ± 1.16) in the GY (Fig. 6c, Table 3). The NO$_3^-$: PO$_4^{3-}$ ratio reached stable and uniform values in bottom waters of the MA and the GY between 13.80 and 15.20 (Fig. 6c). A linear regression (R$^2$=0.998) was observed between NO$_3^-$ and PO$_4^{3-}$ concentrations during the OUTPACE cruise. The slope was 15.01, which was close to the RR of 16:1 (Fig. 6d). In the upper waters, the depletion of NO$_3^-$ is higher than the depletion of PO$_4^{3-}$. A deficit in NO$_3^-$ compared to the value expected

by RR (dashed black line) was observed for all the data (Fig. 6d).

### 3.3 Organic pools

Maximum PON and POP concentrations were observed in the SW of the MA. The PON concentrations ranged between 0.271 and 1.175 µmol L$^{-1}$ (Fig. 7a) and the concentrations of POP ranged between 0.016 and 0.057 µmol L$^{-1}$ (Fig. 7b). The minimum concentrations were observed in the SW of the GY, with respective PON and POP values between 0.166 and 0.315 µmol L$^{-1}$

(Fig. 7a) and between 0.011 and 0.019 µmol L$^{-1}$ (Fig. 7b). On average, the PON and POP concentrations were approximately





twice as high in the SW of the MA ($PON_{mean}$ =0.549 ± 0.183 µmol L$^{-1}$, $POP_{mean}$ =0.029 ± 0.007 µmol L$^{-1}$) as in the SW of the GY ($PON_{mean}$ =0.209 ± 0.036 µmol L$^{-1}$, $POP_{mean}$ =0.014 ± 0.003 µmol L$^{-1}$) (Table 1). In the UTW, significant attenuation of POM concentrations was observed. The mean concentrations, however, remain higher in the UTW of MA ($PON_{mean}$ =0.210 ± 0.114 µmol L$^{-1}$, $POP_{mean}$ =0.013 ± 0.007 µmol L$^{-1}$) than in the UTW of the GY ($PON_{mean}$ =0.151 ± 0.078 µmol L$^{-1}$, $POP_{mean}$

=0.009 ± 0.005 µmol L$^{-1}$) (Fig. 7a, Fig. 7b, Table 2). POM concentrations decrease to reach minimum and relatively stable values in the LTW. PON concentrations in the LTW of MA ranged from 0.051 to 0.134 µmol L$^{-1}$ (mean=0.084 ±0.019) and POP concentrations ranged from 0.003 to 0.008 µmol L$^{-1}$ (mean=0.005 ± 0.001) (Fig. 7a, Fig. 7b, Table 3). The PON concentrations in the LTW of the GY were between 0.050 and 0.097 µmol L$^{-1}$ (mean=0.067 ± 0.017) and the POP concentrations were between 0.003 and 0.006 µmol L$^{-1}$ (mean=0.005 ± 0.001). (Fig. 7a, Fig. 7b, Table 3).

For all the stations and for all the depths sampled, the PON:POP ratio was between 10.08 and 25.78 (Fig. 7c). The average value of the PON:POP ratio was 16.89 ± 2.37, which was close to the RR. Nevertheless, a strong variability was observed between the different oceanic regions (MA *vs* GY) and on the water masses considered. In the SW, the PON:POP ratio varied from 14.56 to 25.32 (mean=18.50 ± 2.44) in the MA and between 12.59 and 20.15 (mean=15.58 ± 1.57) in the GY (Fig. 7c, Table 1). In the UTW, the PON:POP ratio was between 13.07 and 20.04 (mean=16.39 ± 1.48) in the MA and between 14.50

and 18.65 (mean=16.27 ± 1.05) in the GY (Fig. 7c, Table 2). In the LTW, the PON:POP ratio showed a strong inter-station disparity. The PON:POP ratio was between 12.85 and 25.78 (mean=16.79 ± 3.19) in the MA and between 13.80 and 15.80 (mean=14.91 ± 0.92) in the GY (Fig. 7c, Table 3). For PON concentrations not exceeding 0.400 µmol L$^{-1}$ and 0.025 µmol L$^{-1}$ in POP, there was a strong correlation ($R^2$ =0.962) and a slope of 15.145 close to the RR for the GY waters (blue markers in Fig. 7d). The highest MOP concentrations showed an excess of PON over POP with a slope of 19.963 and a significant

deviation from the expected RR for the MA stations (red markers in Fig. 7d). PON in excess of the expected RR had the highest concentration between surface and σ22.5 (≈0-25 dbar) for the MA stations with a maximum PON excess of 0.370 µmol L$^{-1}$ (Fig. 8). PON excess decreased between σ22.5 and σ23.5. In the UTW and LTW, PON excess disappeared and the PON concentrations were close to those calculated on the basis of the RR (Fig. 8).

In the SW, the DON and DOP concentrations were maximum. DON concentrations ranged from 4.77 to 6.81 µmol L$^{-1}$

(mean=5.49 ± 0.44) in the SW of the MA and between 4.74 and 6.39 µmol L$^{-1}$ (mean=5.13 ± 0.46) in the SW of the GY (Fig. 9a, Table 1). In the SW of MA, DOP concentrations ranged between 0.10 and 0.30 µmol L$^{-1}$ (mean=0.16 ± 0.03) and between 0.16 and 0.23 µmol L$^{-1}$ (mean=0.19 ± 0.03) in the SW of the GY (Fig. 9b, Table 1). DOM concentrations decreased in the UTW and then remained relatively stable in the LTW. The DON concentrations ranged between 2.77 and 4.25 µmol L$^{-1}$ (mean=3.50 ± 0.38) in the LTW of MA and between 2.78 and 4.01 µmol L$^{-1}$ (mean=3.45 ± 0.41) in the LTW of the GY (Fig.

9a, Table 3). DOP concentrations ranged from 0.02 to 0.12 µmol L$^{-1}$ (mean=0.06 ± 0.03) in the LTW of the MA and between 0.02 and 0.08 µmol L$^{-1}$ (mean=0.04 ± 0.02) in the LTW of GY (Fig. 9b, Table 3). For all the stations and the entire water column, the DON:DOP ratio was far from the RR. Minimum values were observed in the SW with nevertheless higher values in the MA. The ratio DON:DOP ranged between 18.70 and 57.70 (mean=34.15 ± 5.96) in the SW of the MA and between 22.48 and 32 (mean=27.83 ± 2.72) in the SW of the GY (Fig. 9c, Table 1). An increase of the DON: DOP ratio and its

dispersion were observed in the UTW and the LTW. In the LTW, the DON:DOP ratio varied from 31.75 to 182 (Fig. 9c, Table 3).

### 3.4 Distribution of N*, $N_2$ fixation rates and variations of $PO_4^{3-}$ pool

The range of N* concentrations varied between -2.52 and 2.15 µmol L$^{-1}$ along the transect (Fig. 10). A positive N* anomaly between 0.23 and 2.15 µmol L$^{-1}$ was observed in the SW, (0-50 dbar) coupled with a strong disparity of the signal marked by

values between 0.23 and 1.19 µmol L$^{-1}$ in the SW of the GY and between 0.25 and 2.15 µmol L$^{-1}$ in the SW of the MA (Fig. 10). The UTW and LTW (100-500 dbar) showed a significant difference between the MA and GY waters. In the MA, the N* signal in UTW was positive (mean=1.11 ± 0.53) with values ranging between -0.12 and 2.13 µmol L$^{-1}$ and negative in the GY



(mean = -0.14 ± 0.49) with values between -1.13 and 0.75 µmol L$^{-1}$ (Fig. 10b, Table 2). In the LTW, a disparity in N* concentrations was observed, marked by a difference between the MA and GY waters. The latter showed a weaker and mostly

negative signal from 150 dbar depth (Fig. 10a). In the LTW, the N* concentrations ranged between -0.49 and 1.52 µmol L$^{-1}$ (mean=0.75 ± 0.54) in the waters of the MA and between -1.55 and 0.72 µmol L$^{-1}$ (mean= -0.51 ± 0.78) in GY (Fig. 10b, Table 3). The N* signal had a minimum value of -0.77 µmol L$^{-1}$ at 500 dbar depth in the MA waters and -1.55 µmol L$^{-1}$ at 400 dbar depth in the GY waters (Fig.10a). Above 500 dbar depth, the dispersion of the signal was attenuated, and the concentrations gradually decreased to reach at 2000 dbar depth values which was mostly negative for all the stations, ranging between -2.24

and -1.19 µmol L$^{-1}$ in the MA waters and between -1.35 and -0.51 µmol L$^{-1}$ in the GY waters (Fig. 10a).

Figure 11 shows the variability of N* concentrations over the isopycnals σ24.7 and σ26.3. Independently of the isopycnal, the N* anomaly was minimum in the GY and maximum in the MA waters. In the MA waters, between SD 1 and SD 12, the N* anomaly was within σ24.7 between 1.05 ± 0.01 and 1.77 ± 0.02 µmol kg$^{-1}$ and between -0.02 ± 0.02 and 1.21 ± 0.02 µmol kg$^{-1}$ on σ26.3. A drastic decrease of the N* anomaly was observed on σ24.7 and σ26.3 between SD 12 and SD 15. A minimum

was observed at SD 15 station with values of -0.68 ± 0.07 µmol kg$^{-1}$ on σ24.7 and -1.46 ± 0.02 µmol kg$^{-1}$ on σ26.3 (Fig. 11). N$_2$ fixation rates integrated upon the water column in the MA was between 100.39 and 840.56 µmol N m$^{-2}$ d$^{-1}$ (red circle in Fig. 11) and the average integrated fixation rate was 442.81 ± 203.53 µmol N m$^{-2}$ d$^{-1}$. A sharp decrease in integrated N$_2$ fixation rates was observed in the SW of the GY, with values between 4.79 and 72.53 µmol N m$^{-2}$ d$^{-1}$ (blue circle in Fig. 11) with an average value of 27.88 ± 38.69 µmol N m$^{-2}$ d$^{-1}$.

The average difference in PO$_4^{3-}$ concentrations observed between the SW of the MA and the GY (ΔPO$_4^{3-}$$_{observed}$) was -0.07 ± 0.01 µmol L$^{-1}$. From the estimation of N* in the SW, the difference between the average concentrations observed in N* (ΔN*) in the SW of the MA and the GY was 1.05 ± 0.37 µmol L$^{-1}$. The variation in PO$_4^{3-}$ concentrations induced by the excess supply of nitrogen in the MA (ΔPO$_4^{3-}$$_{estimated}$) was equal to -0.07 ± 0.03 µmol L$^{-1}$.

### 3.5 Distribution of N* signal at the South Pacific scale

On σ24.7, the N* signal was minimum between latitude 5° S and 20° S and longitude 80° W and 100° W (P21, P18, P19 sections in Fig. 12a) with an average value of about -4.63 ± 0.80 µmol kg$^{-1}$ on sections P21 and a minimum value ranged between -6.40 and -6.20 µmol kg$^{-1}$ on sections P18 and P19 (Fig. 12a), respectively. Between latitude 5° S and 15° S, the N* signal observed in the OMZ region increased during its westward transport by the northern branch of the SEC. Between latitude 5° S and 15° S, the N* signal reached -3.50 ± 0.90 µmol kg$^{-1}$ at longitude 135° W (P16S section in Fig. 12a), -1.83 ± 0.60

µmol kg$^{-1}$ at longitude 150-155° W (P16C section in Fig. 12a), -1.34 ± 0.55 µmol kg$^{-1}$ at longitude 170° W (P15 section in Fig. 12a). The N* signal carried by the NVJ (Fig. 2b) increased in the WTSP and reached -0.27 ± 0.97 µmol kg$^{-1}$ at longitude 155° E (P11S section in Fig. 12a). South of 15° S, the strongly negative N* signal increased more rapidly in the UTW. During the westward transport of the UTW by the southern branch of the SEC, N* signal increased from -4.37 to -0.44 µmol kg$^{-1}$ (mean = -2.00 ± 1.00) between longitude 100° W and 150° W (P21 section in Fig. 12a). In the WTSP between longitude 175° E and

150 °E, N* signal increased sharply during the westward transport of waters by the NCJ (Fig. 2b) reaching values of 0.39 ± 0.58 µmol kg$^{-1}$ (P21 section in Fig. 12a). N* signal was greatest in the UTW of the Coral and Tasman seas. Between latitude 23° S and 30° S, N* reached 1.69 ± 0.16 µmol kg$^{-1}$ (P14C section), and 1.86 ± 0.19 µmol kg$^{-1}$ (P15 section in Fig. 12a).

On σ26.3, N* signal reached a mean minimum value of -18.08 ± 1.69 µmol kg$^{-1}$ in the OMZ region between longitude 80° W and 90° W and latitude 18° S (P21, GT13 sections in Fig. 12b, Fig. S2). N* signal was minimum between latitude 5° S and

15° S and longitude 90° W with an average value of -10.31 ± 6.58 µmol kg$^{-1}$ and between latitude 15° S and 23° S with an average value of -11.62 ± 4.13 µmol kg$^{-1}$ (P19 sections in Fig. 12b, Fig. S2). Maximum values of N* was observed in the MA waters of the OUTPACE section in the southern Coral Sea and Tasman Sea. The N* signal reached 0.52 ± 0.12 µmol kg$^{-1}$ between latitude 23° S and 32° S and at longitude 175° E (P14C section in Fig. 12b, Fig. S2). At latitude 30° S and between longitude 150° E and 180 °E in the Tasman Sea and north of New Zealand, N* signal was 0.66 ± 0.46 µmol kg$^{-1}$ (P06 section



in Fig. 12b, Fig. S2). N* signal remained relatively high in the central part of the South Pacific between latitude 23° S and 32°
S and longitude 175° E and 135° W, with a mean value of $0.77 \pm 0.23$ µmol kg$^{-1}$ at longitude 170° W (P15 section) and $0.29 \pm 0.14$ µmol kg$^{-1}$ at longitude 135° W (P16 section in Fig. 12b, Fig.S2). At latitude 32° S, N* signal remained relatively stable between longitude 180° and 120° W with a mean value of $0.34 \pm 0.27$ µmol kg$^{-1}$ between longitude 180° and 150° W and $0.21 \pm 0.16$ µmol kg$^{-1}$ between longitude 150° W and 120° W (P06 section in Fig. 12b, Fig. S2).  A drastic decrease of N* signal

was observed at longitude 110° W and at longitude 85°-90° W (P18, P19 sections in Fig. 12b, Fig. S2). At latitude 32° S the
N* signal dropped sharply to $-3.51 \pm 2.72$ µmol kg$^{-1}$ at longitude 90° W and $-17.33 \pm 2.63$ µmol kg$^{-1}$ between longitude 84° W
and 82° W (P06 section in Fig. 12b, Fig. S2). South of latitude 32° S, on σ26.3 the N* signal showed a positive anomaly with
a mean value of $0.29 \pm 0.55$ (P15 section in Fig. 12b, Fig. S2) and became predominantly negative between longitude 150° W
and 90° W with an average value of $-0.62 \pm 0.33$ µmol kg$^{-1}$, $-0.69 \pm 0.76$ µmol kg$^{-1}$ and $-0.71 \pm 0.35$ µmol kg$^{-1}$ respectively at

longitude 135° W, 110° W and 90° W (P16, P18, P19 sections in Fig. 12b, Fig. S2).

## 4. Discussion

### 4.1 Sharp increase of N* in surface waters of the MA

A positive N* anomaly for all the stations, with minimum values ($\approx 0.59 \pm 0.24$ µmol L$^{-1}$) in the GY waters and maximum
values ($\approx 1.64 \pm 0.50$ µmol L$^{-1}$) in the SW of the MA was observed (Fig. 10b). A difference of $1.05 \pm 0.37$ µmol L$^{-1}$ was

observed between MA and GY surface waters. Several factors may explain this difference.

### 4.1.1 Nitrogen supply and PO$_4$$^{3-}$ uptake by N$_2$ fixing organisms in the SW of the MA

A low concentration of NO$_3$$^{-}$ characteristic of oligotrophic environment was observed in the upper water of the gyre. Deutsch
et al. (2007) suggested that waters deficient in NO$_3$$^{-}$ were a favorable environment for N$_2$ fixation by probably limiting
interspecific competition and favoring organisms adapted physiologically to nitrogen-deficient environments. Although the

absence of NO$_3$$^{-}$ in the SW of the gyre is a necessary condition for N$_2$ fixation, this did not seem sufficient to stimulate N$_2$
fixation. The average PO$_4$$^{3-}$ concentrations are higher in the SW of the GY (PO$_4$$^{3-}$$_{obs(GY)}$ = $0.15 \pm 0.02$ µmol L$^{-1}$) than in the SW
of the MA (PO$_4$$^{3-}$$_{obs(AM)}$ = $0.08 \pm 0.03$ µmol L$^{-1}$), where concentrations in PO$_4$$^{3-}$ are almost depleted. On the other hand, the
variatibility of PO$_4$$^{3-}$ between the suface MA's waters and of that of the GY ($\Delta$PO$_4$$^{3-}$$_{estimated}$), associated with the difference of
NO$_3$$^{-}$ in excess ($\Delta$N*), were $-0.07 \pm 0.03$ µmol L$^{-1}$ and are of the same order of magnitude as the variations of PO$_4$$^{3-}$ observed

between both regions ($\Delta$PO$_4$$^{3-}$$_{observed}$ = $-0.07 \pm 0.01$ µmol L$^{-1}$).  If in the SW, the N* variations are solely due to the diazotrophic
activity, these results suggest that the decrease in PO$_4$$^{3-}$ and the associated increase in N* in the MA waters are controlled by
diazotrophic organisms. The high concentrations of PO$_4$$^{3-}$ in the SW of the GY suggest a low use by diazotrophs, as N$_2$ fixation
is hypothezised to be limited by dissolved iron (DFe) availability in GY waters (Moutin et al., 2008; Bonnet et al., this issue)
maintaining a low N* anomaly in this region. The availability of PO$_4$$^{3-}$ was suggested to control N$_2$ fixation in the WTSP

(Moutin et al., 2005; Moutin et al., 2008) which may explain that N* signal intensity may be limited by the availability of
PO$_4$$^{3-}$ in this oceanic area. The N$_2$ fixation gradient observed during the OUTPACE cruise (Bonnet et al., this issue) is likely
related to higher DFe availability in the SW of the MA (Guieu et al., under review) and much lower concentrations in the
eastern SW of the GY, which are among the lowest reported in the global ocean (Blain et al., 2008). Our results support those
of Deutsch et al. (2001), who initially suggested that the high N* values observed in the SW in the WTSP near Australia are

due to significant N$_2$ fixation in this region (Messer et al., 2016; Bonnet et al., 2015), before proposing higher N$_2$ fixation rates
in the GY (Deutsch et al., 2007). The higher DFe availability coupled with an absence of NO$_3$$^{-}$ in the SW of the MA could
stimulate the diazotrophic activity, which in turn will introduce excess nitrogen, deplete the SW in PO$_4$$^{3-}$ and be the explanation
for the positive N* anomaly in the MA waters.



**4.1.2 Excess nitrogen supply via remineralization of diazotrophic organic material**

Direct POM measurements showed significant accumulation of particles in the SW and attenuation at the top of the UTW (Fig. 7a, 7b). Despite extremely low nutrient concentrations, a significant amount of POM was measured in particular for the MA stations with PON and POP concentrations reaching 1.175 and 0.057 μmol L$^{-1}$ in the SW, respectively. Typical PON concentrations generally measured in oligotrophic regions are lower than 0.6 μmol L$^{-1}$ (Martini et al., 2013). The observed values, particularly at station LD B, are only slightly less than those measured in the Low Nutrient High Chlorophyll (LNHC)

zones of northern latitudes (Sarmiento and Gruber, 2006), which generally have a lower limit of 1.2 μmol L$^{-1}$ in PON (Martini et al., 2013). The PON:POP ratios averaged over the entire water column are very close to the RR, except for the most productive zones located in the MA, which have an excess of PON compared to POP, in particular at the top of SW (Fig. 8). This excess of PON can be as high as 0.370 μmol L$^{-1}$ compared to RR, despite undetectable NO$_3^-$ concentrations, confirming an external nitrogen supply to the system that can support organic production. Organic matter contained in diazotrophic

organisms is naturally enriched in nitrogen relative to phosphorus with a PON:POP ratio of up to 125 (Karl et al., 1992). Nevertheless, the stoichiometric PON:POP ratios observed during the OUTPACE cruise are much lower, with a maximum of 25.3 observed at station LD B. This difference could be explained because PON:POP ratio of 125 observed by Karl et al. (1992) is an exceptionally high value related to an intense bloom that probably does not reflect the general relationships associated with diazotrophs growth. Letelier and Karl (1998) observed mean ratios associated with *Trichodesmium* sp.

diazotrophs between 40 and 50, which nevertheless remains much higher than the observations made during the OUTPACE cruise. Most studies reporting extremely high ratios were carried out either in the North Pacific Gyre (Karl et al., 1992), or in the Atlantic (Hansell et al., 2004), the WTSP remaining to date an under-sampled region. Oligotrophic gyres in the northern hemisphere have lower concentrations of PO$_4^{3-}$ than those measured in the southern hemisphere (Moore et al., 2013). Since the cellular stoichiometry of organisms is a function of nutrient availability (Karl et al., 2002), PO$_4^{3-}$ deficiency largely higher

in the MA than in the GY (Moutin et al., this issue) may explain the higher PON:POP ratios in the northern hemisphere than in the WTSP. Finally, according to Mulholland et al. (2007), the more or less significant amount of nitrogen associated with diazotrophic biomass can after mineralization in the productive layer eventually be at the origin of the establishment of a particular food web. The occurrence of diazotrophs and the associated nitrogen supply would allow the concomitant development of non-diazotrophic phytoplankton and zooplankton (Mulholland et al., 2007). A much more intensive pattern of

diatom development has been observed in MA waters (Brunet, 2016, pers. comm.). It is generally accepted that diatoms are characterized by a low PON:POP ratio (Moore et al., 2013), which could contribute to reducing the observed PON excess in comparison with the ratios measured by Karl et al. (1992) and Letelier and Karl (1998).

On an annual time scale, the strong anomalies of surface N* can be explained, on the one hand by the formation of diazotrophic organic matter, which together with the fixation of N$_2$ reduces the stock of PO$_4^{3-}$ and positively influences the N* signal. On

the other hand, part of the excess of PON which goes after remineralization towards the stock of fixed nitrogen may therefore also positively influence the N* signal. These results corroborate those of Yoshikawa et al. (2015) who suggest that nitrogen supply by remineralization of diazotroph biomass and consumption of PO$_4^{3-}$ by nitrogen fixers induces an excess of nitrogen in the WTSP east of 160°E. Nevertheless, Benavides and Voss (2015) point out that the N* signal could be overestimated by nitrogen inputs that are not attributable to the diazotrophic process, such as: mineralization of non-diazotrophic organisms with

a high PON:POP ratio, lateral advection of DOM with a high DON:DOP ratio (Loetscher et al., 2013), and atmospheric nitrogen input. Although atmospheric deposition of fixed nitrogen has tripled in the ocean since 1860 (Moore et al. 2013), atmospheric fluxes in the waters of the MA and the GY remain low (Caffin et al., this issue; Wagener et al. 2008).

The N* signal showed a significant difference between the thermocline waters of the MA and the one of the GY. As highlighted by Deutsch et al. (2001), the uptake of PO$_4^{3-}$ by diazotrophic organisms no longer occurs and therefore will have no impact on

the N* signal. In the thermocline waters of the MA, the variability of N* may be caused by remineralization, Redfieldian or





not, of organic matter and/or by the circulation and mixing of waters masses. The results showed that UTW are a preferential remineralization zone, marked by increasing AOU values (Fig. 4b) and a significant attenuation of POM concentrations (Fig.7a, 7b). One of the possible hypotheses that might explain the positive anomaly of N* in the thermocline waters observed in the MA is the nitrogen supply by remineralization of excess PON. The decrease of the N* anomaly in the thermocline waters

of the GY can be explained in various ways: $N_2$ fixation rates (Fig. 4d) and associated excess PON being almost zero (Fig. 7, Fig. 8), it is likely that the remineralization process has no or very little impact on the N* anomaly. These results can partly explain the weak anomaly observed. The concept of N* reveals its full complexity in the thermocline waters. Indeed, as pointed out by Deutsch et al. (2001), signal coupling between denitrification regions and $N_2$ fixation regions should be considered in the interpretation of the local N* signal, which is due to $N_2$ fixation and the mixing process undergone by a water parcel.

**4.2 The WTSP, a source of excess nitrogen for the thermocline waters of the South Pacific**

From a biogeochemical point of view, the thermocline waters are of a crucial importance. In the global ocean thermocline waters, the spatial distribution and concentration gradient of N* reflects the global distribution of $N_2$ fixation and denitrification (Gruber and Sarmineto, 1997; Hansell et al., 2004; Deutsch and Weber, 2012). At the global ocean scale, calculation of N* signal from the new GLODAPv2 database showed a strong spatial decoupling between the thermocline waters of the ETSP

and those of the WTSP. A strongly positive N* anomaly was observed in the thermocline waters of the WTSP in the Coral and Tasman Seas. Conversely, the OMZ region was marked by a strong negative N* anomaly.

**4.2.1 Westward transport of the negative N* signal**

This negative N* anomaly increased with water subduction centered on σ24.7 in the central Pacific (18°-20° S, 100°-150° W, Fig. 12a, Fig. S1). Donguy and Henin (1977) showed that in the South Pacific between longitude 170° W and 110° W and

latitude 20° S, surface waters have a salinity and a temperature greater than 36.8 and 26 °C, respectively. These surface and subsurface waters sink with a subduction rate of 6-7 Sv around the Polynesian region near the Societies islands (12°-25° S; 100°- 150° W) (O'Connor et al. 2002; Fiedler and Talley, 2006) under the lighter and less salty surface waters produced by an excess of precipitation over evaporation in the West. This water masse is named Northern component of Subtropical Lower Water (SLW) (Wyrtki, 1962a; Solokov and Rintoul, 2000) or the South Pacific Equatorial Water (SPEW) (Donguy and Henin,

1977; Emery and Meincke, 1986; Tomczak and Hao, 1989; Donguy, 1994) and had a density between 24.4 and 24.8 and a high salinity (>35.7) in the Coral Sea. Tomczak and Hao, (1989) and Solokov and Rintoul, (2000) showed that SLW (or SPEW) was transported eastward from the central part of the Pacific and had a strong influence on the water masses in the Coral Sea and the Solomon Sea. Solokov and Rintoul, (2000) showed that SEC transport 55 Sv toward the Coral Sea, with a transport concentrated in the upper thermocline. These authors concluded that the northern arm of the SEC transport a salty,

warm and lower oxygen water in the north of 16° S between Solomon and Vanuatu Islands and the southern branch of the SEC transport a less salty, less warm and more oxygen water in the south of 16° S between Vanuatu and New Caledonia. From OUTPACE data and the new GLODAPv2 database, our results showed that the high saltinity centered on σ24.7 (Fig. 4a) is associated with a negative N* anomaly which gradually increased from the central Pacific towards the OUTPACE region (Fig. 12a). We also found in the WTSP a front, with a strong negative N* anomaly in the north of 15° S observed on P14C, P15

sections and cruise (20), and a sharp increase of N* signal in the south of 15° S observed on P21, P14C, P15, and OUTPACE sections, (Fig. 12a).

The increase in N* (or decrease in P*=$PO_4^{3-}$-$NO_3^-$/16, Deutsch et al. 2007) towards the centre of the gyre corresponded to the trend observed by Deutsch et al. (2001; 2007) and Moutin et al. (2008). It has been shown that the thermocline waters and more particularly those located at the depth of the σ26.3 (Fig. 12b) will be also the seat of variations of N* signal induced by

the preferential transport through the SEC of the strongly negative N* anomaly from the OMZ in the ETSP. Deutsch et al.





(2001) showed that N* values below -4 µmol kg$^{-1}$ were visible up to longitude 150° W in the South Tropical Pacific despite oxygen concentrations excluding any denitrification process. These results suggest that the local N* signal on OUTPACE section reflects the combined effect of remineralization nitrogen supply and mixing processes with the strongly negative N* anomaly from OMZ.

**4.2.2 Eastward transport of the positive N* signal**

In the WTSP, the positive N* signal observed on σ24.7 and σ26.3 south of 15° S can be explained by the high levels of N$_2$ fixation observed in this region. Bonnet et al. (2017) combined the N$_2$ fixation measurements from 6 cruises in the WTSP (including OUTPACE cuise) and concluded for a hot spot of N$_2$ fixation in this region, consistent with the sharp increase of N* in the thermocline waters of the Coral Sea in the WTSP (Fig. 12a, 12b, Fig. S2).

In the south of the Coral Sea and in the Tasman Sea between Australia and New Zealand and north of latitude 40° S, the N* signal was positive in the lower thermocline water (Fig. 12b). Solokov and Rintoul, (2000) showed that in the south of latitude 25° S on P11 section (Fig. 3), the lower thermocline water was supplied by the lighter, northern variety of SLW described above which were characterized by an excess nitrogen on σ24.7. The northern SLW outcroped in Winter in the central Tasman Sea and were converted into at the dense southern variety of SLW (Solokov and Rintoul, 2000). In the Tasman Sea, there was an eastward flow in a band between 22° S and 27° S (Wyrtki, 1962). The EAC separated from the coast near 30° S into a series of filament-like eastward flows in the Tasman Sea (Ridgway and Dunn, 2003) and transport 30 Sv of SLW (Solokov and Rintoul, 2000). We hypothesize that the positive N* signal observed in the thermocline waters of the Tasman Sea could reflect the southern and then eastern transport via the EAC recirculation (Fig. 2b) (Stramma et al., 1995; Wijffels et al., 2001) of the positive N* signal observed in the MA waters and in the Coral Sea (Fig. 12).

Our results showed that the N* signal was positive in the thermocline waters of the entire South Pacific between latitude 23° S and 32° S and up to longitude 100° W. On P06-2009 section (32° S, 150° E-80° W, Fig. 3), the N* signal remained relatively high and stable on σ26.3. A sharp decrease of the N* signal was observed in the easternmost part of the basin at longitude 80°-90° W (Fig. 12b). This trend is also observed for sections P06-1992 and P06-2003 (Fig. S5e, Fig. S5f). Indeed, Solokov and Rintoul, (2000) suggest that ventilated SLW carried eastward by the EAC spread around the subtropical gyre of the South Pacific. From a linear inverse model combined with P06 cruise data, Wijffels et al. (2001) showed that 7 Sv of thermocline waters recircule west to TKR (Fig. 2a) and that the thermocline flow is predominantly zonal between 177° W and 125° W. These results suggest, on the one hand, that the denitrified waters of the southern part of the OMZ (30°-32° S, 80° W) remained confined in the eastern part of the Pacific. On the other hand, the southern branch of the subtropical gyre (Fig. 2b) was probably the main vector of excess nitrogen transport in the thermocline waters of the South Pacific.

**4.2.3 Spatial decoupling between N sources and sinks in the South Pacific?**

Deutsch et al. (2007) hypothesized from a biogeochemical model coupled with ocean circulation that a geographically close spatial coupling exists between zones of high denitrification and zones of high N$_2$ fixation. Gruber (2016) still confirms using an inverse modeling approach the hypothesis of Deutsh et al. (2007) that the coupling between denitrification and N$_2$ fixation would be spatially narrow, and concludes that N$_2$ fixation rates of the order of 500 µmol N m$^{-2}$ d$^{-1}$ are present near the Chilean coast (80°-100° W). The model results also showed N$_2$ fixation rates of the order of 280 to 400 µmol N m$^{-2}$ d$^{-1}$ in the subtropical gyre between 100° W and 160° W despite the fact that recent fixation measurements in the central and eastern part of the subtropical Pacific showed very low levels of N$_2$ fixation (Bonnet et al., 2017).

In the ETSP and the South Pacific gyre, between longitude 80° W and 100° W and latitude 20 °S, Dekaezemaker et al. (2013) measured N$_2$ fixation rates between 0 and 148 µmol N m$^{-2}$ d$^{-1}$, while Knapp et al. (2016) measured a N$_2$ fixation rate between 0 and 23 µmol N m$^{-2}$ d$^{-1}$. Moutin et al. (2008) also report extremely low N$_2$ fixation rates (≈ 0.12 nmol N L$^{-1}$ d$^{-1}$) in the



subtropical gyre waters and extremely low $N_2$ fixation rates were measured in the SW of the GY during the OUTPACE cruise (4.79-72.53 μmol N $m^{-2}$ $d^{-1}$) (Fig.11).

Currently there is therefore a clear disagreement between the model results and in situ data and geochemical calculation. Indeed, according to the first results, it seems that the spatial coupling between the strong denitrification zones and the strong $N_2$ fixation zones could exist. Nevertheless the extremely high $N_2$ fixation rates in the MA waters (Fig. 4d, Fig. 11), as well as the positive N* signal observed in the thermocline waters of the Coral and Tasman seas and in the southern part of the basin (Fig. 12, Fig. S2) suggest that the spatial coupling would occur at the basin scale through the westwards and eastward transport

of the SEC, the EAC and finally of the SPC between a High Iron/High Nitrogen/High Phosphate (HI-HN-HP) zone in the eastern part and a High Iron/Low Nitrogen/Low Phosphate (HI-LN-LP) zone in the western part of the subtropical Pacific. As hypothesized by Bonnet et al. (2017) and Moutin et al. (this issue), the spatial coupling could thus be spatially more distant than predicted by Deutsh et al. (2007) and Gruber (2016).

If the $N_2$ fixation was elusive in the GY, how will it be possible to explain the increase in N* signal of the thermocline waters

during their westward transport by the nothern branch of the SEC between latitude 5° S and 23° S. Moutin et al. (2008) argued that $N_2$ fixation is not the only process driving the increase in N* (or decrease in P*), and hypothesize that the export of material with a N:P ratio lower the RR could be at the origine of N* increase. Wijffels et al. (2001) suggested that on P06 transect west of TKR, the thermocline waters were subject to a serie of energetic northward recirculation and that the main northward flow on the thermocline waters (0-1000 m) is found between 125° W and 80° W.We hypothesize that the nitrogen excess observed

on σ26.3 in the WTSP is advected from the Tasman sea first eastward and then northward in the circulation of the gyre and could influence positively the thermocline waters of the South Pacific being thus at the origin of the westward increase of the strongly negative N* signal transported by the SEC. The northward transport of the positive N* signal could be more prononced on the one hand in the west of the TKR and on the other hand in the eastern part of the South Pacific.

This N* study in the thermocline waters of the South Pacific is based on all cruises data available in the GLODAPv2 database

updated in 2016. Using datasets that have occurred at different seasons and years and thus not taking into account seasonal and interannually variability in N* interpretation is questionable. Nevertheless, we found the same pattern from the cruises which have been repeated several times at the same location in different seasons and years (Table S1, Fig. S4, Fig. S5). A strong negative N* signal was observed on σ24.7 and σ26.3 roughly between latitude 5° S and 20°-23° S and a positive N* signal was observed in the WTSP and in the southern part of the basin roughly between latitude 23° S and 32° S (Fig. S4, Fig.

S5).

**Conclusions**

As part of the OUTPACE cruise (160° E-160° W, 18°-20° S), changes in N* signal, a tracer to visualize the combined effect of denitrification and diazotrophy on nitrogen variations, were examined in the WTSP. We revealed the presence of two oceanic regions that are quite distinct from a biogeochemical point of view. On the one hand, the waters located in the

Melanesian Archipelago (160° E-170° W) have a deficit of $NO_3^-$ and $PO_4^{3-}$ in the productive layer (0-100 dbar), an excess of PON relative to POP associated in part with diazotrophic POM formation, and a positive N* anomaly in surface and thermocline waters. This nitrogen excess over $PO_4^{3-}$ measured in the waters of the MA was observed in parallel with records rates of $N_2$ fixation. On the other hand, the surface waters of the South Pacific Gyre (170° W-160° W) are characterized by $NO_3^-$<QL, significant $PO_4^{3-}$ concentrations, a lack of PON in excess, and a N* anomaly close to zero or negative. Despite the

high concentrations of $PO_4^{3-}$ in surface waters, the $N_2$ fixation rates are close to zero in the South Pacific Gyre suggesting a limitation by DFe availability, and preventing N* signal increase.

At the basin scale, the analysis from the recent GLODAPv2 dataset (Olsen et al. 2016) highlighs a strong spatial decoupling between the thermocline waters of the ETSP and those of the WTSP. A strong positive N* anomaly is observed in the



thermocline waters of the WTSP in the Coral and Tasman Seas and in the southern part of the South Pacific, roughly between 23° S and 32 °S. We hypothesize that the southern branch of the subtropical gyre is probably the main vector of excess nitrogen transport in the thermocline waters of the South Pacific.

Finally, the $N_2$ fixation flux observed at the surface and the N* signal observed in the thermocline waters of the OUTPACE section showed a similar spatial trend. Nevertheless, the extraction of quantitative information as a budget at the regional scale from N* is not possible directly since it requires to subtract all the effects not attributable to the diazotrophic activities. To do

so, a precise description of the general and mesoscale circulation of water masses, accompanied by a detailed knowledge of their origins, their properties during formation and their mixing is needed. Only few cruises have occurred in the South Pacific Ocean and, unlike in the Atlantic Ocean, nomenclature, formation processes, circulation and mixing of water masses are still subject to debate and make interpretation from N* signal at regional scale yet difficult. Further work is required to explore differents unsolved points.

**Acknowledgements**

This is a contribution of the OUTPACE (Oligotrophy from Ultra-oligoTrophy PACific Experiment) project (https://outpace.mio.univ-amu.fr/) funded by the French research national agency (ANR-14-CE01-0007-01), the LEFE-CyBER program (CNRS-INSU), the GOPS program (IRD) and the CNES (BC T23, ZBC 4500048836). The OUTPACE cruise (http://dx.doi.org/10.17600/15000900) was managed by MIO (OSU Institut Pytheas, AMU) from Marseilles (France) and

received funding from European FEDER Fund under project 1166-39417. The authors thank the crew of the RV *L'Atalante* for outstanding shipboard operations. G. Rougier and M. Picheral are warmly thanked for their efficient help in CTD rosette management and data processing, as well as C. Schmechtig for the LEFE-CyBER database management. All data and metadata are available at the following web address: http://www.obs-vlfr.fr/proof/php/outpace/outpace.php. The authors would like to acknowledge all the voyages leaders, participants and crew involved in the Global Ocean Data Analysis Project version 2 that

were used in this study.

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





Table 1. Mean values, standard deviations (sd), and minimum and maximum values observed for all parameter in surface waters (σ21.7-σ23.5).

| | | MA | | | | GY | | | |
|---|---|---|---|---|---|---|---|---|---|
| | | mean | sd | [min | max] | mean | std | [min | max] |
| | | SW [σ21.7-σ23.5] | | | | | | | |
| $N_2$ fixation | nmol $L^{-1}$ $d^{-1}$ | 12.38 | 9.80 | [1.41 | 42.25] | 0.75 | 0.90 | [0.01 | 2.29] |
| Depth | dbar | 23.68 | 16.46 | [4.12 | 71.30] | 28.37 | 17.49 | [5.12 | 57.89] |
| $PO_4^{3-}$ | µmol $L^{-1}$ | 0.08 | 0.03 | [0.05 | 0.17] | 0.15 | 0.02 | [0.11 | 0.17] |
| $NO_3^-$ | µmol $L^{-1}$ | < QL | < QL | < QL | | < QL | < QL | < QL | |
| $NO_3^-$ : $PO_4^{3-}$ | / | < QL | < QL | < QL | | < QL | < QL | < QL | |
| N* | µmol $L^{-1}$ | < QL | < QL | < QL | | < QL | < QL | < QL | |
| POP | µmol $L^{-1}$ | 0.029 | 0.007 | [0.016 | 0.057] | 0.014 | 0.003 | [0.011 | 0.019] |
| PON | µmol $L^{-1}$ | 0.549 | 0.183 | [0.271 | 1.175] | 0.209 | 0.036 | [0.166 | 0.315] |
| PON : POP | / | 18.50 | 2.44 | [14.56 | 25.32] | 15.58 | 1.57 | [12.59 | 20.15] |
| DON | µmol $L^{-1}$ | 5.49 | 0.44 | [4.77 | 6.81] | 5.13 | 0.46 | [4.74 | 6.39] |
| DOP | µmol $L^{-1}$ | 0.16 | 0.03 | [0.10 | 0.30] | 0.19 | 0.03 | [0.16 | 0.23] |
| DON : DOP | / | 34.15 | 5.96 | [18.70 | 57.70] | 27.83 | 2.72 | [22.48 | 32] |

Table 2. mean values, standard deviations (sd), minimum and maximum values observed for all properties in the waters of the upper thermocline (σ23.5-σ25.4).

| | | MA | | | | GY | | | |
|---|---|---|---|---|---|---|---|---|---|
| | | mean | sd | [min | max] | mean | std | [min | max] |
| | | UTW [σ23.5 σ25.4] | | | | | | | |
| $N_2$ fixation | nmol $L^{-1}$ $d^{-1}$ | 0.30 | 0.49 | [0.01 | 1.73] | 0.44 | 0.40 | [0.01 | 1.18] |
| Depth | dbar | 134.37 | 53.01 | [35.42 | 251.52] | 140.85 | 53.79 | [60.73 | 249.17] |
| $PO_4^{3-}$ | µmol $L^{-1}$ | 0.24 | 0.11 | [0.05 | 0.63] | 0.28 | 0.15 | [0.11 | 0.59] |
| $NO_3^-$ | µmol $L^{-1}$ | 2.35 | 1.59 | [0.05 | 7.22] | 2.19 | 1.90 | [0.05 | 5.72] |
| $NO_3^-$ : $PO_4^{3-}$ | / | 8.09 | 3.65 | [0.36 | 13.04] | 5.49 | 3.62 | [0.36 | 9.77] |
| N* | µmol $L^{-1}$ | 1.11 | 0.53 | [-0.12 | 2.13] | -0.14 | 0.49 | [-1.13 | 0.75] |
| POP | µmol $L^{-1}$ | 0.013 | 0.007 | [0.004 | 0.036] | 0.009 | 0.005 | [0.003 | 0.019] |
| PON | µmol $L^{-1}$ | 0.210 | 0.114 | [0.070 | 0.547] | 0.151 | 0.078 | [0.048 | 0.272] |
| PON : POP | / | 16.39 | 1.48 | [13.07 | 20.04] | 16.27 | 1.05 | [14.50 | 18.65] |
| DON | µmol $L^{-1}$ | 4.45 | 0.56 | [3.24 | 5.93] | 4.36 | 0.59 | [3.20 | 5.29] |
| DOP | µmol $L^{-1}$ | 0.09 | 0.03 | [0.04 | 0.16] | 0.12 | 0.05 | [0.03 | 0.21] |
| DON : DOP | / | 51.57 | 16.98 | [29.85 | 120.25] | 42.74 | 21.25 | [23.05 | 115.00] |






Table 3. mean values, standard deviations (sd), minimum and maximum values observed for all properties in the waters of the lower thermocline (σ25.4-σ26.7).

| | | MA | | | | GY | | | |
|---|---|---|---|---|---|---|---|---|---|
| | | mean | sd | [min | max] | mean | std | [min | max] |
| | | LTW [σ25.4 σ26.7] | | | | | | | |
| Depth | dbar | 349.04 | 80.87 | [249.14 | 501.21] | 339.07 | 60.80 | [249.48 | 402.19] |
| $PO_4^{3-}$ | µmol L$^{-1}$ | 0.77 | 0.31 | [0.35 | 1.31] | 0.92 | 0.29 | [0.57 | 1.44] |
| $NO_3^-$ | µmol L$^{-1}$ | 10.07 | 4.53 | [4.13 | 18.35] | 11.33 | 4.46 | [6.42 | 18.69] |
| $NO_3^- : PO_4^{3-}$ | / | 12.92 | 0.82 | [11.33 | 14.33] | 12.06 | 1.16 | [10.64 | 14.00] |
| N* | µmol L$^{-1}$ | 0.75 | 0.54 | [-0.49 | 1.52] | -0.51 | 0.78 | [-1.55 | 0.72] |
| POP | µmol L$^{-1}$ | 0.005 | 0.001 | [0.003 | 0.008] | 0.005 | 0.001 | [0.003 | 0.006] |
| PON | µmol L$^{-1}$ | 0.084 | 0.019 | [0.051 | 0.134] | 0.067 | 0.017 | [0.050 | 0.097] |
| PON : POP | / | 16.79 | 3.19 | [12.85 | 25.78] | 14.91 | 0.92 | [13.80 | 15.80] |
| DON | µmol L$^{-1}$ | 3.50 | 0.38 | [2.77 | 4.25] | 3.45 | 0.41 | [2.78 | 4.01] |
| DOP | µmol L$^{-1}$ | 0.06 | 0.03 | [0.02 | 0.12] | 0.04 | 0.02 | [0.02 | 0.08] |
| DON : DOP | / | 80.36 | 42.37 | [31.75 | 181.50] | 100.85 | 49.18 | [40.75 | 182] |

Table 4. mean values, standard deviations (sd), minimum and maximum values observed for all properties in SAMW / AAIW (σ26.7- σ27.3).

| | | MA | | | | GY | | | |
|---|---|---|---|---|---|---|---|---|---|
| | | mean | sd | [min | max] | mean | std | [min | max] |
| | | SAMW/AAIW [σ26.7 σ27.3] | | | | | | | |
| Depth | dbar | 694.64 | 142.85 | [496.91 | 1008] | 635.82 | 130.69 | [500.75 | 801.29] |
| $PO_4^{3-}$ | µmol L$^{-1}$ | 1.88 | 0.29 | [1.37 | 2.32] | 2.03 | 0.25 | [1.67 | 2.35] |
| $NO_3^-$ | µmol L$^{-1}$ | 26.91 | 4.32 | [19.62 | 33.81] | 29.16 | 4.17 | [23.60 | 34.26] |
| $NO_3^- : PO_4^{3-}$ | / | 14.41 | 0.27 | [13.87 | 15.12] | 14.31 | 0.31 | [13.80 | 14.58] |
| N* | µmol L$^{-1}$ | -0.07 | 0.56 | [-1.13 | 1.26] | -0.47 | 0.35 | [-1.14 | -0.14] |






**Figures caption**

Fig. 1. Conceptual diagram of the different processes influencing the intensity of N*.

Fig. 2. (a) location of the OUTPACE section superimposed on a bathymetry map of the Western Tropical South Pacific Ocean, (b) map of the South Pacific and location of the OUTPACE section. The red markers correspond to the location of the stations sampled in the MA (SD 1 to LD B) and the blue markers to those sampled in the GY (SD 13 to SD 15). Large-scale ocean currents are represented by black lines (SEC=South Equatorial Current; NVJ=North Vanuatu Jet; NCJ=North Caledonian Jet; EAC=East Australian Current; NQC=North Queensland Current; STCC=South Tropical Counter Current; EAUC= East Auckland Current; SPC=South Pacific Current; PCC= Peru-Chile Current; ACC= Circumpolar Antarctic Current), OMZ= Oxygen Minimum Zone; TKR=Tonga Kermadec Ridge. The synthesis was carried out from the literature,

Fig. 3. Location of the GLODAPv2 section available for the South Pacific. The detail of sections is available in the table S1.

Fig. 4. (a) $\Theta$-S diagrams for all the 18 deep casts of the OUTPACE cruise, (b) AOU concentration ($\mu$mol kg$^{-1}$), (c) vertical profiles of chlorophyll-a concentration (mg m$^{-3}$), (d) N$_2$ fixation rates (nmol N L$^{-1}$ d$^{-1}$) *vs*. density (kg m$^{-3}$), (red marker = MA, blue marker = GY).

Fig. 5. Concentrations ($\mu$mol L$^{-1}$) of (a) NO$_3^-$, (b) PO$_4^{3-}$, (c) PON, and (d) POP along the OUTPACE transect (0-500 dbar).

Fig. 6. (a) Concentrations of NO$_3^-$, (b) PO$_4^{3-}$ ($\mu$mol L$^{-1}$) and (c) NO$_3^-$: PO$_4^{3-}$ ratio *vs*. density (kg m$^{-3}$), (d) concentrations of PO$_4^{3-}$ ($\mu$mol L$^{-1}$) *vs* NO$_3^-$ ($\mu$mol L$^{-1}$), (red marker = MA, blue marker = GY). The dashed black line shows the relationship existing if formation and remineralization of organic matter were the only processes involved and considering RR.

Fig. 7. Concentrations of (a) PON ($\mu$mol L$^{-1}$), (b) POP ($\mu$mol L$^{-1}$) and (c) PON: POP ratio *vs*. density (kg m$^{-3}$), (d) concentrations of POP ($\mu$mol L$^{-1}$) *vs*. PON ($\mu$mol L$^{-1}$), (red marker = MA, blue marker = GY). The dashed black line shows the relationship existing if formation and remineralization of organic matter were the only processes involved and considering RR.

Fig. 8. Excess PON concentration ($\mu$mol L$^{-1}$) compared to the value predicted considering the RR (dashed black line), (red marker = MA, blue marker = GY).

Fig. 9. Concentrations of (a) DON ($\mu$mol L$^{-1}$), (b) DOP ($\mu$mol L$^{-1}$) and (c) DON: DOP ratio *vs*. density (kg m$^{-3}$), (red marker = MA, blue marker = GY).

Fig. 10. (a) N* ($\mu$mol L$^{-1}$) *vs*. pressure (dbar), (b) *vs*. density (kg m$^{-3}$), (red marker = MA, blue marker = GY, cross marker =N*$_{surf}$).

Fig. 11. (a) Integrated N$_2$ fixation rates ($\mu$mol m$^{-2}$ d$^{-1}$) between the surface and $\sigma$23.5; the stations are numbered at the top of the figure, (red marker = MA, blue marker = GY), (c) longitudinal variations of the mean concentrations ($\pm$ sd) in N* ($\mu$mol kg$^{-1}$) on the isopycnes 24.7 (green line), 26.5 (gray line) measured on the OUTPACE section.





Fig. 12. Map of N* (µmol kg⁻¹) in the thermocline (a) on isopycnal surfaces 24.7, (b) on isopycnal surface 26.3. Data from GLODAPv2 database (Table S1) and from OUTPACE. The depth of isopycnal surfaces 24.7 and 26.3 are shown in the figures S1. Continuous black lines represent major large-scale currents in the South Pacific (details are available in figure 2).


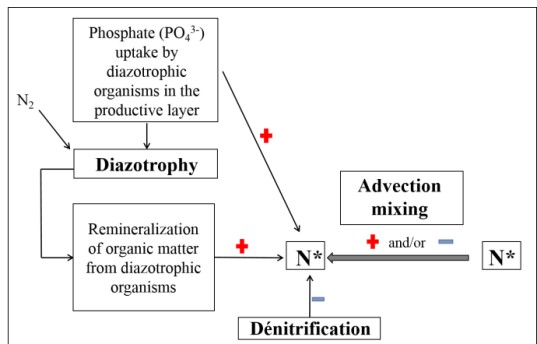

**Figure 1**


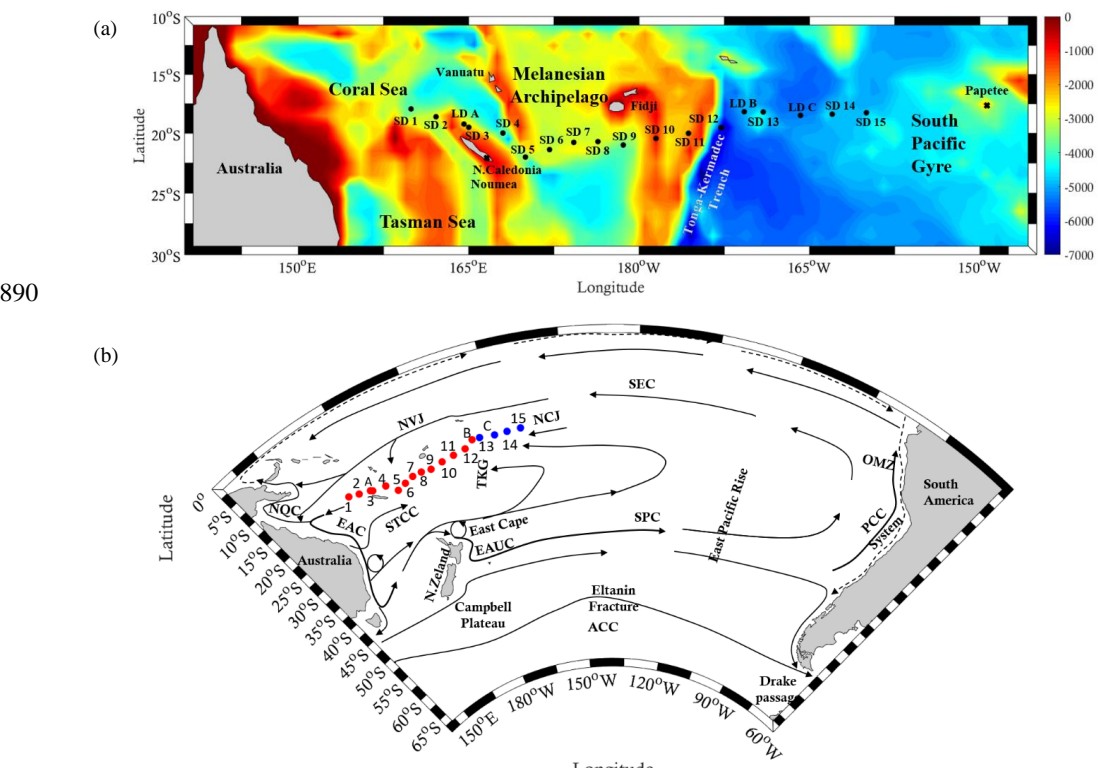

**Figure 2. a.b**





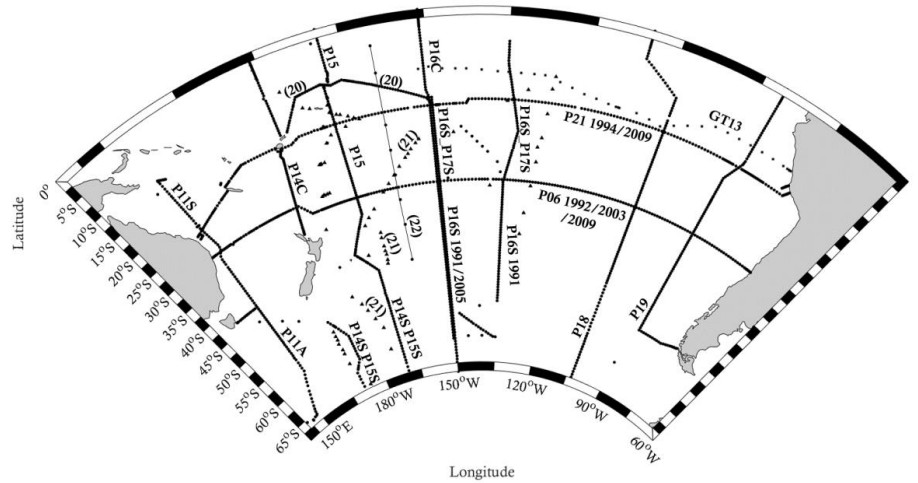

**Figure 3**





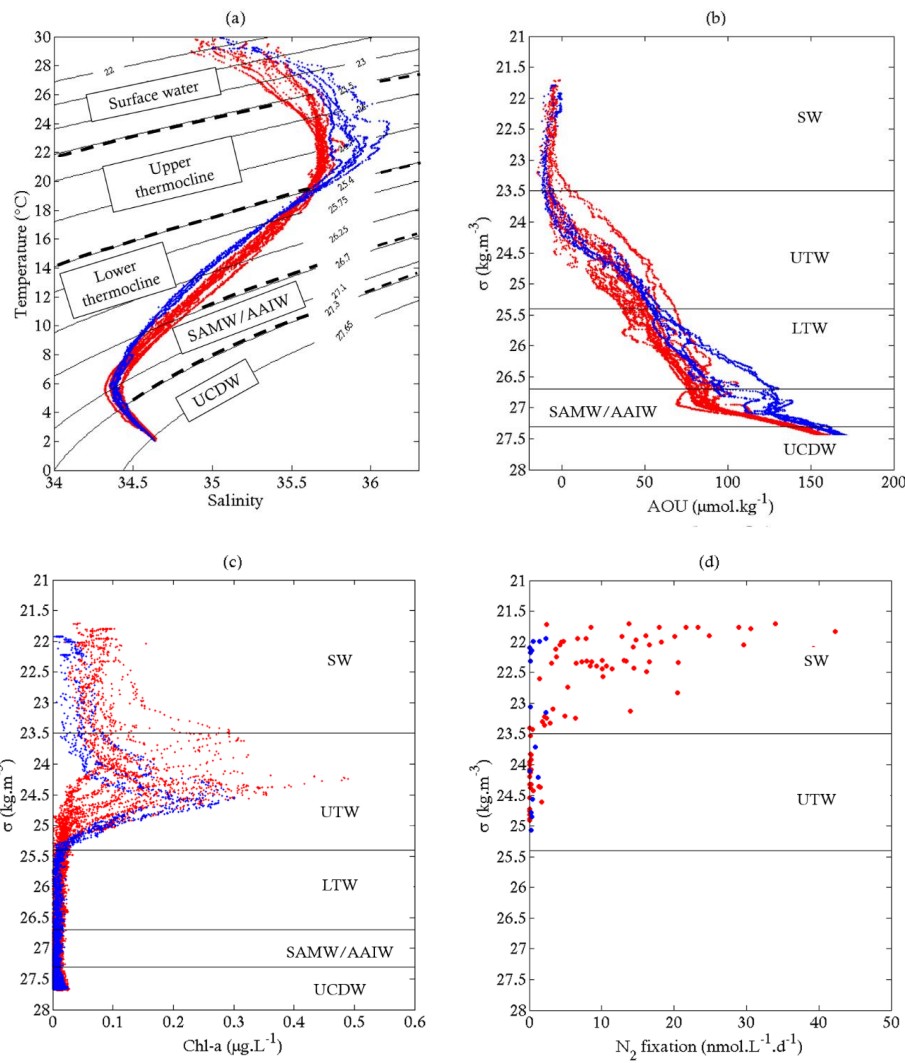

Figure 4. a.b.c.d






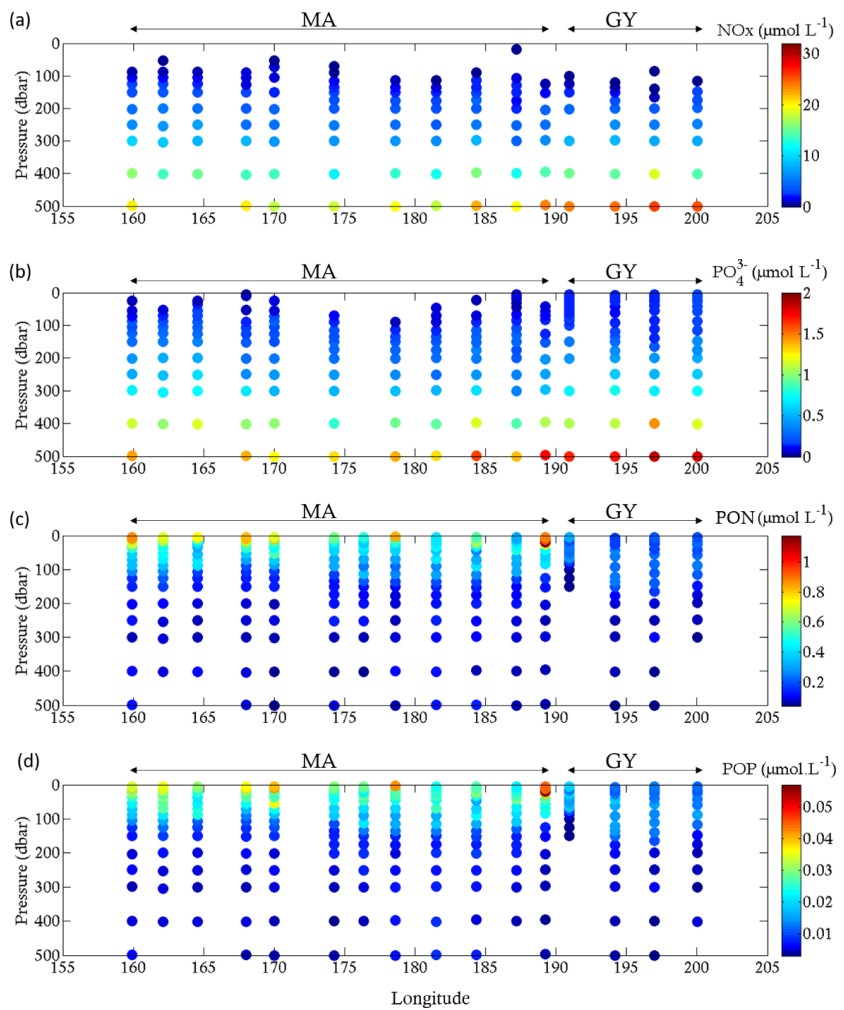

**Figure 5. a.b.c.d.**




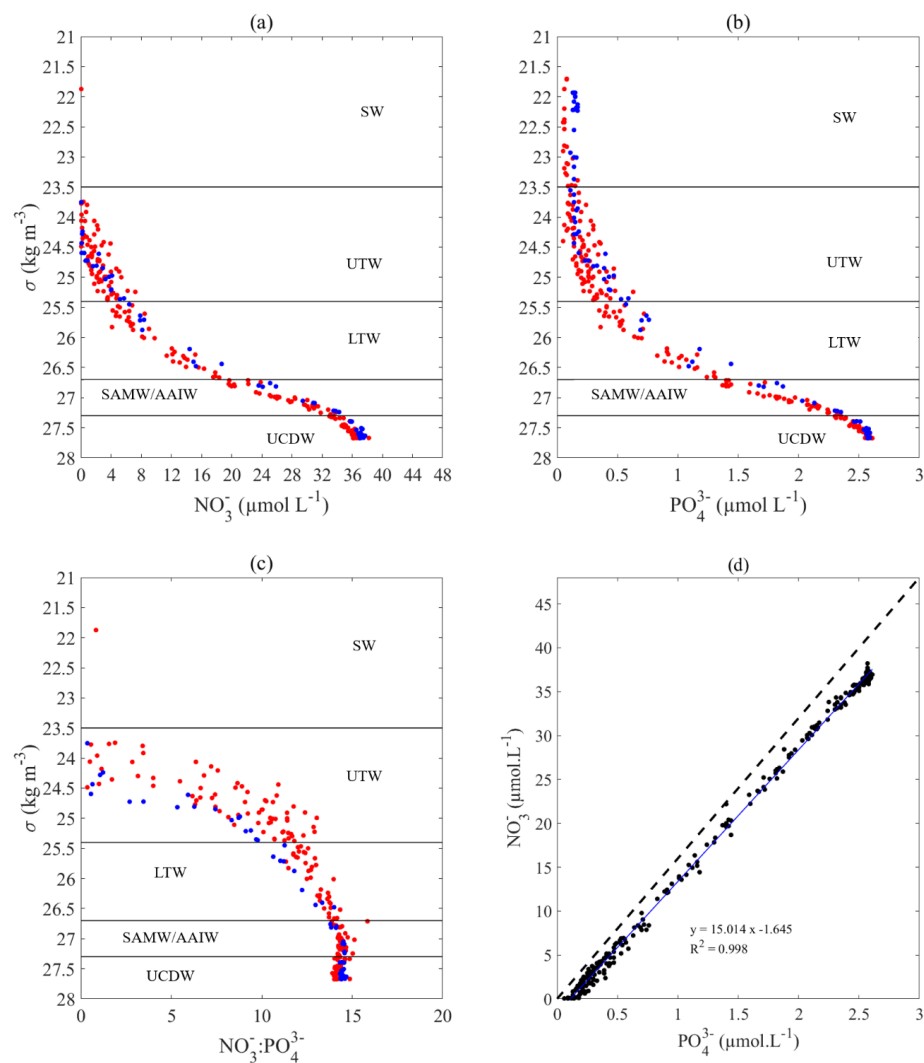

**Figure 6. a.b.c.d.**





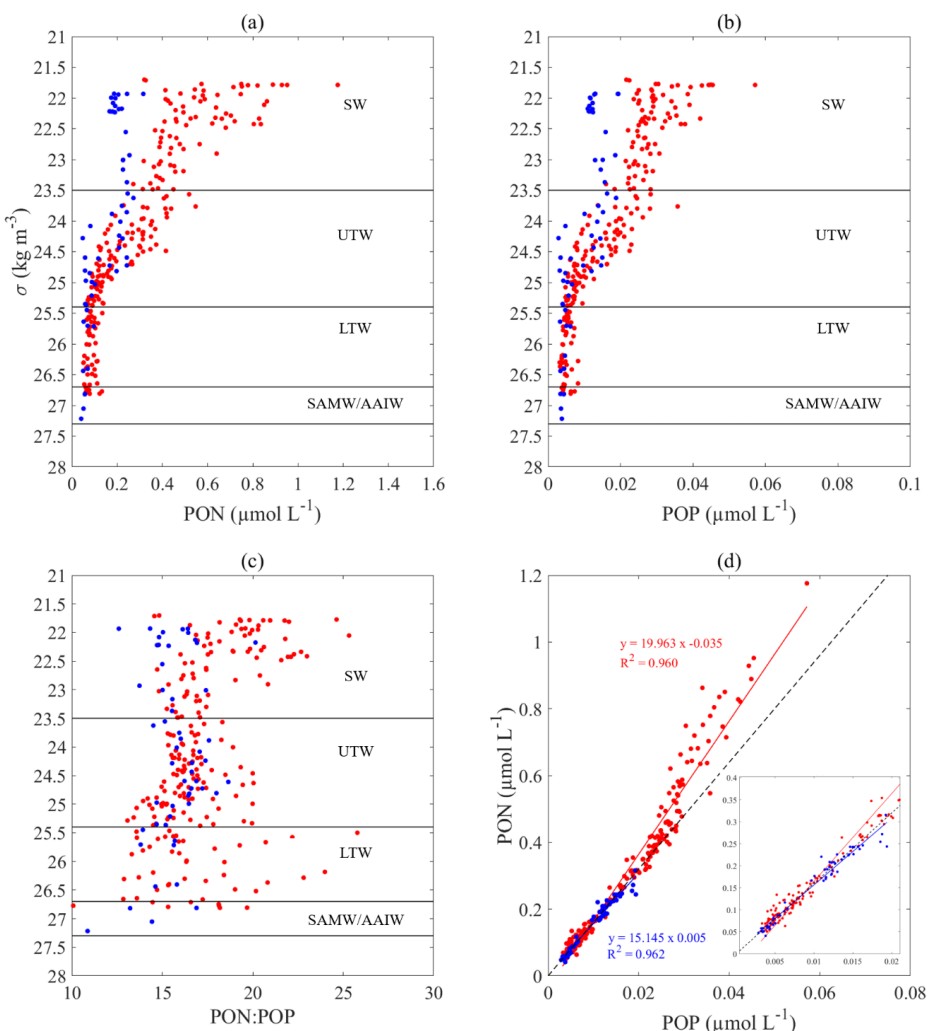

**Figure 7. a.b.c.d.**





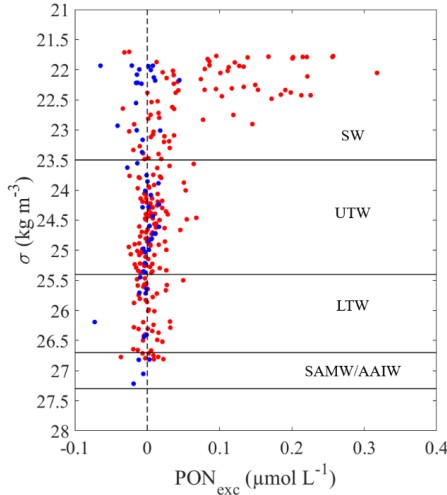

**Figure 8**



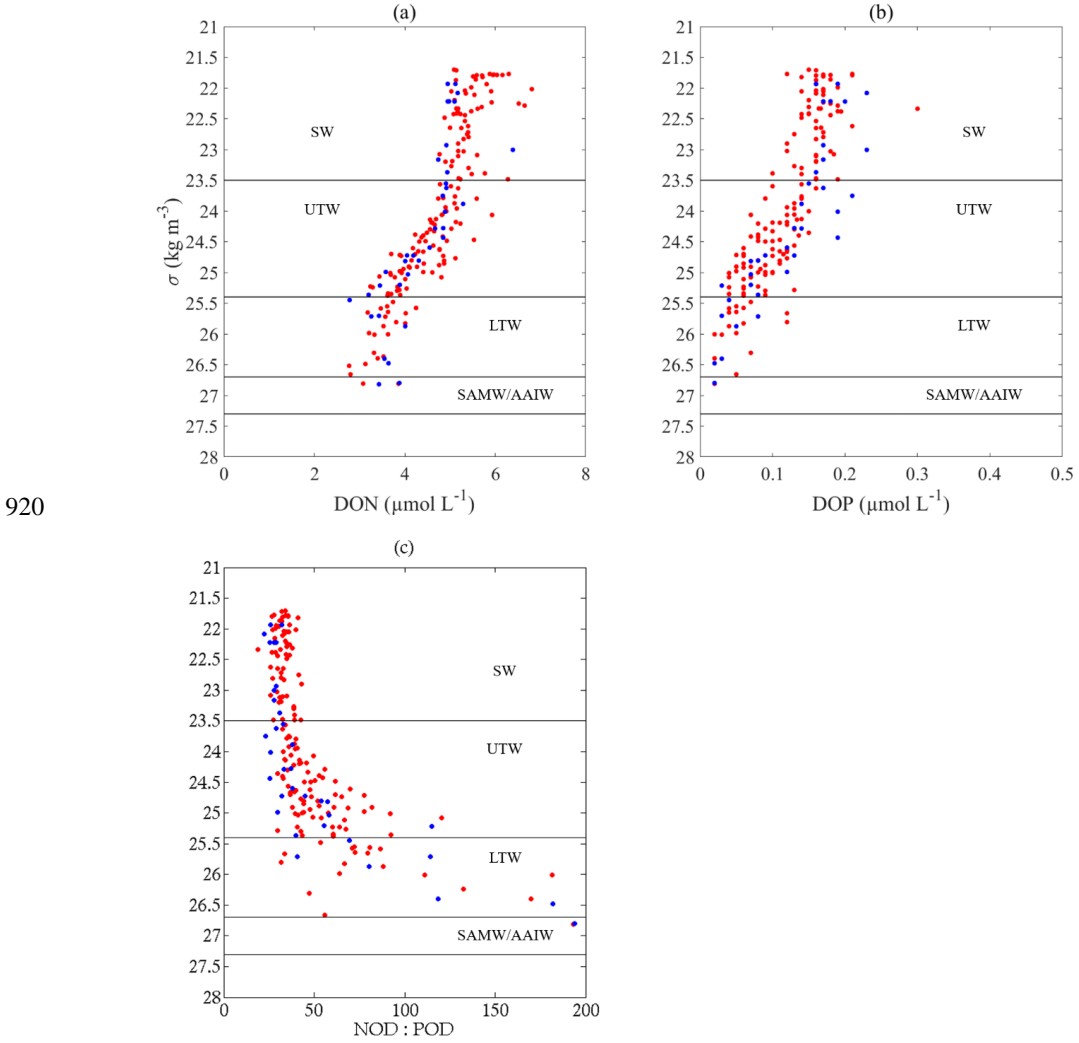


**Figure 9. a.b.c.**



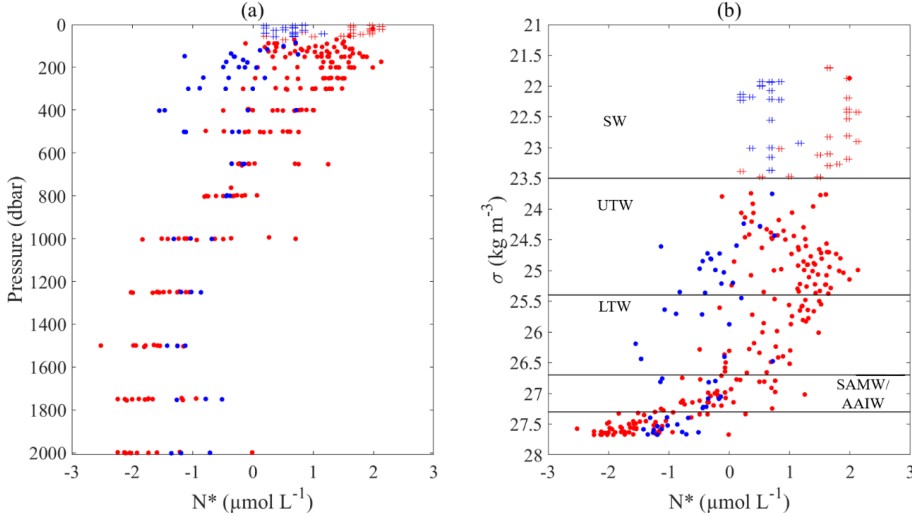

**Figure 10. a.b.**

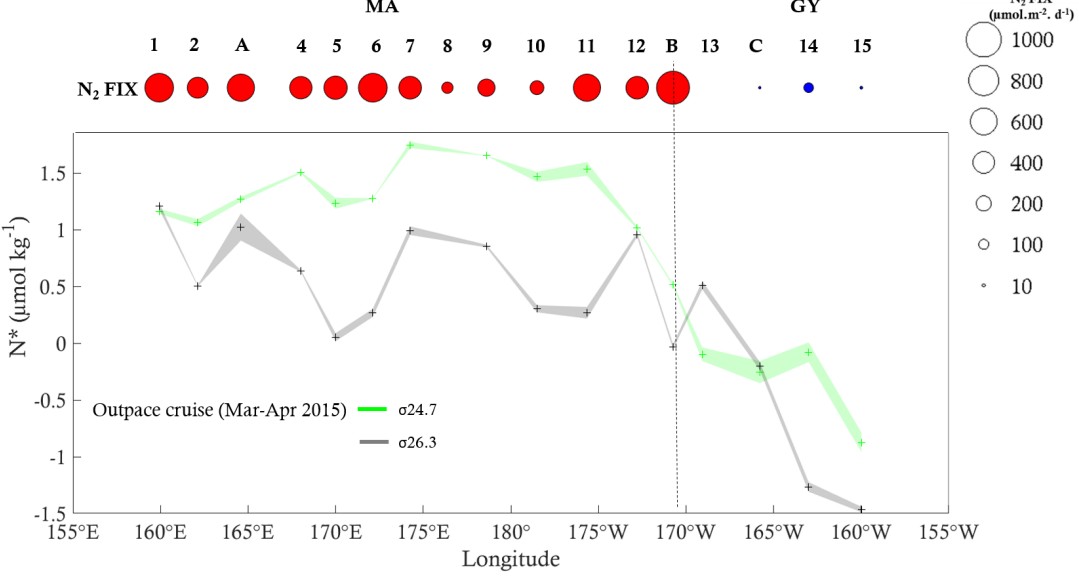

**Figure 11**


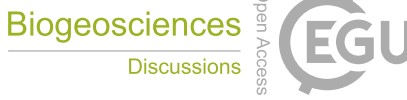



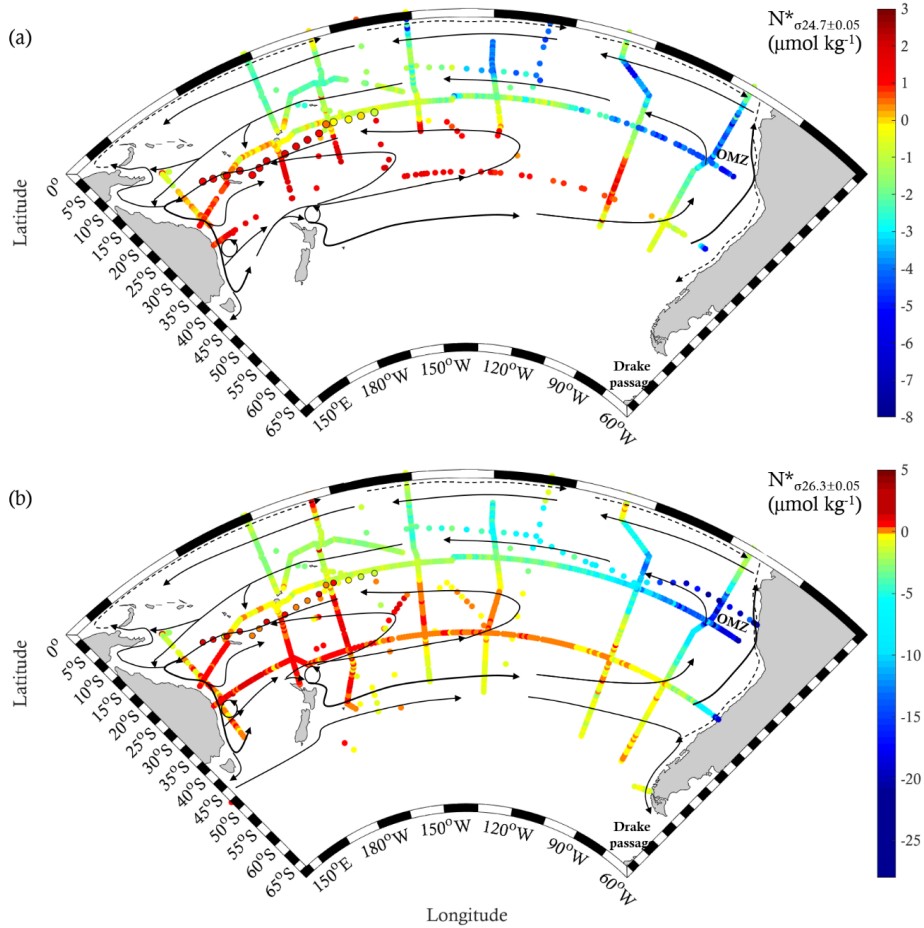

Figure 12. a.b.
