# Peer review of "Excess nitrogen as a marker of intense dinitrogen fixation in the Western Tropical South Pacific Ocean: impact on the thermocline waters of the South Pacific"

_Biogeosciences, 2017_

## Referee Comment (RC1) · Anonymous Referee #1 · 25 Feb 2018

Fumenia and coauthors have presented biogeochemical data from the OUTPACE expedition that took place in the western tropical South Pacific Ocean. Presented data are important but it lacks a robust analysis. There are too many speculative statements to list them all. I try to list some of main points below:

Authors have tried to interpret their biogeochemical data (such as N*) based on community structure (such as lines 385-388 on page 10, "In the SW, the N* variations. . . . . . . . . . . . ...). But the authors present no record of biology data, so all such statements (including the ones are PON:POP stoichiometry) remain speculative!

[Figure]

The manuscript seems like more or less a data reporting manuscript with no robust statistical analysis. All the figures and tables list some data with repetition but no rigorous analysis. For example, Figures 10, 11 and 12 are very similar to each other and it is not clear what the authors are trying to convey. What is the main hypothesis that the authors are trying to address?

Line 21 suggests positive N* while line 24 suggests N* to be zero. I presume these must be from different regions. Please specify.

Line 53: But nothing can be better than in situ measurements better! Criticism of in situ measurements should be with caution and proper justification.

Line 63: Let the equation come first before it is to be referred.

Line 83: area should be replaced with areal

Line 114: Addition of HgCl2 might degrade Organic matter and one my overestimate nutrients. Convention is just to freeze the water for nutrient analysis. Why was HgCl2 added despite that samples were frozen?

Line 140: Provide lot no for the 15N2 gas as these have been subject to contamination recently.

Why 2.90 was not added in equation 2 but in 3. Is PONexcess zero without 2.90. Reasons should be provided why is it necessary to make it zero. I believe the equation was originally given by (Gruber and Sarmiento, 1997; Michaels et al., 1996) and not by Deutsch et al (2001) as authors seem to suggest (lines 150 and 191)

If inorganic and particulate values of N and P were considered for "excess" analysis here (equations 2 and 3) then why not dissolved organic values (DON and DOP) as well? That is considered to be an equally significant pool of N and P?

Provide the reproducibility (precision) and detection limit of all the measured parameters in the method section. Can PON and POP be measured in three decimal digits of

mico-mol/L, which is almost like nmol/L in 3 decimal digits?

Line 383: there is a typo in "surface"

Lines 424-425: sentence "the . . . . . . . . ..zooplankton" is not clear

Line 428-431: " On an annual. . .. . .N* signal". See (Mills and Arrigo, 2010; Singh et al., 2017) for an alternative hypothesis.

Lines 433-436: "Nevertheless. . . . . . . . . .atmospheric nitrogen input". See (Jickells et al., 2017; Singh et al., 2013) for more details.

Line 521: how can "0" rates be measured.

Line 555: "deficit of NO3 and PO4". Mathematically (from the N* concept used here), both nutrients cannot be deficit.

Line 572: "occurred" should be replaced by "been undertaken"

Line 680: Letscher is misspelled (Letscher et al., 2013)

References:

Gruber, N., Sarmiento, J.L., 1997. Global patterns of marine nitrogen fixation and denitrification. Glob. Biogeochem. Cycles 11, 235–266.

Jickells, T., Buitenhuis, E., Altieri, K., Baker, A., Capone, D., Duce, R., Dentener, F., Fennel, K., Kanakidou, M., LaRoche, J., Lee, K., Liss, P.S., Middelburg, J.J., Moore, J.K., Okin, G., Oschlies, A., Sarin, M.M., Seitzinger, S., Sharples, J., Singh, A., Suntharalingam, P., Uematsu, M., Zamora, L.M., 2017. A reevaluation of the magnitude and impacts of anthropogenic atmospheric nitrogen inputs on the ocean. Glob. Biogeochem. Cycles 31, 289–305.

Letscher, R.T., Hansell, D.A., Carlson, C.A., Lumpkin, R., Knapp, A.N., 2013. Dissolved organic nitrogen in the global surface ocean: Distribution and fate. Glob. Biogeochem. Cycles 27, 141–153.

Michaels, A., Olson, D., Sarmiento, J., Ammerman, J., Fanning, K., Jahnke, R., Knap, A., Lipschultz, F., Prospero, J., 1996. Inputs, losses and transformations of nitrogen and phosphorus in the pelagic North Atlantic Ocean, in: Nitrogen Cycling in the North Atlantic Ocean and Its Watersheds. Springer, pp. 181–226.

Mills, M.M., Arrigo, K.R., 2010. Magnitude of oceanic nitrogen fixation influenced by the nutrient uptake ratio of phytoplankton. Nat. Geosci. 3, 412–416.

Singh, A., Bach, L.T., Fischer, T., Hauss, H., Kiko, R., Paul, A.J., Stange, P., Vandromme, P., Riebesell, U., 2017. Niche construction by non‐diazotrophs for N2 fixers in the eastern tropical North Atlantic Ocean. Geophys. Res. Lett.

Singh, A., Lomas, M., Bates, N., 2013. Revisiting N2 fixation in the North Atlantic Ocean: Significance of deviations from the Redfield Ratio, atmospheric deposition and climate variability. Deep Sea Res. Part II Top. Stud. Oceanogr. 93, 148–158.

---

## Referee Comment (RC2) · Anonymous Referee #2 · 20 Jul 2018

General: The manuscript presents measurements of dissolved inorganic, dissolved and particulate organic N and P from the OUTPACE experiment in the oligotrophic South Pacific. Elemental ratios, and in particular N*, are related to measurements of N2-fixation (from a companion paper) and other basic data from the cruise, and set into context of GLODAPv2 data from the South Pacific. Along the OUTPACE track two very different regions are identified (MA and GY waters).

The presentation of the material is highly descriptive, in particular the results section is excessive in its low to details. Overall the manuscript is a data report and not a

scientific paper.

Large parts of the text (Intro and Discussion in particular) are characterized by awkward and ambigious sentences. This partly seems to be a language problem, hence I strongly suggest to make use of appropriate editing from a person (co-author ?) with appropriate knowledge of the subject and the English language. Please note that the review process is not a substitution of proper preparation of a manuscript.

I provide an annotated manuscript, with 540+ comments and corrections. This is by far the largest number of annotations I ever had with a manuscript. Given this huge number of issues, there is no meaning to select major vs. minor points.

In the current form the manuscript can only be rejected, with the possibility to resubmit to BG. I suggest that the authors collectively work on an improved new version of this manuscript.

A few overall things, just that they are not lost in front of the huge number of comments in the annotated file.

I strongly suggest to add a table with accronyms to the paper since you use them a lot. Also make sure to keep accronyms free of typos, e.g. p13 TKR instead of TKG (Fig. 2b)

Results: This section is unreadable. You repeat basically every single number you show in the tables 1-3. Reduce to 1/3 or less. Point the reader to patterns, differences. Otherwise the paper your paper has the nature of a data report and not a scientific paper. BG does not publish data reports.

Discussion This section is characterized by many awkward and ambigious sentences.

P13, 516ff. This is clearly a wrong interpretation of the Gruber 2016 paper!! Please read and understand a paper before citing it. Gruber (2016) is a commentary on the paper by Knapp et al. from the same PNAS issue. Gruber does not provide any new results, but just introduces the uncertainties related to both the Deutsch (inverse model) study and the Knapp observational study. The results of Knapp should be cited here, which are in line with Bonnet et al. 2017. Don't be misguided by the tone of the Gruber paper, which tries to rescue the 'spatial proximity' hypothesis. But this is only fort he reason that he himself introduced the general hypothesis in a paper in 2001.

Please also note the supplement to this comment:
https://www.biogeosciences-discuss.net/bg-2017-557/bg-2017-557-RC2-supplement.pdf

―――――――――――――――――――――

**Supplement:**

[revised manuscript text omitted]

---

## Author Comment (AC1) · 18 Oct 2018

**Responses to Review #1**

We thank Reviewer #1 for taking the time to read the submitted manuscript and for his/her comments. Below is the original response from Reviewer 1 (in italic/bold), with our own responses interspersed within. The original text of the manuscript is written in blue typefaceand the changes to the text are shown in red.

1. ***Fumenia and coauthors have presented biogeochemical data from the OUTPACE expedition that took place in the western tropical South Pacific Ocean. Presented data are important but it lacks a robust analysis. There are too many speculative statements to list them all. I try to list some of main points below***

Response: We thank Reviewer#1 for mentioning the important dataset presented in our manuscript. Reviewer#1 emphasizes the lack of a robust analysis. The dataset presented in our manuscript is supported by a robust chemical analysis. Only the chemical data ($NO_3^-$, $PO_4^{3-}$) above the quantification limit (QL = 0.05 µM) were taken into account to calculate the derived ratio (N*). The same remark is valid for the chemical data, including PON, POP and the derived ratio PON:POP. Furthermore, high analytical repeatability was found for $NO_3^-$, $PO_4^{3-}$ and for organic particulate matter (see tables R4 below). Therefore, the increases in both PON, PON:POP and N* observed in the Melanesian Archipelago surface waters are supported by robust chemical analysis. Concerning data extracted from the GLODAPv2 database, only data which have passed the two-quality controls, as recommended by Olsen et al. (2016), have been used in this study. Furthermore, a statistical analysis (t-test and Pearson test) has been added in the revised text of the manuscript (see the details below in the answer to comment 3).

**General comment:** The title of our paper has been rephrased.

Modified text:

The effect of $N_2$ fixation on the thermocline waters of the South Pacific Ocean traced by excess nitrogen

2. *Authors have tried to interpret their biogeochemical data (such as N\*) based on community structure (such as lines 385-388 on page 10, "In the SW, the N\* variations. . .. . .. . .. . ...). But the authors present no record of biology data, so all such statements (including the ones are PON:POP stoichiometry) remain speculative!*

3. *The manuscript seems like more or less a data reporting manuscript with no robust statistical analysis.*

Original text (lines 385-388): "If in the SW Pacific, the N\* variations are solely due to diazotrophic activity, these results suggest that the decrease in $PO_4^{3-}$ and the associated increase in N\* in the MA waters are controlled by diazotrophic organisms. The high concentrations of $PO_4^{3-}$ in the SW of the GY suggest a low use by diazotrophs, as $N_2$ fixation is hypothesized to be limited by dissolved iron (DFe) availability in GY waters (Moutin et al., 2008; 2018; Blain et al., 2008; Guieu et al., 2018) maintaining a low N\* anomaly in this region.

Response: We apologize for this confusing section (4.1.1). The whole of the paragraph above has been deleted from the revised manuscript. In the revised text, the Discussion has been rewritten in a clearer and more concise way. The Discussion has been reviewed, and awkward and ambiguous sentences corrected. Section 4.1.1 has been deleted, and we have focused the Discussion on the effect of intense $N_2$ fixation on the particulate organic N pools, and on the N\* distribution, first in the WTSP, and then in the thermocline waters at the scale of the South Pacific Ocean.

But we disagree with the reviewer concerning the speculative statements of our study. Our results show significant $N_2$ fixation rates in the surface waters and at the top of the upper thermocline waters of the Melanesian Archipelago. In agreement with the high diazotrophic activity observed in the region, relatively high PON concentrations (0.32-1.18 µmol kg$^{-1}$), an excess of PON concentration relative to that expected from Redfield stoichiometry, and positive N\* in the surface waters and in the upper and lower thermocline waters of the Melanesian Archipelago were observed. In contrast, the surface waters and the top of the upper thermocline waters of the South Pacific Gyre (170° W-160° W) were characterized by low $N_2$ fixation rates and low PON concentrations.

As requested by Reviewer#1, a statistical analysis (Pearson test) has been added. We showed a significant relationship between $N_2$ fixation rates and PON in excess (r²=0.86, p<0.001), and between $N_2$ fixation rates and N\* values (r²=0.60, p<0.001). Our results clearly show that the intense $N_2$ fixation observed in the WTSP significantly influences the dissolved inorganic (N excess) and particulate organic (PON excess) N pools of the surface and thermocline waters in the WTSP. Following an analysis of South Pacific nutrients database (GLODAPv2), we find a strong positive N\* signature in the thermocline waters of the WTSP, the Coral/Tasman Seas, and the southern part of the subtropical gyre between 23° S and 32° S. Together with a basin-scale analysis of water mass circulation, we

hypothesize that the outstandingly high $N_2$ fixation rates observed at the surface influence the N status of thermocline waters across the whole South Pacific Ocean.

Several additional statistical tests have been added to provide a robust analysis of our data, as requested by Reviewer#1. The equality of mean concentrations in the Melanesian Archipelago and in the gyre was tested as follows. Firstly, a Fisher-Snedecor test was used to establish the equality of variances. Secondly, a Student's test (level significance $p = 0.01$) was used when variances were equal and Satterthwaite's approximate t-test was performed otherwise. The results are presented in the supplementary material for surface waters, the UTW and the LTW (see tables R1, R2 and R3 below).

Table R1. Results of Student's test or Satterthwaite's approximate t test (marked by a red asterisk) in the surface waters of the Melanesian Archipelago (MA) and of the gyre (GY); t is the t value, DF is the degree of freedom, p is the p-value. The results marked by a black asterisk have a p-value < 0.01, n is the number of samples for the MA and the GY areas.

| | | Surface Waters | | | |
|---|---|---|---|---|---|
| | | n | t | DF | p < 0.01 |
| $N_2$ fixation | MA/GY | 57/10 | 9.16* | 60.75 | * |
| $PO_4^{3-}$ | MA/GY | 19/20 | 7.18* | 24.97 | * |
| $N^*_{surf}$ | MA/GY | 20/21 | 7.88* | 28.24 | * |
| PON | MA/GY | 77/20 | 15.03* | 92.31 | * |
| POP | MA/GY | 77/20 | 15.49* | 80.53 | * |
| PON : POP | MA/GY | 77/20 | 5.08 | 95.00 | * |
| NOPexcess | MA/GY | 77/20 | 8.22* | 94.90 | * |

Table R2. Results of Student's test in the UTW of the Melanesian Archipelago (MA) and of the gyre (GY), t is the t value, DF is the degree of freedom, p is the p-value. The results marked by a black asterisk have a p-value < 0.01, n is the number of samples for the MA and the GY areas.

| | | UTW | | | |
|---|---|---|---|---|---|
| | | n | t | DF | p < 0.01 |
| $N_2$ fixation | MA/GY | 24/7 | 0.70 | 29 | |
| NO3- | MA/GY | 70/18 | 1.24 | 86 | |
| $PO_4^{3-}$ | MA/GY | 85/25 | 1.49 | 86 | |
| N :P | MA/GY | 70/17 | 2.60 | 85 | * |
| N* | MA/GY | 70/17 | 8.95 | 85 | * |
| PON | MA/GY | 97/25 | 2.48 | 120 | * |
| POP | MA/GY | 97/25 | 2.40 | 120 | * |
| PON : POP | MA/GY | 97/25 | 0.39 | 120 | |

Table R3. Results of Student's test in the LTW of the Melanesian Archipelago (MA) and of the gyre (GY), t is the t value, DF is the degree of freedom, p is the p-value. The results marked by a black asterisk have a p-value < 0.01, n is the number of samples for the MA and the GY areas.

| | | LTW | | | |
|---|---|---|---|---|---|
| | | n | t | DF | p < 0.01 |
| NO3- | MA/GY | 38/9 | 0.76 | 45 | |
| $PO_4^{3-}$ | MA/GY | 38/9 | 1.39 | 45 | |
| N :P | MA/GY | 38/9 | 2.64 | 45 | * |
| N* | MA/GY | 38/9 | 5.72 | 45 | * |
| PON | MA/GY | 39/6 | 1.96 | 43 | |
| POP | MA/GY | 39/7 | 1.10 | 44 | |
| PON : POP | MA/GY | 39/6 | 1.42 | 43 | |

**4. *Reviewer: All the figures and tables list some data with repetition but no rigorous analysis.***

Response:

We agree with Reviewer#2 regarding the excessive repetition and excessive details provided in the manuscript. In the revised paper, the Results section has been rewritten in a clearer and more concise way. Furthermore, the manuscript has been shortened and additional statistical tests were taken into account in the interpretation of our results. In addition, the aims of the study have also been clarified and new scientific knowledge highlighted.

**General comment:** We propose a new version of the results section (3.1, 3.2 and 3.3) in the revised manuscript.

Modified text:

**3.1 Water masses and general biogeochemical trends**

Figure 2a shows Θ-S diagrams (0-2000 dbar) for all 18 stations sampled during the OUTPACE cruise. The temperature range observed on the OUTPACE section varied from 29.9 °C at the surface to 2.0 °C at 2000 dbar depth (Fig. 2a). Salinity varied from 34.3 to 36.1 (Fig. 2a). The MLD ranged between σ21.9 and σ22.6, with an average value of 22.2 ± 0.2 kg m$^{-3}$ (corresponding to 19 ± 9 dbar), (Fig. 2b). The temperature and salinity observed in the surface waters were 29.4 ± 0.4 °C and 35.1 ± 0.2, respectively (Fig. 2a). A salinity maximum ($S_{max}$) was observed in the UTW between σ24 and σ25.4, centered on σ24.7 (Fig. 2a). Between σ25.4 and σ26.7 (249-501 dbar, Table 3), a strong decreasing gradient of temperature and salinity marked the presence of the LTW.
In the surface waters of the Melanesian Archipelago, $N_2$ fixation rates ranged from 2.4 to 42.2 nmol N L$^{-1}$ d$^{-1}$ (average rate 15.4 ± 10.4 nmol N L$^{-1}$ d$^{-1}$; Fig.3, Table 1). Rates decreased drastically towards the

gyre waters, with values ranging between <0.01 to 2.3 nmol N $L^{-1}$ $d^{-1}$ (Fig.3, Table 1). $N_2$ fixation rates showed high values at the top of the UTW above σ23.5 in the Melanesian Archipelago, and homogeneous low values between σ23.5 and σ25.4 (Fig.3). Conversely, the $N_2$ fixation rates were very low in the UTW of the gyre waters (Fig.3; Table 2). Below σ25.4, the $N_2$ fixation rates were undetectable (Fig.3; Table 2).

**3.2 Inorganic nutrient pools**

The concentrations of $NO_3^-$ were below the quantification limit in the surface waters for all sampling stations (Fig. 4a). The concentrations of $PO_4^{3-}$ were below the quantification limit or low in the surface waters of the Melanesian Archipelago, with a mean value (for measurable samples) of $0.07 \pm 0.01$ µmol $L^{-1}$ (Fig. 4b, Table 1). Significant higher $PO_4^{3-}$ concentrations (p < 0.01, Table S3) were observed in the surface waters of the gyre, with a mean value of $0.15 \pm 0.01$ µmol $L^{-1}$ (Fig. 4b, Table 1). The $NO_3^-$ concentrations became quantifiable below σ23.5. The $NO_3^-$ and $PO_4^{3-}$ concentrations increased gradually in the upper and lower thermocline waters. The bottom waters (> σ27.5, 1000 - 2000 dbar, 2000 dbar corresponding to the deeper depth sampled) had relatively homogeneous maximum concentrations, with values ranging between 35.1 and 36.5 µmol $L^{-1}$ and between 2.40 and 2.55 µmol $L^{-1}$ for $NO_3^-$ and $PO_4^{3-}$, respectively (Fig. 4a; Fig.4b).

The $NO_3^-$: $PO_4^{3-}$ ratio increased sharply in the UTW, and was significantly higher (p < 0.01, Table S4) in the Melanesian Archipelago than in the gyre (Fig. 4c, Table 2). In the LTW, the $NO_3^-$: $PO_4^{3-}$ ratio increased slightly and was significantly higher (p < 0.01, Table S5) in the Melanesian Archipelago than in the gyre (Fig. 4c, Table 3). In deep waters (below 27.5), the $NO_3^-$: $PO_4^{3-}$ ratio reached stable and uniform values, ranging between 13.8 and 15.2 (Fig. 4c). During the OUTPACE cruise, the distribution of $NO_3^-$ *versus* $PO_4^{3-}$ concentrations followed a linear regression ($r^2 = 0.99$) with a slope of 15.0, which was close to the Redfield ratio of 16:1 (Fig. 4d). In the upper waters, $NO_3^-$ concentrations become depleted before those of $PO_4^{3-}$. In this layer, a deficit in $NO_3^-$ compared to the value expected from a Redfield point of view (dashed black line) was observed for all data (Fig. 4d).

**3.3 Organic pools**

The concentrations of PON and POP showed maximum values in the surface waters of the Melanesian Archipelago, with values ranging between 0.32 and 1.18 µmol $L^{-1}$, and between 0.02 and 0.06 µmol $L^{-1}$, respectively. They were significantly higher (p < 0.01, Table S3) in the Melanesian Archipelago than in the gyre (Fig. 5a, Fig. 5b; Table 1). In the UTW, a significant attenuation of POM concentrations was observed. The mean concentrations of PON and POP remained, however, significantly higher (p < 0.01, Table S4) in the UTW of the Melanesian Archipelago than those of the gyre (Fig. 5a, Fig. 5b, Table 2).

POM concentrations decreased to reach minimum and relatively constant values in the LTW. A significant relationship (p<0.001; Pearson test) was obtained between PON concentrations and $N_2$ fixation ($r^2$=0.56). The PON concentrations in excess showed a contrasted distribution in the surface waters of the studied areas. Indeed, in the Melanesian Archipelago, the highest values were observed between the surface and the top of the UTW ($< \sigma22.5$, $\approx$ 0-25 dbar), with a maximum of 0.37 µmol $L^{-1}$ (Fig. 5c). In the gyre, the PON concentrations in excess exhibited values around zero. Below $\sigma23.5$, the PON concentrations in excess showed a similar distribution in both areas studied, with values close to those calculated based on Redfield (Fig. 5c). A significant relationship (p<0.001; Pearson test) was obtained between PON concentrations in excess and $N_2$ fixation ($r^2$=0.86). For all the sampling stations, the PON:POP ratio was between 10.08 and 25.78 (figure not shown). The average value of the PON:POP ratio was 16.9 ± 2.4, which included the Redfield ratio (16.0). Nevertheless, a strong variability was observed between the different oceanic regions (Melanesian Archipelago *vs* gyre) and the different water masses considered. In the surface waters, the PON:POP ratio was significantly higher ($p < 0.01$, Table S3) in the Melanesian Archipelago than in the gyre (Table 1). In the UTW, the PON:POP ratio was not significantly different (Table S4) between the Melanesian Archipelago and the gyre. The distribution of PON *versus* POP concentrations followed a linear regression in both studied areas. In the gyre, a slope of 15.1 was observed, which was close to the Redfield ratio (blue line in Fig. 4d). In the Melanesian Archipelago, the slope observed deviates from the expected Redfield ratio, showing a value of 20.0 (red line in Fig. 5d).

5. *For example, Figures 10, 11 and 12 are very similar to each other and it is not clear what the authors are trying to convey. What is the main hypothesis that the authors are trying to address?*

Response: Figures 10, 11 and 12 of the original manuscript presented the N*, but they do not all have the same objective. In figure 10, we showed the N* observed in the water column of the Melanesian Archipelago during the cruise. Figure 11 indicated both the integrated $N_2$ fixation rates and the N* variations on two isopycnals (24.7 and 26.3), which are characteristics of the upper and lower thermocline waters, respectively. The aim of figure 11 is to show the exact location of the N* increase in the thermocline. We found a strong increase of N* in the thermocline waters of the Melanesian Archipelago, an area strongly subject to $N_2$ fixation in its surface layer. To visualize the input of N by the high $N_2$ fixation observed in the Melanesian Archipelago, we used the quality dataset (GLODAPv2; http://cdiac.ornl.gov/oceans/GLODAPv2/). Figure 12 presented the large-scale ocean transport of thermocline waters, obtained from the literature. The analyses of the most recent available nutrient data in the South Pacific Ocean enabled us to show a high positive N* in the thermocline waters of the whole South Pacific gyre. We consider that a major result is our hypothesis that the local high N* observed in

the Melanesian Archipelago is transported eastward *via* the circulation, and consequently that $N_2$ fixation in the WTSP may influence the nutrient distribution at the scale of the South Pacific basin.

**General comment:** In the revised manuscript, the Results sections (3.4 and 3.5) have been rewritten in a clearer and more concise way. We propose a new version in the revised manuscript.

Modified text:

**3.4 Regional distribution of N\* and $N_2$ fixation rates**

At the scale of the OUTPACE cruise, N\* showed positive values ranging between 0.23 and 2.15 µmol $L^{-1}$ in the surface waters. N\* was significantly higher ($p < 0.01$, Table S3) in the Melanesian Archipelago than in the gyre (Fig. 6). In the UTW and the LTW (100-500 dbar), N\* showed a significant difference ($p < 0.01$, Table S4 and S5) between the Melanesian Archipelago and the gyre waters. In the UTW, N\* mean was positive in the Melanesian Archipelago and negative in the gyre (Fig. 6b, Table 2). In the LTW, below 150 dbar depth, N\* exhibited a marked decrease, with a minimum value of -1.55 µmol $L^{-1}$ at 400 dbar in the gyre. In the same layer, N\* was significantly higher ($p < 0.01$, Table S5) in the Melanesian Archipelago than in the gyre (Fig. 6b, Table 3), with a minimum value of -0.77 µmol $L^{-1}$ at 500 dbar (Fig.6a). Below 500 dbar, N\* variability decreased with depth and reached mostly negative values at 2000 dbar for all sampling stations (Fig. 6a).

Regarding the horizontal variability of N\*, over the upper ($\sigma$24.7) and lower ($\sigma$26.3) thermocline layers (Fig. 7), N\* showed higher values in the Melanesian Archipelago than in the gyre. In the Melanesian Archipelago (between SD1 and SD12), N\* ranged between $1.05 \pm 0.01$ and $1.77 \pm 0.02$ µmol $kg^{-1}$ and between $-0.02 \pm 0.02$ and $1.21 \pm 0.02$ µmol $kg^{-1}$ in the upper and lower thermocline layers, respectively. Beyond SD 12, a drastic decrease of N\* was observed in both upper and lower thermocline layers (between SD12 and SD15), reaching minimum values of $-0.68 \pm 0.07$ µmol $kg^{-1}$ and $-1.46 \pm 0.02$ µmol $kg^{-1}$ in the upper and lower thermocline layers, respectively (Fig. 7).

In this study, a significant relationship ($p < 0.001$; Pearson test) was obtained between N\* and $N_2$ fixation ($r^2=0.59$). Vertically integrated $N_2$ fixation rates in the surface waters exhibited the same trend as N\*, with higher values in the Melanesian Archipelago than in the gyre. In the Melanesian Archipelago, $N_2$ fixation rates ranged between 100 and 840 µmol N $m^{-2}$ $d^{-1}$, with an average integrated $N_2$ fixation rate of $443 \pm 204$ µmol N $m^{-2}$ $d^{-1}$ (red circle in Fig. 7). As for N\*, a sharp decrease in integrated $N_2$ fixation rates was observed in the gyre, with values ranging between 5 and 73 µmol N $m^{-2}$ $d^{-1}$ and with an average value of $28 \pm 39$ µmol N $m^{-2}$ $d^{-1}$ (blue circle in Fig. 7).

**3.5 Large scale distribution of N* in the South Pacific**

At the basin scale of the South Pacific, N* was minimum in the upper thermocline ($\sigma$24.7) between 5° S and 20° S, and between 80° W and 100° W (P21, P18, P19 sections in Fig. 8a). Between 5° S and 15° S, N* showed a westward increase along the northern branch of the SEC. In the WTSP, N* increased and exhibited maximum values at 155° E, with an average value of -0.27 ± 0.97 µmol kg$^{-1}$ (P11S section in Fig. 8a). South of 15° S, N* increased more rapidly than north of 15° S. In the WTSP, between 175° E and 150 °E, N* increased sharply during the westward transport of waters by the NCJ (Fig. 1b), reaching a value of 0.39 ± 0.58 µmol kg$^{-1}$ (P21 section in Fig. 8a). The maximum of N* was observed in the Coral and Tasman seas, between 23° S and 30° S, reaching a value of 1.86 ± 0.19 µmol kg$^{-1}$ (P15 section in Fig. 8a).

As observed in the upper thermocline, N* showed minimum values in the lower thermocline around the OMZ region, between 80° W and 90° W and at 18° S (P21, GT13 sections in Fig. 8b, Fig. S2), and increased westward. The maximum values of N* were observed in the Melanesian Archipelago waters, in the southern Coral and Tasman seas, exhibiting an average value of 0.52 ± 0.12 µmol kg$^{-1}$ (P14C section in Fig. 8b, Fig. S2). In the Tasman Sea and north of New Zealand, at 30° S, and between 150 °E and 180 °E, N* showed high values, with an average of 0.66 ± 0.46 µmol kg$^{-1}$ (P06 section in Fig. 8b, Fig. S2). These high N* values persist in the central part of the South Pacific, between 23° S and 32° S and 175° E and 135° W, with an average value of 0.42 ± 0.31 µmol kg$^{-1}$. In the eastern part of the South Pacific, at 110° W and between 85° W and 90° W (P18, P19 sections in Fig. 8b, Fig. S2), a drastic decrease of N* was observed. At 32° S and close to the South America coast, N* dropped sharply to reach an average value of -17.33 ± 2.63 µmol kg$^{-1}$ (P06 section in Fig. 8b, Fig. S2).

6. *Reviewer: Line 21 suggests positive N\* while line 24 suggests N\* to be zero. I presume these must be from different regions. Please specify.*

Original text (lines 21-22): A positive N* anomaly was observed in the productive layer reflecting the combined effect of phosphate uptake by diazotrophic organisms and remineralization of excess particulate organic nitrogen. The South Pacific Gyre waters between 170° W and 160° W were depleted in nitrate but rich in phosphate. Surface waters exhibited very low dinitrogen fixation rates, an absence of excess particulate organic nitrogen and a N* signal close to zero.

Response: The positive N* anomaly observed in the productive layer described in this sentence refers to the Melanesian Archipelago (160° E- 170° W) (red marker on Figure 2b of the original manuscript for the positions of station; red markers on Figure10b of the original manuscript for the values of positive N* anomaly). The N* signal close to zero refers to the South Pacific Gyre surface waters (blue markers,

Figure 10b for the values of N* signal close to zero). To avoid any confusion, the sentence has been rephrased in the revised version of the manuscript, and we propose a new version of the abstract in the revised version. All mistakes have been corrected, and the paragraph was rephrased as follows:

Modified text:

**Abstract**. As part of the Oligotrophy to UlTra-oligotrophy PACific Experiment (OUTPACE), which took place in the Western Tropical South Pacific (WTSP) during the austral summer (March-April 2015) between the Melanesian Archipelago and the South Pacific gyre, we present the effects of intense dinitrogen ($N_2$) fixation on the dissolved inorganic nitrogen ($NO_3^-$) and particulate organic nitrogen (PON) pools. The stoichiometric nitrogen-to-phosphorus (N:P) ratio of the inorganic and particulate organic material, and the distribution of the geochemical tracer N* are described. N* has been previously used to trace changes in the proportion of nitrate ($NO_3^-$) relative to phosphate ($PO_4^{3-}$) associated with denitrification and/or diazotrophy in the open ocean. Our results show that the surface waters of the Melanesian Archipelago between 160° E and 170° W are characterized by significant $N_2$ fixation rates and relatively high PON concentrations (0.32-1.18 µmol kg$^{-1}$). In the surface and upper thermocline waters PON was present in excess (PON excess) relative to that expected from Redfield stoichiometry. In agreement with the high diazotrophic activity observed in the region, positive N* values (N excess) were observed in the Melanesian Archipelago, probably reflecting the combined effect of $PO_4^{3-}$ uptake by diazotrophic organisms and the remineralization of organic matter enriched in nitrogen (N). PON excess and N* were significantly correlated to $N_2$ fixation rates, probably reflecting the supply of N to the inorganic and organic N pools *via* $N_2$ fixation. The excess N observed in the thermocline waters of the WTSP is probably related to the sedimentation of N-rich matter produced at the surface by intense $N_2$ fixation activity. Following an analysis of the South Pacific nutrients database (Global Ocean Data Analysis Project version 2), we find a strong positive N* signature in the thermocline waters of the WTSP, the Coral/Tasman Seas, and the southern part of the subtropical gyre between 23 and 32° S. Together with a basin-scale analysis of water mass circulation, we hypothesize that the outstandingly high $N_2$ fixation rates observed at the surface influence the N status of thermocline waters across the whole South Pacific Ocean.

7. *Reviewer: Line 53: But nothing can be better than in situ measurements! Criticism of in situ measurements should be with caution and proper justification.*

Original text (lines 53): The spatial and temporal variability of the ocean nitrogen budget remains difficult to quantify from in situ measurements.

Response: We entirely agree with Reviewer#1. Our sentence is not a criticism of *in situ* measurements, but rather refers to the lack of *in situ* measurements in the WTSP. Only a few oceanographic cruises have been undertaken in this specific area (P21-1994, P21-2009 and OUTPACE cruise, Figure 3 of the original manuscript). Nevertheless, we recognize that the sentence may cause confusion, and it has been deleted in the revised manuscript.

8. ***Reviewer: Line 63: Let the equation come first before it is to be referred.***

Original text (lines 63): Despite the apparent simplicity of the proposed formula (Eq. 3), there is nonetheless a certain complexity, especially in the deconvolution of the different processes influencing the intensity of this tracer.

Response: Our thanks to Reviewer#2 for this comment. The equation has been added inline in the text.

Modified text: Michaels et al. (1996), Gruber and Sarmiento (1997), and Deutsch et al. (2001) proposed the $N^*$ ($N^* = NO_3^- - 16 \times PO_4^{3-} + 2.90$) as a tracer to determine changes in the stock of dissolved inorganic N as deviations from the Redfield ratio resulting from the combined effect of denitrification and/or of diazotrophy.

9. ***Reviewer: Line 83: area should be replaced with areal***

Original text (lines 83): with an average aera rates of 570 $\mu$mol $m^{-2}$ $d^{-1}$ (Bonnet et al., 2017)

Response: We agree with Reviewer#2. We have corrected it in the revised manuscript.

10. ***Reviewer: Line 114: Addition of HgCl2 might degrade Organic matter and one my overestimate nutrients. Convention is just to freeze the water for nutrient analysis. Why was HgCl2 added despite that samples were frozen?***

Response: The samples used for inorganic nutrient measurements were not frozen but poisoned with $HgCl_2$. This is a choice we have opted for in the laboratory for many years whenever samples must be transported over thousands of miles and for several months. Reviewer#1 is right in saying that current convention is just to freeze, but it has not always been the case, and alternative sampling preservation methods is still a subject of research (Wong et al., 2017). We have ourselves also undertaken a lot of tests in order to check our preservation procedure. Degradation of organic matter was never shown to significantly influence our measurements in our relatively low organic matter (oligotrophic) areas.

Please note that we used pre-filtered samples to measure nutrients and that we get relatively high quantification limits for nutrients (0.05 µM).

**11. Reviewer: Line 140: Provide lot no for the 15N2 gas as these have been subject to contamination recently.**

Response: We are aware that Dabundo et al. (2014) reported potential contamination of some commercial $^{15}N_2$ gas stocks with $^{15}N$- enriched $NH_4^+$, $NO_3^-$ and/or $NO_2^-$, and nitrous oxide ($N_2O$). Nonetheless, that does not significantly affect our results, as explained by Caffin et al. (2018) in another paper from the OUTPACE special issue. In their study, Dabundo et al. (2014) analyzed various brands of $^{15}N_2$ (Sigma, Cambridge Isotopes, Campro Scientific), and found that the Cambridge Isotopes brand (i.e., the one used in this study) contained low concentrations of $^{15}N$ contaminants. By using this contamination level, the potential overestimated $N_2$ fixation rates modeled would range from undetectable to 0.02 nmol N $L^{-1}$ $d^{-1}$. The $N_2$ fixation rates measured in our study were on average ~10 nmol N $L^{-1}d^{-1}$, suggesting that the stock contamination would be too low to affect the results reported here. To verify this, the Cambridge Isotopes batches that are routinely used by our team have been analyzed for potential contamination in Julie Granger and Richard Dabundo's lab. The results confirm that the contamination of the $^{15}N_2$ gas stock was low: 1.4 x $10^{-8}$ mol of $^{15}NO_3^-$ per mol of $^{15}N_2$, and $1.1x10^{-8}$ mol $NH_4^+$ per mol of $^{15}N_2$. The application of this contamination level to our samples using the model described in Dabundo et al. (2014) indicates that our rates could only be overestimated by 0.01 to 0.12 %. We thus confirm that the stock contamination issue did not affect the results reported here.

**12. Reviewer: Why 2.90 was not added in equation 2 but in 3. Is PONexcess zero without 2.90.**

Response: "PON excess" was calculated in order to show whether a N excess exists in the organic matter when compared to the classical composition of the organic matter, defined by the Redfield ratio (PON:POP = 16:1). There is no reason to add a 2.90 in this equation.

**13. Reasons should be provided why is it necessary to make it zero. I believe the equation was originally given by (Gruber and Sarmiento, 1997; Michaels et al., 1996) and not by Deutsch et al (2001) as authors seem to suggest (lines 150 and 191)**

Original text (lines 59-61) : Michaels et al. (1996) and Gruber and Sarmiento (1997) proposed the N* as a tracer to determine the proportion of changes in the fixed nitrogen stock (all nitrogenous forms except atmospheric $N_2$) resulting from the combined effect of denitrification and/or diazotrophy.

Response: We agree with Reviewer#1; the N* definition was indeed introduced first by Michaels et al. (1996). These authors first defined the N* parameter as the concentration of nitrogen in excess (or deficit) compared to that expected from remineralization of phosphate at Redfield stoichiometry. The first equation was:

$$N^* = NO_3^- - 16 \times PO_4^{3-} + 2.72$$

Thereafter, Gruber and Sarmiento (1997) have used the N* and modified it to investigate the distribution of nitrogen fixation and denitrification in the world ocean. The equation became:

$$N^* = (NO_3^- - 16 \times PO_4^{3-} + 2.90) \times 0.87$$

In our manuscript, we emphasized the fact that Michaels et al. (1996) and Gruber and Sarmiento (1997) were the first to define the N*. Nevertheless, in the South Pacific, the N* was described in detail by Deutsch et al. (2001). These authors used the equation:

$$N^* = NO_3^- - 16 \times PO_4^{3-} + 2.90$$

In our study, we chose the same definition, and we added the constant 2.90 only to compare the observations with the literature data (Deutsch et al., 2001).

**14. Reviewer: If inorganic and particulate values of N and P were considered for "excess" analysis here (equations 2 and 3) then why not dissolved organic values (DON and DOP) as well? That is considered to be an equally significant pool of N and P?**

Response: Reviewer#1 is right and it would have been a good idea to also show the N excess in the dissolved organic matter, but it was beyond the scope of this paper. The elemental ratio of dissolved organic matter does indeed deviate widely from the canonical Redfield ratio (i.e. Hopkinson and Vallino, 2015; Letscher and Moore, 2015). Neverthless, the turnover of the dissolved organic pool which is a gret deal longer than that of the particulate organic pool, making the N excess interpretation for the dissolved pool more difficult. Considering only the particulate matter (biomass) and inorganic matter (nutrient) is the simplest scheme. However, since this simplest scheme enables us to understand the main trends, it does not seem necessary to add more complexity and detail to the story. To clarify our manuscript, we have removed the part related to the dissolved organic matter.

**15. Reviewer: Provide the reproducibility (precision) and detection limit of all the measured parameters in the method section. Can PON and POP be measured in three decimal digits of mico-mol/L, which is almost like nmol/L in 3 decimal digits?**

Response: The precision (repeatability) and quantification limit (above detection limit) were established for all parameters and indicated table R4. Please note that a large volume was filtrated for particulate organic matter measurement, allowing, for example, a quantification limit of 1 nmol L$^{-1}$ for POP measurements.

Table R4. Accuracy (repeatability) and quantification limits of inorganic nutrients and particular organic N and P. CV for coefficient of variation.

|  | Repeatability (CV in %) | Quantification limit ($\mu$mol L$^{-1}$) |
| --- | --- | --- |
| $NO_3^-$ | 0.2 | 0.05 |
| $PO_4^{3-}$ | 0.7 | 0.05 |
| DON | 3.0 | 0.51 |
| DOP | 3.7 | 0.05 |
| PON | 5.3 | 0.02 |
| POP | 3.2 | 0.001 |

**16. Reviewer: Line 383: there is a typo in "surface"**

Response: Yes, we apologize for this mistake. In fact, the sentence has been deleted in the revised manuscript.

**17. Reviewer: Lines 424-425: sentence "the . . .. . .. . ...zooplankton" is not clear**

Original text (lines 424-425): The occurrence of diazotrophs and the associated nitrogen supply would allow the concomitant development of non-diazotrophic phytoplankton and zooplankton (Mulholland et al., 2007).

Response: Although we measured strong $N_2$ fixation rates, our PON:POP ratio was significantly lower than that observed by Karl et al., 1992 and Letelier and Karl (1998). In this part, we discuss the possible causes of the observed differences, and in particular, the fate of the nitrogen fixed by diazotrophs and the impact on the PON:POP ratio observed during the cruise. Sheridan et al. (2002), and then Mulholland (2007), showed that 85% of Trichodesmium colonies were inhabited by other organisms. Therefore, the occurrence of diazotrophs would allow the concomitant development of non-diazotrophic organisms (Caffin et al., 2018; Leblanc et al., 2018; Mulholland, 2007). The concomitant development

of non-diazotrophs organisms could explain the difference in the ratio of particulate organic matter observed between our observations and those of Karl et al. (1992) and Letelier and Karl (1998).

**General comment:** We propose a new version of the whole paragraph in the revised manuscript.

Modified text:

POM measurements showed a significant accumulation of particles in the surface waters and at the top of the UTW, followed by an attenuation in the central and lower part of the UTW (Fig. 5a, 5b). Despite the extremely low surface nutrient concentrations observed, a significant amount of POM was measured in the Melanesian Archipelago sampling stations, with PON and POP concentrations reaching 1.175 and 0.057 µmol $L^{-1}$, respectively. PON concentrations in oligotrophic regions are typically <0.6 µmol $L^{-1}$ (Hebel and Karl, 2001; Moutin and Raimbault, 2002; Martini et al., 2013). The observed values, and particularly at station LD B, were in the same order of magnitude as those measured in subpolar waters of northern latitudes (Sarmiento and Grüber, 2006), which are usually >1.2 µmol $L^{-1}$ in PON (Martini et al., 2013; Gruber 2008). The PON:POP ratios averaged over the entire water column were very close to the Redfield ratio, except for the most productive zones located in the Melanesian Archipelago, which had an excess of PON relative to that expected from Redfield stoichiometry, particularly close to the surface (Fig. 5d). This excess of PON was as high as 0.37 µmol $L^{-1}$, suggesting that the POM was enriched in N, despite the undetectable $NO_3^-$ concentrations.

Most PON:POP ratio data available in the literature corresponds to cruises in the North Pacific (Karl et al., 1992; 2001) and North Atlantic (Hansell et al., 2004), while the WTSP has so far remained an undersampled region. These studies showed that the chemical composition of the suspended particulate matter collected during *Trichodesmium* blooms was enriched in N relative to P with PON:POP ratios ranging between 40 (Letelier and Karl, 1998) and 125 (Karl et al., 1992). The stoichiometric PON:POP ratios observed during the OUTPACE cruise were much lower, with a maximum of 25.3 observed at station LD B. The differences observed in the PON:POP ratios may be explained by differences in nutrient availability, as previously suggested by Karl et al. (2002). Oligotrophic gyres in the northern hemisphere have lower concentrations of $PO_4^{3-}$ than those measured in the southern hemisphere (Moore et al., 2013), theoretically leading to higher PON:POP ratios in the North than in the South Pacific. Adding to in situ nutrient availability, new N derived from diazotrophic activity would allow the successive development of non-diazotrophic microorganisms (Caffin et al., 2018; Leblanc et al., 2018; Mulholland, 2007) with lower N:P content than diazotrophs, leading to a relatively lower PON:POP ratio.

*18. **Reviewer**: Line 428-431: "On an annual. . .. . .N\* signal". See (Mills and Arrigo, 2010; Singh et al.,2017) for an alternative hypothesis.*

*19. **Reviewer**: Lines 433-436: "Nevertheless. . .. . .. . .. . .atmospheric nitrogen input". See (Jickells et al., 2017; Singh et al., 2013) for more details.*

Original text (lines 428-431): On an annual time scale, the strong anomalies of surface N\* can be explained, on the one hand by the formation of diazotrophic organic matter, which together with the fixation of $N_2$ reduces the stock of $PO_4^{3-}$ and positively influences the N\* signal. On the other hand, part of the excess of PON which goes after remineralization towards the stock of fixed nitrogen may therefore also positively influence the N\* signal.

Response: We are aware that there are alternative hypotheses to possibly explain the N\* variation, and it is the subject of the following paragraph. We understand the necessity of discussing all the hypotheses that have been proposed to explain the important variations in N\* against the relatively low $N_2$ fixation rates measured. But it should be borne in mind that the $N_2$ fixation rates measured in the WTSP are exceptionally high (Bonnet et al., 2017), and that strong N\* variations are probably mainly explained by $N_2$ fixation. Furthermore, it is beyond the scope of this (already long) paper to deal with community structure, a subject that is discussed in several of the 26 other papers from the OUTPACE special issue [Interactions between planktonic organisms and biogeochemical cycles across trophic and $N_2$ fixation gradients in the western tropical South Pacific Ocean: a multidisciplinary approach (OUTPACE experiment)], available on the Biogeosciences website.

In the revised manuscript, alternative hypotheses have been added (Mills and Arrigo, 2010; Zamora et al., 2010; Singh et al., 2015;2017), as suggested by Reviewer#1 (see modified text below) and the sentence has been rephrased.

Modified text: However, N\* could be influenced by processes other than $N_2$ fixation, such as the non-Redfield nutrient assimilation by non-diazotrophic phytoplankton (Mills and Arrigo, 2010; Singh et al., 2017), and/or remineralization of non-diazotrophic organisms (Zamora et al., 2010; Singh et al., 2013). Furthermore, N\* could also be influenced by lateral advection of DOM with a high DON:DOP ratio (Letscher et al., 2013), and atmospheric deposition of low N:P ratio aerosols (Jickells et al., 2017). The transport is probably another process which affects N\* distribution in the ocean interior (Gruber and Sarmiento, 1997).

***20. Reviewer: Line 521: how can "0" rates be measured.***

Original text (lines 521): Dekaezemaker et al. (2013) measured $N_2$ fixation rates between 0 and 148 $\mu$mol N m$^{-2}$ d$^{-1}$,

Response: Yes, we agree with Reviewer#1, the sentence was confusing. N* could be influenced by processes other than $N_2$ fixation. This sentence has been removed in the revised manuscript.

***21. Reviewer: Line 555: "deficit of NO3 and PO4". Mathematically (from the N* concept used here), both nutrients cannot be deficit.***

Original text (lines 555): the waters located in the Melanesian Archipelago (160° E-170° W) have a deficit of $NO_3^-$ and $PO_4^{3-}$ in the productive layer (0-100 dbar),

Response: Thank you for these comments, it was indeed a mistake in the formulation of the sentence. This sentence has been deleted in the revised manuscript.

***22. Reviewer: Line 572: "occurred" should be replaced by "been undertaken"***

Original text (lines 572): Only few cruises have occurred in the South Pacific Ocean.

Response: This sentence has been deleted in the revised manuscript (see the modified text below answer to comment 23).

***23. Reviewer: Line 680: Letscher is misspelled (Letscher et al., 2013)***

Response: We apologize for the mistake. The reference has been corrected in the revised manuscript.

Modified text: Letscher, O., T., Hansell, D. A., Carlson, C. A., Lumpkin, R., and Knapp, A. N.: Dissolved organic nitrogen in the global surface ocean: distribution and fate Global Biogeochemical Cycles, 27, 141–153, 2013.

**General comment:** In the modified text, all the corrections referred to above have been taken into account, and we propose a new version of the Conclusion.

Modified text:

As part of the OUTPACE cruise in the western tropical South Pacific (160° E-160° W, 18°-20° S), we have analyzed the effect of the local $N_2$ fixation process on the dissolved and particulate organic N pools. We have revealed the presence of two distinct oceanic areas. The waters located in the Melanesian Archipelago (160° E-170° W) were characterized by significant $N_2$ fixation rates, high PON concentrations and a PON in excess relative to that expected from Redfield stoichiometry in the surface waters and in the top of the upper thermocline waters. In this area, we showed a significant relationship between $N_2$ fixation and PON in excess. Positive N* in the surface waters and in the upper and lower thermocline waters of the Melanesian Archipelago was observed in parallel with record-high rates of $N_2$ fixation. In contrast, the surface waters and the top of the upper thermocline waters of the South Pacific Gyre (170° W-160° W) are characterized by low $N_2$ fixation rates and low PON concentrations. In agreement with the low diazotrophic activity observed, negative or close to zero N* values were observed in the South Pacific Gyre.

The intense $N_2$ fixation observed in the WTSP significantly influences the dissolved inorganic (N excess) and particulate organic (PON excess) N pools of the surface and thermocline waters in the WTSP. The analyses of the most recent available nutrient data in the South Pacific Ocean enabled us to show a high positive N* in the thermocline waters of the whole South gyre. We therefore hypothesized that the local high N* observed in the Melanesian Archipelago is transported eastward *via* the circulation, and consequently that $N_2$ fixation in the WTSP may influence the nutrient distribution at the scale of the South Pacific basin.

---

## Author Comment (AC2) · 18 Oct 2018

**Responses to Review #2**

We thank Reviewer #2 for his relevant comments on our manuscript. Below is the original response from Reviewer#2 (in italic/bold), with our own responses interspersed within. The original text of the manuscript is in blue typeface and changes to the manuscript are shown in red.

1. *The presentation of the material is highly descriptive, in particular the results section is excessive in its low to details. Overall the manuscript is a data report and not a scientific paper.*

2. *Large parts of the text (Intro and Discussion in particular) are characterized by awkward and ambigious sentences. This partly seems to be a language problem, hence I strongly suggest to make use of appropriate editing from a person (co-author?) with appropriate knowledge of the subject and the English language. Please note that the review process is not a substitution of proper preparation of a manuscript. I provide an annotated manuscript, with 540+ comments and corrections. This is by far the largest number of annotations I ever had with a manuscript. Given this huge number of issues, there is no meaning to select major vs. minor points. In the current form the manuscript can only be rejected, with the possibility to resubmit to BG. I suggest that the authors collectively work on an improved new version of this manuscript.*

Response: We appreciate the consideration given by the reviewer to our data set and manuscript, and we understand his concerns regarding the descriptive character of the material and the excessive detail provided in the Results section. In the revised version of the manuscript, both the Material and Methods and Results sections have been rewritten in a clearer and more concise way. We again thank Reviewer#2 for his very thorough work in correcting our manuscript. All annotations have been considered in the revised version of the manuscript, apart from those that have been eliminated in shortened sections. Overall, the text of the paper has been shortened by a third of its initial length. Many sentences have been rewritten; lexical and grammatical mistakes have been corrected. The manuscript has been reviewed by all co-authors and finally corrected by a native English speaker. We only disagree with the reviewer concerning the implications of our study. The reviewer gives a merely descriptive dimension to our study, stating that the "manuscript is a data report and not a scientific paper". We consider that showing the N enrichment of the dissolved inorganic and particulate organic pools in the WTSP are major results, as is our hypothesis that the local high N* observed in the Melanesian Archipelago is transported eastward *via* the circulation, and consequently that $N_2$ fixation in the WTSP may influence the nutrient distribution at the scale of the South Pacific basin.

**General comment:** The title of our paper was "The effect of $N_2$ fixation on the thermocline waters of the South Pacific Ocean traced by excess nitrogen" (see page 1).

3.  *I strongly suggest to add a table with accronyms to the paper since you use them a lot. Also make sure to keep accronyms free of typos, e.g. p13 TKR instead of TKG (Fig.2b).*

Response: We agree with Reviewer#2, the number of acronyms has been reduced, and a table has been added in the Supplementary Material (Table S2).

4.  *Results: This section is unreadable. You repeat basically every single number you show in the tables 1-3. Reduce to 1/3 or less. Point the reader to patterns, differences. Otherwise the paper your paper has the nature of a data report and not a scientific paper. BG does not publish data reports.*

Response: The Results section has been rewritten more clearly, and the excessive detail has been deleted (see the revised Results section, and specific answers below comment 42).

5.  *Discussion This section is characterized by many awkward and ambigious sentences.*

Response: This section has been reworked, and awkward and ambiguous sentences have been corrected (see revised Discussion and specific answers below comment 76).

6.  *P13, 516ff. This is clearly a wrong interpretation of the Gruber 2016 paper!! Please read and understand a paper before citing it. Gruber (2016) is a commentary on the paper by Knapp et al. from the same PNAS issue. Gruber does not provide any new results, but just introduces the uncertainties related to both the Deutsch (inverse model) study and the Knapp observational study. The results of Knapp should be cited here, which are in line with Bonnet et al. 2017. Don't be misguided by the tone of the Gruber paper, which tries to rescue the â spatial proximity' hypothesis. But this is only for the reason that he himself introduced the general hypothesis in a paper in 2001.*

Response: Please see the revised Discussion section and specific answers below.

**Section : Abstract**

7.  *That is a bold simplification.*

Original text (lines 18): N* allows to trace changes in the proportion of fixed nitrogen due to diazotrophy and/or denitrification.

Response:  We agree with Reviewer#2, thank you for this comment. The sentence has been rephrased.

Modified text:  N* has been previously used to trace changes in the proportion of nitrate ($NO_3^-$) relative to phosphate ($PO_4^{3-}$) associated with denitrification and/or diazotrophy in the open ocean.

8.  *-relative to what?*
    *-PON and N:P can not be compared, units do not agree; rewrite the sentence*

Original text (lines 19-21): Our results showed that the Melanesian archipelago waters between 160° E and 170° W are characterized by a deficit of nitrate and phosphate in the productive layer, significant dinitrogen fixation rates and an excess of particulate organic nitrogen compared to the canonical ratio of Redfield.

Response: Thank you for these comments, it was indeed a mistake in the formulation of the sentence. These sentences have been rephrased.

Modified text: Our results show that the surface waters of the Melanesian Archipelago between 160° E and 170° W are characterized by significant $N_2$ fixation rates and relatively high PON concentrations (0.32-1.18 µmol kg$^{-1}$). In the surface and upper thermocline waters PON was present in excess (PON excess) relative to that expected from Redfield stoichiometry.

9.  *-Here and throughout the ms. N\* is already an anomaly (see equation 3) of NO3 compared to RR\*PO4 + C. Hence the word anomaly in the context of N\* is not neede, misleading.*
    *-excess to what? avoide phrases known to experts only, in an abstract*

Original text (lines 22): A positive N* anomaly was observed in the productive layer reflecting the combined effect of phosphate uptake by diazotrophic organisms and remineralization of excess particulate organic nitrogen.

Response: We agree with Reviewer#2 and have deleted 'anomaly' everywhere in the text; the sentence has been simplified.

Modified text: In agreement with the high diazotrophic activity observed in the region, positive N* values (N excess) were observed in the Melanesian Archipelago, probably reflecting the combined effect of $PO_4^{3-}$ uptake by diazotrophic organisms and the remineralization of organic matter enriched in nitrogen (N).

***10. unsuitable phrase for an abstract. overall the sentence reads strange.***

Original text (lines 28): In the thermocline waters, the N* tracer revealed its full complexity, with notably the cumulative effect of the remineralization of particulate organic nitrogen and the effects of the mixing of water masses.

Response: We agree, this sentence has been deleted.

***11. No global interpretation is provided, just a basin scale one!***

Original text (lines 29): At the global ocean scale, calculation of N* signal …

Response: the sentence has been rephrased in the revised version of the manuscript.

Modified text: Following an analysis of the South Pacific nutrients database (Global Ocean Data Analysis Project version 2), we find a strong positive N* signature in the thermocline waters of the WTSP, the Coral/Tasman Seas, and the southern part of the subtropical gyre between 23 and 32° S.

**General comment:** We propose a new version of the Abstract in the revised manuscript. All mistakes have been corrected, and the paragraph rephrased as follows:

Modified text:

**Abstract**. As part of the Oligotrophy to UlTra-oligotrophy PACific Experiment (OUTPACE), which took place in the Western Tropical South Pacific (WTSP) during the austral summer (March-April 2015) between the Melanesian Archipelago and the South Pacific gyre, we present the effects of intense dinitrogen ($N_2$) fixation on the dissolved inorganic nitrogen ($NO_3^-$) and particulate organic nitrogen (PON) pools. The stoichiometric nitrogen-to-phosphorus (N:P) ratio of the inorganic and particulate organic material, and the distribution of the geochemical tracer N* are described. N* has been previously used to trace changes in the proportion of nitrate ($NO_3^-$) relative to phosphate ($PO_4^{3-}$) associated with denitrification and/or diazotrophy in the open ocean. Our results show that the surface waters of the Melanesian Archipelago between 160° E and 170° W are characterized by significant $N_2$ fixation rates and relatively high PON concentrations (0.32-1.18 µmol $kg^{-1}$). In the surface and upper thermocline waters PON was present in excess (PON excess) relative to that expected from Redfield stoichiometry. In agreement with the high diazotrophic activity observed in the region, positive N* values (N excess) were observed in the Melanesian Archipelago, probably reflecting the combined effect of $PO_4^{3-}$ uptake by diazotrophic organisms and the remineralization of organic matter enriched in nitrogen (N). PON

excess and N* were significantly correlated to $N_2$ fixation rates, probably reflecting the supply of N to the inorganic and organic N pools *via* $N_2$ fixation. The excess N observed in the thermocline waters of the WTSP is probably related to the sedimentation of N-rich matter produced at the surface by intense $N_2$ fixation activity. Following an analysis of the South Pacific nutrients database (Global Ocean Data Analysis Project version 2), we find a strong positive N* signature in the thermocline waters of the WTSP, the Coral/Tasman Seas, and the southern part of the subtropical gyre between 23 and 32° S. Together with a basin-scale analysis of water mass circulation, we hypothesize that the outstandingly high $N_2$ fixation rates observed at the surface influence the N status of thermocline waters across the whole South Pacific Ocean.

**Section 1: Introduction**

> **12. I don't see the relationship you refer to here. Where does Moore refer to the efficiency of the biological carbon sequestration in relation to deviations of N:P from RR? Searching pump in Moore, I found the following statement, which does not relate to your citation! 'Physical processes also transport biologically unutilized (so-called 'preformed') nutrients into the ocean interior, leading to a decreased efficiency of the biological pump' (p707)**

Original text (lines 44): Deviations from this ratio have been used by marine geochemists to provide insights into nutrient limitation of primary production, efficiency of the biological carbon sequestration and dinitrogen ($N_2$) fixation (Moore et al., 2013).

Response: We agree with Reviewer#2. The reference of Moore et al. (2013) is related to the limitation of primary production by nutrients. In the revised version of the paper, this sentence has been rephrased, and two references have been added to refer to carbon sequestration, denitrification and dinitrogen fixation.

Modified text: Deviations from this ratio have been used by marine geochemists to provide insights into nutrient limitation of primary production (Moore et al., 2013), efficiency of biological carbon sequestration (Sigman and Boyle, 2000) and to investigate the distribution of denitrification and $N_2$ fixation in the world oceans (Gruber and Sarmiento, 1997).

> **13. -Awk! what do you mean with 'mineral matter'? inorganic particles? hardly have N; or nutrients? if so, say please (but does not make sence with the rest of the sentence).**
> **-Only nitrogen sources and sinks? not P sms, as well?**

Original text (lines 45): The canonical RR of the organic and mineral matter is strongly variable at a regional scale (Moore et al., 2013) and mostly depends on nitrogen sources and sinks.

Response: We agree with Reviewer#2, the sentence was not clear. In the revised manuscript, it is specified that mineral matter relates to dissolved inorganic nutrients. Furthermore, we agree that the deviation from the Redfield N:P ratio could indeed be related to variations of both N and P concentrations. Therefore, the sentence has been rewritten in the revised version of the manuscript.

Modified text: The N:P ratio of both particulate organic and dissolved inorganic matter is strongly variable at regional scale (Martiny et al., 2013; Moore et al., 2013) and depends, among others factors, on N and P sources and sinks.

14. *Awk: what is the difference between turbulent mixing and diffusion? due you refer to molekular diffusion here??*
   *Awk; you mean reduce fixed nitrogen -NH2, but it sounds as if the consequence of N2-fix is an increase in extracelular NH4 concentrations - to the benefit of all non-diazotrophs*

Original text (lines 46-49): The supply of nitrogen to the productive layer comes from turbulent mixing of water masses and diffusion (Oschlies and Garcon, 1998; Capone et al., 2005), atmospheric and river inputs (Hansell et al., 2004; Jickells et al., 2017) and $N_2$ fixation, the conversion of atmospheric $N_2$ into ammonia by diazotrophic organisms (Karl et al., 1992).

Response: We refer to eddy diffusion and not molecular diffusion, and have finally simplified the sentence.

Modified text: The supply of N to the productive layer is due to the mixing of water masses and eddy diffusion (Oschlies and Garcon, 1998; Capone et al., 2005), atmospheric and riverine inputs (Hansell et al., 2004; Jickells et al., 2017) and biological $N_2$ fixation (Karl et al., 1992).

15. *sounds strange in the context of the text; on what time scale, days, weeks? delete word*

Original text (lines 49-52): These sources are continuously balanced by N loss processes, mainly driven by denitrification which converts nitrate ($NO_3^-$) into $N_2$ in Oxygen Minimum Zones (OMZ) (Codispoti and Richards, 1976).

Modified text: At global ocean scale, the N budget is controlled by the balance between the main sources ($N_2$ fixation), and sinks (denitrification and anammox) (Gruber and Sarmiento, 1997; Gruber, 2016), both mediated by microbial processes.

16. *Awk: what is he spatial variability of the ocean N budget?? of what? 'in situ rate measurements'?*

Original text (lines 53): The spatial and temporal variability of the ocean nitrogen budget remains difficult to quantify from in situ measurements.

Response: We agree with Reviewer#2, our sentence was not clear and has been deleted in the revised version of the manuscript.

17. *Awk: sentence is strange. you mean 'rate measurements of N2-fixation and denitrifitation (anamox'). awk: the estimates are dedicated?? 'carried out in the context of process studies with limited spatial and temporal coverage'*

Original text (lines 54-56): Alternatively, the use of geochemical tracers based on nutrients ratios allows to visualize and quantify the excess of nitrogen over large time and space scales contrary to the biological estimates, which are often dedicated to process studies at the local scale and on short time scales.

Response: We agree with Reviewer#2, the sentence was confusing. In the revised manuscript, the sentence has been simplified and rewritten.

Modified text: Geochemical tracers have long been used to gain insights on N cycling at large temporal and spatial scales (Sarmiento and Gruber, 2006).

18. *-An idealized tracer (NO=…). Give the definition in line in the text.*
   *-awk formulation; simplify the sentence by using the proper definition of the tracer that Broecker and Peng (1982) used! (equation inline in the text)*

Original text (lines 56-58): Broecker and Peng (1982) were the first to use a tracer to quantify the $NO_3^-$ deficit in the Indian Ocean and in the Bering Sea by multiplying the observed phosphorus concentrations by the stoichiometric ratio of 15 and by subtracting the result obtained by the observed $NO_3^-$ concentrations (Gruber and Sarmiento, 1997).

Response: In the publication of Broecker and Peng (1982), the NO tracer was not formulated. The NO tracer was provided in the publication of Broecker (1974). We agree with Reviewer#2 that the sentence would be confusing. Therefore, the sentence has been rewritten and the equation was added.

Modified text: Broecker (1974), and then Naqvi and Sen Gupta (1985), used the NO tracer (NO = 9 $NO_3^-$ + $O_2$) to estimate the $NO_3^-$ deficiency observed in the Arabian Sea.

19. *-give the equation inline here in the text*
   *-not really true; fixed N is all N which is not N2, but N\* uses NO3 (or technically often NO3+NO2); GS97 did not suggest to include NH4, urea, DON in general or PON!*
   *-Technically you can not refer to Eq 3 before Eq 1, also you should give an equation close to its reference; hence my suggestion to give the N\* equation inline in the text.*

Original text (lines 60-63): Michaels et al. (1996) and Gruber and Sarmiento (1997) proposed the N* () as a tracer to determine the proportion of changes in the fixed nitrogen stock (all nitrogenous forms except atmospheric $N_2$) resulting from the combined effect of denitrification and/or of diazotrophy.

Response: Our thanks to Reviewer#2 for this comment. The equation has been added inline in the text. The term "fixed nitrogen" has been replaced by "stock of inorganic nitrogen" in the revised version of the manuscript.

Modified text: Michaels et al. (1996), Gruber and Sarmiento (1997), and Deutsch et al. (2001) proposed the N* (N*= $NO_3^-$ -16 × $PO_4^{3-}$ + 2.90) as a tracer to determine changes in the stock of dissolved inorganic N as deviations from the Redfield ratio resulting from the combined effect of denitrification and/or of diazotrophy.

20. *Together with the follow up sentences, it is awk that you specifically refer to this process as being biogeochemically. Is it more bgc compared to remin of diazotrophic matter?*

Original text (lines 64-66): Factors may be biogeochemical, such as phosphate ($PO_4^{3-}$) uptake by diazotrophic organisms in the productive layer that will induce positive N* variations (Fig. 1). The remineralization of organic matter from diazotrophic organisms, being naturally enriched in nitrogen (Redfield et al., 1963; Anderson and Sarmiento, 1995; Hansell et al., 2004), will supply nitrogen to the system and will positively influence the N* (Fig. 1).

Response: We understand that the sentence was confusing, and we agree that both enounced processes are biogeochemical. In the revised manuscript, the paragraph has been rewritten.

Modified text: Positive N* variations may be due to $PO_4^{3-}$ uptake by diazotrophic organisms in the productive upper layer, or to diazotroph biomass (enriched in N) remineralization (Redfield et al., 1963; Anderson and Sarmiento, 1995; Hansell et al., 2004).

**21. add: while organic P is release as PO4. Add anamox**

Original text (lines 67-70): On the opposite, denitrification will cause negative variations of N* (Fig. 1), firstly because of the direct use of $NO_3^-$ as an oxidant by heterotrophic bacteria, and secondly because of the conversion of organic nitrogen to $N_2$. As a result, net positive changes in N* will trace a source of nitrogen relative to phosphorus, and negative net changes will indicate nitrogen loss in the system through the denitrification process.

Modified text: In contrast, denitrification (or annamox) will cause negative variations of N* because of the direct use of $NO_3^-$ (or $NH_4^+$ and $NO_3^-$) as an oxidant by heterotrophic bacteria ultimately lost as $N_2$. As a result, net positive changes in N* will trace a source of N relative to P, and negative net changes will indicate N loss in the system.

**22. Mode waters ! here and elsewhere**

Original text (lines 77): sub-Antarctic Modal Waters (SAMW)

Response: We agree with Reviewer#2. We have corrected this word throughout the manuscript (lines 77; 238).

**23. Areal**

Original text (lines 82): with an average aera rates of 570 µmol m$^{-2}$ d$^{-1}$ (Bonnet et al., 2017)

Response: We agree with Reviewer#2. the mistake has been corrected in revised manuscript.

**General comment:** We propose a new version of the introduction in the revised manuscript. All mistakes were corrected, the figure 1 was removed, the main goal of our paper was rewritten and the introduction was rephrased as follows:

Modified text:

**1 Introduction**

It is commonly accepted that net biological activity is supported by the supply of new N into the surface productive layer (Dugdale and Goering, 1967; Capone et al., 2005). At the global ocean scale, Redfield et al. (1963) reported similar N:P ratios between surface ocean particulate organic matter (POM) and in deep water inorganic nutrients, and since then the canonical Redfield ratio of 16:1 has become a fundamental tenet in marine biogeochemistry (Deutsch and Weber, 2012). Deviations from this ratio have been used by marine geochemists to provide insights into nutrient limitation of primary production (Moore et al., 2013), efficiency of biological carbon sequestration (Sigman and Boyle, 2000) and to investigate the distribution of denitrification and $N_2$ fixation in the world oceans (Gruber and Sarmiento, 1997). The N:P ratio of both particulate organic and dissolved inorganic matter is strongly variable at regional scale (Martiny et al., 2013; Moore et al., 2013) and depends, among others factors, on N and P sources and sinks. The supply of N to the productive layer is due to the mixing of water masses and eddy diffusion (Oschlies and Garcon, 1998; Capone et al., 2005), atmospheric and riverine inputs (Hansell et al., 2004; Jickells et al., 2017) and biological $N_2$ fixation (Karl et al., 1992). At global ocean scale, the N budget is controlled by the balance between the main sources ($N_2$ fixation), and sinks (denitrification and anammox) (Gruber and Sarmiento, 1997; Gruber, 2016), both mediated by microbial processes. Geochemical tracers have long been used to gain insights on N cycling at large temporal and spatial scales (Sarmiento and Gruber, 2006). Broecker (1974), and then Naqvi and Sen Gupta (1985), used the NO tracer ($NO = 9\ NO_3^- + O_2$) to estimate the $NO_3^-$ deficiency observed in the Arabian Sea. Michaels et al. (1996), Gruber and Sarmiento (1997), and Deutsch et al. (2001) proposed the N* ($N* = NO_3^- - 16 \times PO_4^{3-} + 2.90$) as a tracer to determine changes in the stock of dissolved inorganic N as deviations from the Redfield ratio resulting from the combined effect of denitrification and/or of diazotrophy. Despite the apparent simplicity of the N* formulation, there is nonetheless a certain complexity, especially in the deconvolution of the different biogeochemical processes influencing the magnitude of this tracer. Positive N* variations may be due to $PO_4^{3-}$ uptake by diazotrophic organisms in the productive upper layer or to diazotroph biomass (enriched in N) remineralization (Redfield et al., 1963; Anderson and Sarmiento, 1995; Hansell et al., 2004). In contrast, denitrification will cause negative variations of N* because of the direct use of $NO_3^-$ as an oxidant by heterotrophic bacteria ultimately lost as $N_2$. As a result, net positive changes in N* will trace a source of N relative to P, and negative net changes will indicate N loss in the system.

Using the data compiled from the World Ocean Circulation Experiment cruises, Deutsch et al. (2001) showed that the N* in the South Pacific presented maximum values at the surface decreasing with depth. The highest negative N* ($<10\ \mu mol\ kg^{-1}$) for the entire South Pacific was found in the East Tropical South

Pacific (ETSP) extending horizontally westwards from the South American coast at 15° S, which was later confirmed by Rafter et al. (2012) and Yoshikawa et al. (2015). The OMZ of the ETSP is one of the three regions of the world oceans where strong denitrification occurs (Codispoti and Richards, 1976). Conversely, the highest positive N* values were found in surface waters of the western subtropical waters near Australia (Deutsch et al., 2001), which is consistent with the high abundance of diazotrophic organisms, including cyanobacteria *Trichodesmium* sp., found in the warm oligotrophic waters of the WTSP (Moutin et al., 2005; Bonnet et al., 2009; Messer et al., 2015). This region has recently been identified as a global hotspot of $N_2$ fixation, with average areal rates of 570 µmol $m^{-2}$ $d^{-1}$ (Bonnet et al., 2017). However, to the best of our knowledge, the effect of intense $N_2$ fixation on the dissolved inorganic and particulate organic N pools in the WTSP, and on the nutrient distribution in the thermocline waters at the scale of the South Pacific Ocean, have never been investigated. Whether the intense $N_2$ fixation in the WTSP can explain the excess N detected in the thermocline waters of the whole South Pacific Ocean remains an open question.

**Section 2: Materials and methods**

**Section 2.1: Station location and sampling**

*24. I suggest to use a simple numbering instead of SD1 to LDB*

Original text (lines 99): Stations SD 1 to LD B

Response: The present paper is in the framework of the special issue "Interactions between planktonic organisms and biogeochemical cycles across trophic and $N_2$ fixation gradients in the western tropical South Pacific Ocean: a multidisciplinary approach (OUTPACE experiment)". The distinction between short duration (SD) and long duration (LD and the denomination are defined by the introductory paper (Moutin et al., 2017), and used in the 26 other papers participating in the SI. We therefore propose to use the same denomination for the different sampled stations.

**General comment:** We propose a new version of section 2.1 in the revised version of the manuscript. All mistakes have been corrected, the oxygen data have been deleted and section 2.1 has been rephrased as follows:

Modified text:
**2.1 Station location and sampling**

The OUTPACE cruise took place from 18 February to 3 April 2015, along a west to east transect from Noumea (New Caledonia) to Papeete (French Polynesia), (Fig. 1a). A total of 18 stations were sampled referred to as SD or LD, depending on the sampling time (SD: short duration; LD: long duration). Station LD B was located at the geographical boundary between the Melanesian Archipelago and the waters of the gyre (De Verneil et al., 2017). Stations SD 1 to LD B (Fig. 1a) will in the remainder of this study be referred to as Melanesian Archipelago waters, and will be color coded red, and stations SD 13 to SD 15 will be referred to as the waters of the gyre, and will be color coded blue (Fig. 1b). All samples were collected with a CTD Sea-Bird 911+, combined with a carousel equipped with a rosette holding 24 12 L Niskin bottles. Details on the sampling strategy are described in Moutin et al. (2017).

**Section 2.2: Inorganic and organic pools**

**25. rearange order of words**

Original text (lines 121): The samples used for the analysis of nitrogen and particulate organic phosphate (respectively PON and POP) were collected

Modified text: The samples used for the analysis of particulate organic nitrogen and phosphate (PON and POP respectively) were collected

**26. abbreviation not given; proper term is 'detection limit'; I suggest to reduce the number of abbreviations in this paper**

Original text (lines 125): The QL of the POP was 0.001 μmol L$^{-1}$ and of PON 0.02 μmol L$^{-1}$.

Response: As suggested by Reviewer#2, we have reduced the number of abbreviations; for example, we have removed the term QL here and elsewhere in the text. The quantification limit is defined as the least measurable concentration which is higher than the detection limit, corresponding to the least detectable (idem) concentration (IUPAC). Only the concentrations ($NO_3^-$, $PO_4^{3-}$, POP, PON) above the quantification limits (Table R1 below) have been taken into account in the calculation.

Table R1. Accuracy (repeatability) and quantification limits of inorganic nutrients and particular organic N and P. CV for coefficient of variation.

|  | Repeatability (CV in %) | Quantification limit ($\mu$mol L$^{-1}$) |
|---|---|---|
| NO$_3^-$ | 0.2 | 0.05 |
| PO$_4^{3-}$ | 0.7 | 0.05 |
| PON | 5.3 | 0.02 |
| POP | 3.2 | 0.001 |

**27. 'which converts DON (and reduced inorganic N) and DOP to NO3 and PO4', right?**

Original text (lines 128-129): Filtered samples were collected in teflon vials adjusted to 20 ml for wet oxidation. DON and DOP concentrations were obtained by the wet oxidation procedure according to Pujo-Pay and Raimbault (1994).

Response: Yes, that is right. The wet oxidation method allows conversion of the dissolved organic fraction to inorganic nutrients. The organic fraction was then calculated by subtracting the inorganic nutrient concentration measured before the wet oxidation from the total dissolved nutrient concentration obtained after the wet oxidation.

**28. use consistent terminology, see last paragraph**

Original text (lines 130-132): After cooling, the concentration of Total Dissolved Nitrogen (TDN) and Total Dissolved Phosphorus (TDP) was determined using continuous flow analysis (AAIII HR).

Response:  We decided, in order to shorten the text and because it was not discussed in the paper, to delete the section related to the dissolved organic matter in the revised version of the manuscript.

**29. DOP ?**

Original text (lines 133): The QL of POD was fixed

Response: We apologize for the mistake. As explained above, the section related to the dissolved organic matter has been deleted.

**General comment:** We propose a new version of the section 2.2 in the revised manuscript. All mistakes were corrected, the DON and DOP data was deleted of the revised manuscript and the section 2.2 was rephrased as follows:

Modified text:

**2.2 Inorganic and organic pools**

Inorganic nutrient samples were collected from each depth, sampled between 0 and 2000 dbar, in polypropylene scintillation vials closed with a HDPE caps and fixed with 50 µL of $HgCl_2$ (20 mg $L^{-1}$). The samples were stored at 4 °C until analysis in the laboratory. Concentrations of $PO_4^{3-}$ and $NO_3^-$ + nitrites ($NO_2^-$) were determined automatically using the continuous flow method (Aminot and Kérouel, 2007) by means of a SEAL Analitical AA3HR autoanalyzer. $NO_2^-$ concentrations were measured following the Griess reaction, adapted for seawater by Benschneider and Robinson (1952). $NO_3^-$ concentrations were measured following the protocol described by Wood et al. (1976). $PO_4^{3-}$ concentrations were measured with the method adapted for seawater by Murphy and Riley (1962). The quantification limit for all nutrients was 0.05 µmol $L^{-1}$. No nutrient data were available at SD 8. The samples used for the analysis of particulate organic nitrogen and phosphate (PON and POP respectively) were collected for each depth in polycarbonate bottles. A volume of 1.2 L was filtered through a precombusted (24h, 450 °C) glass fiber filter (Whatman GF/F, 25 mm). PON and POP concentrations were quantified using the wet oxidation method, based on persulfate digestion at 120 °C (Pujo-Pay and Raimbault, 1994). $NO_3^-$ and $PO_4^{3-}$ formed by oxidation were analyzed using the autoanalyzer as described above. The quantification limits of POP and PON were 0.001 µmol $L^{-1}$ and 0.02 µmol $L^{-1}$, respectively.

**Section 2.3: $N_2$ fixation rates**

> *30. -What about the bubble trouble issue??*
> *-What about the inorganic N contamination problem. You can not publish 15N-tracer addition experimental data without refering to these issues!*

Response: We are aware that Dabundo et al. (2014) reported potential contamination of some commercial $^{15}N_2$ gas stocks with $^{15}N$- enriched $NH_4^+$, $NO_3^-$ and/or $NO_2^-$, and nitrous oxide ($N_2O$). Nonetheless that does not affect significantly our results, as explained by Caffin et al. (2018) in another paper from the OUTPACE special issue. In their study, Dabundo et al. (2014) analyzed various brands of $^{15}N_2$ (Sigma, Cambridge Isotopes, Campro Scientific) and found that the Cambridge Isotopes brand (i.e., the one used in this study) contained low concentrations of $^{15}N$ contaminants. By using this contamination level, the potential overestimated $N_2$ fixation rates modeled would range from undetectable to 0.02 nmol N $L^{-1}$ $d^{-1}$.

The $N_2$ fixation rates measured in our study were on average ~10 nmol N $L^{-1}d^{-1}$, suggesting that the stock contamination would be too low to affect the results reported here. To verify this, the Cambridge Isotopes batches that are routinely used by our team have been analyzed for potential contamination in Julie Granger and Richard Dabundo's lab. The results confirm that the contamination of the $^{15}N_2$ gas stock was low: 1.4 x $10^{-8}$ mol of $^{15}NO_3^-$ per mol of $^{15}N_2$, and 1.1x$10^{-8}$ mol $NH_4^+$ per mol of $^{15}N_2$. The application of this contamination level to our samples using the model described in Dabundo et al. (2014) indicates that our rates could only be overestimated by 0.01 to 0.12 %. We thus confirm that the stock contamination issue did not affect the results reported here.

> **31. This is the original paper that presents the N2 data you show in your study, or are your data independent (addition to) the data presented in Bonnet? Please clarify.**

Response: It was indeed, the same database. This point has been clarified in the revised manuscript.

**Section 2.4: Calculation of PON excess**
**Section 2.5: Calculation of N\***

> **32. simplify and replace RnitrN/P by the number 16, as you did in equ 2**

Original text (lines 152-153): The concentrations of $NO_3^-$ and $PO_4^{3-}$ are linked together by the stoichiometric ratio $r^{N/P}_{nitr} = 16$ determined by Redfield et al. (1963) and Takahashi et al. (1985).

Response: We agree with Reviewer#2. The term RnitrN/P has been replaced by 16 in the revised manuscript (see the modified text below answer to comment 37).

> **33. -the biological pump is defined in terms of a carbon pump; I would avoid this word here; instead: see text changes**
> **-you mean the 'global mean' or 'global integral' value, right**
> **-awk phrase: clarify please**

Original text (lines 153-155): Equation (3) assumes that the effect of the biological pump on the absolute value of N\* is zero, that is the difference between $NO_3^-$ and remineralized $PO_4^{3-}$ is constant (Gruber and Sarmiento, 1997).

Response: Deutsch et al. (2001) define the N* as a tracer "which would be unchanged by the normal biological pump". Nerveless, in order to clarify our text, we have rephrased our definition of N* in the revised manuscript

No, I do not mean the "global mean" or the "global integral' value, but just the definition of N*. The exact definition of Deutsch et al. (2001) has been added.

The sentence has been rephrased in the revised version of the manuscript (see the modified text below answer to comment 37).

**34. -you mean 'global'?**

**-really only base on the GEOSECS data? or the WOCE+GEOSECS data; that are at least the data Deutsch uses; double check please**

Original text (lines 155-156):  The constant 2.90 makes it possible to force to zero the overall average value of the N* calculated from GEOSECS data (Geochemical Ocean Sections Study, 1972-1978) (Deutsch et al., 2001).

Response: Yes, in this case we have added "global mean".

"WOCE data" was added in the revised manuscript (see the modified text below answer to comment 37).

**35. Could you please express this simple thing in simpler words!**

Original text (lines 104-107):  In surface waters, above isopycnal 23.5 ($\sigma 23.5$), the $NO_3^-$ concentrations were not measurable and did not allow direct calculation of the N* anomaly. Considering only $PO_4^{3-}$ data above QL in surface waters, an N* estimate ($N^*_{surf}$) was carried out considering that below QL, $NO_3^-$ concentrations were between 0 and 0.05 µmol $L^{-1}$:

$$-16 \times PO_4^{3-} + 2.9 \leq N^*_{surf} \leq 0.05 - 16 \times PO_4^{3-} + 2.9 \qquad (4)$$

Response: We understand the concerns of the Reviewer#2. The sentence was confusing. It has been rewritten in the revised version of the manuscript (see the modified text below answer to comment 37).

**36. You mention Fig. 4 before Fig. 3; this is technically not allowed. Improve please.**

Original text (lines 169): To visualize the variations of N* in the thermocline waters of the OUTPACE section, the N* anomaly has been represented on isopycnal surfaces 24.70 and 26.30, which respectively correspond to the cores of the upper and lower thermoclines observed on the Θ-S diagram (Fig. 4a).

Response: The mistake has been corrected in the revised manuscript. Figure 3 has been moved to a panel (c) of Fig. 12 (of the original manuscript) (see the modified text below answer to comment 37).

*37. Samples*

Original text (lines 171): The mean and standard deviation of interpolated N* values was calculated for all stations on σ24.70 ±0.05 and σ26.30 ±0.05.

Response: The mistake has been corrected in the revised manuscript (see the modified text below).

**General comment:** We propose a new version of section 2.5 in the revised manuscript. All mistakes have been corrected and section 2.5 has been rephrased as follows:

Modified text:
The N* was calculated from the definition of Deutsch et al (2001).

$$N^* = NO_3^- - 16 \times PO_4^{3-} + 2.90 \ (\mu mol \ L^{-1}) \tag{2}$$

$PO_4^{3-}$ is the concentration of "soluble reactive phosphate" ($\mu mol \ L^{-1}$). The concentrations of $NO_3^-$ and $PO_4^{3-}$ are linked together by the stoichiometric ratio (N:P = 16) determined by Redfield et al. (1963) and Takahashi et al. (1985). Equation (2) assumes that the effect of production and remineralization of organic matter on the definition of the tracer of N* is zero (Deutsch et al., 2001). The constant 2.90 makes it possible to force to zero the global mean value of the N* calculated from GEOSECS (Geochemical Ocean Sections Study, 1972-1978) and WOCE data (Deutsch et al., 2001). This constant was included in this study in order to compare the observations with the literature data.
In the surface waters and in the top of the UTW above σ23.5, the $NO_3^-$ concentrations were below the quantification limit, preventing a direct N* calculation. In this layer, $N^*_{surf}$ was estimated only for sampling depths where $PO_4^{3-}$ concentrations exhibited a value higher than the quantification limit and considering two extreme $NO_3^-$ concentrations, 0 and 0.05 $\mu mol \ L^{-1}$ (Eq. 3).

$$-16 \times PO_4^{3-} + 2.9 \leq N^*_{surf} \leq 0.05 - 16 \times PO_4^{3-} + 2.9 \tag{3}$$

To visualize the variations of N* in the thermocline waters of the OUTPACE section, the N* will be represented on the isopycnal surfaces characterized by σ = 24.70 and σ = 26.30, which respectively correspond to the cores of the upper and lower thermoclines observed on the Θ-S diagram (Fig. 2a).

The density, $NO_3^-$, $PO_4^{3-}$, and N* data were interpolated linearly, each 1dbar, over the sampling depth. The mean and standard deviation of the interpolated N* values were calculated for all stations samples between σ24.65 and σ24.75 and between σ26.25 and σ26.35.

**Section 2.6: Variability of the $PO_4^{3-}$ pool**

*38. Reading this section, I wondered what it will be used for! This needs to be introduced to the reader.*
*add overbars to visually express that these are means, also to the PO4(GY) term*
*use overbars to visually express that this is a mean*

Original text (lines 173-181): The difference in average concentration observed in $PO_4^{3-}$ between the MA and the GY waters ($\Delta PO_4^{3-}{}_{observed}$) was calculated from means (±sd) observed in surface waters ($< σ23.5$) and is equal to:

$$\Delta PO_4^{3-}{}_{observed} = PO_4^{3-}{}_{observed\ (MA)} - PO_4^{3-}{}_{observed\ (GY)} \tag{5}$$

From the estimate of N*$_{surf}$ in surface waters ($< σ23.5$), the difference between the mean concentrations observed in N* ($\Delta$N*) in the surface MA and surface GY waters is equal to:

$$\Delta N^* = N^*_{mean\ (MA)} - N^*_{mean\ (GY)} \tag{6}$$

Response: Following the comment by Reviewer#2, and to avoid any confusion, this unnecessary section has been deleted from the revised version of the manuscript.

**Section 2.7: GLODAPv2 database**

*39. awk phrase for a scientific paper*

Original text (lines 187): …as well as 168 additional cruises were grouped together and have undergone one and only high-quality control, based on two steps (primary and secondary QC) applied to each data.

Response: Thank you for your comment. The sentence has been rephrased as follows:

Modified text: … as well as 168 additional cruises, were grouped together and have undergone, a high-quality control based on two steps (primary and secondary quality control) applied to each data.

***40. odd phrase***

Original text (lines 188-189): The GLODAPv2 database is available at
http://cdiac.ornl.gov/oceans/GLODAPv2/ and the details of the overall production strategy are
explained in Olsen et al. (2016).

Response: We agree, the sentence has been rephrased.

Modified text: The GLODAPv2 database was provided by Olsen et al. (2016), and is available at
http://cdiac.ornl.gov/oceans/GLODAPv2.

***41. the essential part is missing: the density level to which you interpolated***

Original text (lines 193-194): Similar to OUTPACE data processing, the density, $NO_3^-$, $PO_4^{3-}$, and N*
anomaly data were interpolated linearly between σ24.65 and σ24.75 and between σ26.25 and σ26.35.

Response: Data were interpolated over the whole sampling depth each 1 dbar. The information has been
added in the revised manuscript.

Modified text: As for the OUTPACE data processing, the density, $NO_3^-$, $PO_4^{3-}$, and N* data were
interpolated linearly, each 1dbar, over the sampling depth. To get insight on the spatial variability of N*
in the thermocline waters, N* data were averaged between σ24.65 and σ24.75 and between σ26.25 and
σ26.35.

**General comment:** We propose a new version of section 2.7 in the revised manuscript. All mistakes have
been corrected and section 2.7 has been rephrased as follows:

Modified text:
The Global Ocean Data Analysis Project version 2 (GLODAPv2) is an international effort to consolidate
all data from oceanographic bottle samples collected during numerous oceanographic cruises (Olsen et
al., 2016). Previous databases, including WOCE/JGOFS combined in GLODAPv1 in 2004 (Sabine et al.,
2005; Key et al., 2004), CARINA (CARbon IN the Atlantic) in 2009/2010 (Key et al., 2010; Tanhua et
al., 2009), and PACIFICA (PACIFIc ocean Interior CArbon) in 2013 (Suzuki et al., 2013), as well as 168
additional cruises, were grouped together and have undergone, a high-quality control based on two steps
(primary and secondary quality control) applied to each data. The GLODAPv2 database was provided by
Olsen et al. (2016) and is available at http://cdiac.ornl.gov/oceans/GLODAPv2. It includes samples of

core variables such as salinity, oxygen, macronutrients, and seawater $CO_2$ chemistry from 724 oceanic cruises. In this study, we focused on two variables: $NO_3^-$ and $PO_4^{3-}$ and we calculated N* according to the formula of Deutsch et al. (2001) (Eq. 2) for all available data and cruises in the South Pacific (Table S1). As for the OUTPACE data processing, the density, $NO_3^-$, $PO_4^{3-}$, and N* data were interpolated linearly, each 1dbar, over the sampling depth. To obtain insight on the spatial variability of N* in the thermocline waters, N* data were averaged between σ24.65 and σ24.75 and between σ26.25 and σ26.35.

**Section 3: Results**

42. ***The presentation of the material is highly descriptive, in particular the results section is excessive in its low to details. Overall the manuscript is a data report and not a scientific paper. This section is unreadable. You repeat basically every single number you show in the tables 1-3. Reduce to 1/3 or less. Point the reader to patterns, differences. Otherwise the paper your paper has the nature of a data report and not a scientific paper. BG does not publish data reports.***

Response: We agree with Reviewer#2 regarding the excessive detail provided in the manuscript. In the revised manuscript, the Results section has been rewritten more clearly, and the excessive detail deleted. Moreover, the aims of the study have also been clarified and new scientific knowledge highlighted.

43. ***misleading, your survey is not global, but basin scale***

Original text (lines 196-197): In order to replace the OUTPACE cruise within the context of the general circulation in the South Pacific we start by a global survey (Fig. 2b).

Modified text: In order to place the OUTPACE cruise within the global hydrological context of the South Pacific, we have provided a basin scale description of the general circulation (Fig. 1b).

44. ***-too unspecific, could be at depth of 2000m***
   ***-the coupling is possible since these waters are salty? awkward contruction of sentence***

Original text (lines 199-201): At depth, the SEC transports thermocline waters which are salty (larger than 35), and thus makes it possible to spatially couple the ETSP and the WTSP.

Response: The sentence was indeed confusing. The SEC transports thermocline waters, but the depth is not detailed in the references. This has been rewritten in the revised manuscript.

Modified text:  The South Equatorial Current (SEC), which represents the northern branch of the southern subtropical anticyclonic gyre, transports thermocline waters (characterized by salinity values higher than 35) from the ETSP to the WTSP (Wyrtki, 1975; Solokov and Rintoul, 2000; Talley et al., 2011; Kessler and Cravatte, 2013).

**45. Tense ! divides into**

Original text (lines 200): Arriving to the MA, the SEC is divided into two branches:

Modified text:  Arriving at the Melanesian Archipelago, the SEC divides into two branches:

**46. -where does it bring the water?**
   **-again a flow direction whould help the reader to follow your text**

Original text (lines 201-204): The North Vanuatu Jet (NVJ) which brings salty, warm and relatively low oxygenated waters (Wyrtki, 1962b; Solokov and Rintoul, 2000; Webb, 2000) and the North Caledonian Jet (NCJ) which flows near 18° S between the surface and 1500 m and transports more oxygenated waters (Gourdeau et al., 2008; Kessler and Cravatte, 2013; Gasparin et al., 2014).

Response: Thank you for this comment. This has been specified in the revised manuscript.

Modified text: The North Vanuatu Jet (NVJ) which brings salty, warm and relatively poorly oxygenate waters to the west (Wyrtki, 1962b; Solokov and Rintoul, 2000; Webb, 2000), and the North Caledonian Jet (NCJ) which flows westward near 18° S between the surface and 1500 m and transports more highly oxygenated waters (Gourdeau et al., 2008; Kessler and Cravatte, 2013; Gasparin et al., 2014).

**47. Arriving at**

Original text (lines 204): At its arrival to the Australian coasts, the SEC bifurcates and becomes the East Australian Current (EAC)

Modified text: Approaching the Australian coast, the SEC bifurcates and becomes the East Australian Current (EAC) to the south and the North Queensland Current (NQC) toward the north and the Solomon Sea.

**48. wrong word! perhaps in French the word is the same for atmospheric whirls and marine eddies? in English this sounds odd**

Original text (lines 206): another one returning to the north through whirlwinds and meanders (Church 1987; Solokov and Rintoul 2000; Marchesiello and Middleton, 2000).

Response: We agree with Reviewer#2 and apologize for the mistake. 'Whirlwinds' has been deleted and replaced by 'eddies' in the revised manuscript.

Modified text: another returning to the north through eddies and meanders (Church 1987; Solokov and Rintoul 2000; Marchesiello and Middleton, 2000).

**49. Latter**
**50. awk, it cant flow through the island, if you mean that it flows through Cock Strait, say so (but that does not fit with 'north of'**
**51. awk (delete 'and')**

Original text (lines 207-209): The last part flows eastward along the Tasman Front (TF) (40°-45° S) through the north of New Zealand to become East Auckland Current (EAUC). It then joins back into and the circulation of the anticyclonic gyre through the South Pacific Current (SPC) which mostly circulates nearby the latitude of 30° S (Stramma et al., 1995; Talley et al., 2011).

Response: The mistakes have been corrected in the revised manuscript.

Modified text: The latter flows eastward along the Tasman front (40°-45° S) then toward the north of New Zealand to become the East Auckland Current (EAuC). The EAuC then flows back to join the circulation of the anticyclonic gyre *via* the South Pacific Current (SPC), which mostly circulates near the latitude of 30° S (Stramma et al., 1995; Talley et al., 2011).

**52. I suggest that you locate the OUTPACE cruise in the context of the circulation you described in this paragraph that was the idea of the paragraph, wasn't it.**

Original text (lines 211): The CTD profiles collected during the OUTPACE cruise take place in the southwestern Pacific region when the SEC encounters the Fiji and New Caledonia (Fig, 2a; 2b).

Response: We locate the OUTPACE cruise in the context of the circulation (see the modified text below).

**General comment:** We propose a new version of the "basin scale description of the general circulation" in the revised version of the manuscript. All mistakes have been corrected and the section has been rephrased as follows:

Modified text:

In order to place the OUTPACE cruise within the global hydrological context of the South Pacific, we have provided a basin scale description of the general circulation (Fig. 1b). The South Equatorial Current (SEC), which represents the northern branch of the southern subtropical anticyclonic gyre, transports thermocline waters (characterized by salinity values higher than 35) from the ETSP to the WTSP (Wyrtki, 1975; Solokov and Rintoul, 2000; Talley et al., 2011; Kessler and Cravatte, 2013).

Arriving at the Melanesian Archipelago, the SEC divides into two branches: The North Vanuatu Jet (NVJ) which brings salty, warm and relatively poorly oxygenate waters to the west (Wyrtki, 1962b; Solokov and Rintoul, 2000; Webb, 2000), and the North Caledonian Jet (NCJ) which flows westward near 18° S between the surface and 1500 m and transports more highly oxygenated waters (Gourdeau et al., 2008; Kessler and Cravatte, 2013; Gasparin et al., 2014). The OUTPACE cruise took place in the southwestern Pacific region where the SEC encounters the Fidji and New Caledonia islands (Fig, 1a; 1b). Approaching the Australian coast, the SEC bifurcates and becomes the East Australian Current (EAC) to the south and the North Queensland Current (NQC) toward the north and the Solomon Sea. Near 30° S, the EAC breaks down into several branches: one continuing south of 40° S, another returning to the north through eddies and meanders (Church 1987; Solokov and Rintoul 2000; Marchesiello and Middleton, 2000). The latter flows eastward along the Tasman front (40°-45° S), then toward the north of New Zealand to become the East Auckland Current (EAuC). The EAuC then flows back to join the circulation of the anticyclonic gyre *via* the South Pacific Current (SPC), which mostly circulates near the latitude of 30° S (Stramma et al., 1995; Talley et al., 2011).

**Section 3.1 Water masses and general biogeochemical trends**

> *53. This section is extremely descriptive; you seen to report each single number from the three tables. I suggest to reduce it by pointing to the essential differences and highlights. For the rest you have the tables and figures.*

Response: Yes, we agree with Reviewer#2. This section has been reviewed and rewritten in the revised manuscript (see the modified text below answer to comment 61).

**_54. in this sentence you provide many details of the SW, but I miss a clear definition of SW, is this the upper xx m? or what_**

Original text (lines 215-216): The surface waters (SW) were in the range between σ21.7 and σ23.5 (4.12-71.30 dbar, Table 1) and had a temperature range of 25.60 °C to 29.93 °C and were marked by small longitudinal variations (Fig. 4a).

Response: We agree with Reviewer#2. We have added a section "Division of the water column" in the Materials and Method (section 2.5 of the revised manuscript). We redefine the surface waters as the layer between the surface and the Mixed Layer Depth (MLD), calculated for each station's sample. The UTW and LTW is also defined in detail in this section.

Modified text:

2.5. Division of the water column

In order to describe and analyze the OUTPACE transect, four successive vertical layers were considered to divide the water column using density levels. The surface water layer corresponds to the layer between the surface and the Mixed Layer Depth (MLD). The MLD was defined as the depth where the density was greater than 0.125 kg m$^{-3}$ with respect to its value at 10 dbar (Levitus, 1982; Toyoda et al., 2015). The three layers defined below the MLD correspond to different water masses, and were obtained from an analysis of a Θ-S diagram (Fig. 2a) and of literature. In the WTSP, the Upper Thermocline Waters (UTW) are characterized by a maximum subsurface salinity (Tomczak and Hao 1989; Donguy 1994; Donguy and Henin 1997). The UTW layer was defined between the MLD and the depth where density (σ) = 25.40 (σ25.40). This layer includes the maximum of salinity observed on the Θ-S diagram (Fig. 2a). The underlying waters of the permanent thermocline are marked by wide ranges of temperature and salinity (Emery and Meincke, 1986; Sprintall and Tomczak, 1992; Tomczak and Godfrey, 1994). The Lower Thermocline Waters layer (LTW) is defined between σ25.40 and σ26.70, and corresponds to the waters marked by temperature and salinity gradients (Fig. 2a). The last layer between σ26.7 and σ27.3, corresponding to Subantarctic Mode Water and Intermediate Antarctic waters, is characterized by a minimum of salinity (McCartney, 1977; Tomczak & Godfrey, 1994) centered on σ27.1 (Fig. 2a).

**55. AOU is not a really meaningful concept in SW. I suggest to delete the two sentences on AOU in SW from the text.**

Original text (lines 218-219): The concentrations of AOU were close to zero or slightly negative in the SW for all the stations (Fig. 4b). A minimum of AOU concentrations of about -14 µmol kg$^{-1}$ was visible between σ23 and σ23.5 (Fig. 4b).

Response: We agree with Reviewer#2, the sentences have been removed from the reviewed version of the manuscript (see the modified text below answer to comment 61).

**56. Thermocline starts in 35 m depth? Hence part of the euphotic zone is in the thermocline, according to your definitions. Again, make clear why and how define SW, UTW, LTW. For example AOU<=0 holds down to sigma=24, for all except 1 station.**

Original text (lines 222): Between σ23.5 and σ25.4, the Upper Thermocline Waters (UTW) (35.42-251.52 dbar, Table 2) were characterized by a temperature and salinity range between 19.26 °C and 26.55 °C and 35.58 and 35.90, respectively (Fig. 4a).

Response: Please see the answer to comment 54.

**57. while you presented other aspect to the very detail you provide us with a very vague sentence here; increase in MA relative to GY? or do you only want to say that you observed a subsurface Chl-a maximum on (most?) stations?**

Original text (lines 226): An increase in Chl-a concentrations was observed in the upper part of the UTW (Fig. 4c).

Response: We have deleted the figure and the sentence relative to the chlorophyll in the revised manuscript (see the modified text below answer to comment 61)

**58. there is no gradient in this density range in Fig. 4d**

Original text (lines 226-228): N$_2$ fixation rates decreased sharply between σ23.5 and σ24.7 and reached homogeneous values between 0.01 and 1.73 nmol N L$^{-1}$ d$^{-1}$ in the MA waters and between 0.01 and 1.18 nmol N L$^{-1}$ d$^{-1}$ in the GY waters (Fig. 4d).

Response: We agree with Reviewer#2, the sentence has been corrected in the reviewed manuscript (see the modified text below answer to comment 61).

**59. Put your definition of LTW upfront of this sentence**

Response:  Please see the answer to comment 54.

**60. here and elsewhere; use precision carefully; 5 digits??**

Original text (lines 239): (496.91-1008 dbar, Table 4)

Response: We agree with Reviewer#2, all depths have been rounded in the revised manuscript.

**61. this excess in detail is torture ...**

Original text (lines 230-234): Between σ25.4 and σ26.7 (249.14-501.21 dbar, Table 3), a strong decreasing gradient of temperature and associated salinity marked the presence of Lower Thermocline Waters (LTW). The values of temperature and salinity varied little from east to west on σ25.40 and were $19.34 \pm 0.11\,°C$ and $35.61 \pm 0.03$ (Fig. 4a), respectively. In the LTW, the temperature and salinity ranged from 19.26 °C to 9.34 °C and from 35.72 to 34.54, respectively (Fig. 4a).

**General comment:** We propose a new version of section 3.1 in the revised manuscript. The excessive detail has been deleted and section 3.1 has been rephrased as follows:

Modified text:

**3.1 Water masses and general biogeochemical trends**

Figure 2a shows Θ-S diagrams (0-2000 dbar) for all 18 stations sampled during the OUTPACE cruise. The temperature range observed on the OUTPACE section varied from 29.9 °C at the surface to 2.0 °C at 2000 dbar depth (Fig. 2a). Salinity varied from 34.3 to 36.1 (Fig. 2a). The MLD ranged between σ21.9 and σ22.6, with an average value of $22.2 \pm 0.2$ kg m$^{-3}$ (corresponding to $19 \pm 9$ dbar), (Fig. 2b). The temperature and salinity observed in the surface waters were $29.4 \pm 0.4\,°C$ and $35.1 \pm 0.2$, respectively (Fig. 2a). A salinity maximum ($S_{max}$) was observed in the UTW between σ24 and σ25.4, centered on σ24.7 (Fig. 2a). Between σ25.4 and σ26.7 (249-501 dbar, Table 3), a strong decreasing gradient of temperature and salinity marked the presence of the LTW.

In the surface waters of the Melanesian Archipelago, $N_2$ fixation rates ranged from 2.37 to 42.25 nmol N $L^{-1}$ $d^{-1}$ (average rate 15.40 ± 10.44 nmol N $L^{-1}$ $d^{-1}$; Fig.3, Table 1). Rates decreased drastically towards the gyre waters, with values ranging between 0.01 and 2.29 nmol N $L^{-1}$ $d^{-1}$ (average 0.63 ± 0.80 nmol N $L^{-1}$ $d^{-1}$; Fig.3, Table 1). $N_2$ fixation rates showed high values at the top of the UTW above σ23.5 in the Melanesian Archipelago, and homogeneous low values between σ23.5 and σ25.4 (Fig.3). Conversely, the $N_2$ fixation rates were very low in the UTW of the gyre waters (Fig.3; Table 2). Below σ25.4, the $N_2$ fixation rates were undetectable (Fig.3; Table 2).

**Section 3.2: Inorganic nutrient pools**

*62. again much to many details!!!! rewrite and make this section readable by not reporting each single number of the tables, but by concentrating on the essential. Note that I do not add specific comment to this section, since I expect that you will strongly rewrite it and reduce the size.*

Response: We agree with Reviewer#2. This section was completely rewritten, and its size was reduced of approximatively 40% in the revised manuscript.

Modified text:

**3.2 Inorganic nutrient pools**

The concentrations of $NO_3^-$ were below the quantification limit in the surface waters for all sampling stations (Fig. 4a). The concentrations of $PO_4^{3-}$ were below the quantification limit or low in the surface waters of the Melanesian Archipelago, with a mean value (for measurable samples) of 0.07 ± 0.01 µmol $L^{-1}$ (Fig. 4b, Table 1). Significant higher $PO_4^{3-}$ concentrations (p < 0.01, Table S3) were observed in the surface waters of the gyre, with a mean value of 0.15 ± 0.01 µmol $L^{-1}$ (Fig. 4b, Table 1). The $NO_3^-$ concentrations became quantifiable below σ23.5. The $NO_3^-$ and $PO_4^{3-}$ concentrations increased gradually in the upper and lower thermocline waters. The bottom waters (> σ27.5, 1000 - 2000 dbar, 2000 dbar corresponding to the deeper depth sampled) had relatively homogeneous maximum concentrations, with values ranging between 35.1 and 36.5 µmol $L^{-1}$ and between 2.40 and 2.55 µmol $L^{-1}$ for $NO_3^-$ and $PO_4^{3-}$, respectively (Fig. 4a; Fig.4b).

The $NO_3^-$: $PO_4^{3-}$ ratio increased sharply in the UTW, and was significantly higher (p < 0.01, Table S4) in the Melanesian Archipelago than in the gyre (Fig. 4c, Table 2). In the LTW, the $NO_3^-$: $PO_4^{3-}$ ratio increased slightly and was significantly higher (p < 0.01, Table S5) in the Melanesian Archipelago than in the gyre (Fig. 4c, Table 3). In deep waters (below 27.5), the $NO_3^-$: $PO_4^{3-}$ ratio reached stable and uniform values, ranging between 13.8 and 15.2 (Fig. 4c). During the OUTPACE cruise, the distribution

of $NO_3^-$ *versus* $PO_4^{3-}$ concentrations followed a linear regression ($r^2 = 0.99$) with a slope of 15.0, which was close to the Redfield ratio of 16:1 (Fig. 4d). In the upper waters, $NO_3^-$ concentrations become depleted before those of $PO_4^{3-}$. In this layer, a deficit in $NO_3^-$ compared to the value expected from a Redfield point of view (dashed black line) was observed for all data (Fig. 4d).

*63. or NOx, compare Fig. 5 ??; be consistent*

Response: We agree with Reviewer#2 and we apologize for the mistake. However, in order to simplify our manuscript, we remove Figure 5. Inorganic nutrients are already presented and detailed in figure 4 of the revised manuscript.

**Section 3.3: Organic pools**

*64. my general comment to 3.2 applies here as well!!*

Response: We agree with Reviewer#2. In the revised manuscript, this section was entirely rewritten and its size was reduced by approximatively 45%. Furthermore, as said above, we removed the part related to the dissolved organic matter. Indeed, the elemental ratio of dissolved organic matter deviates widely from the canonical Redfield ratio (i.e. Hopkinson and Vallino, 2015; Letscher and Moore, 2015) mainly because of the turnover of the dissolved organic pool which is largely longer than that of the particulate organic pool, making the N excess interpretation for the dissolved pool more difficult. Considering only the particulate matter (biomass) and inorganic matter (nutrient) is the simplest scheme. However, since this simplest scheme allows to understand the main trends, it does not seem necessary to add more complexity and details to the story. The high PON excess may be explained by a significant $N_2$ fixation in the WTSP ($r^2 = 0.86$, p<001).

*65. ??*

Original text (lines 294): The highest MOP concentrations

Response: We apologize for the mistake, and we have corrected it.

Modified text: The highest POM concentrations

**General comment:** We propose a new version of section 3.3 in the revised manuscript. The excessive detail has been deleted and section 3.3 rephrased as follows:

Modified text:

**3.3 Organic pools**

The concentrations of PON and POP showed maximum values in the surface waters of the Melanesian Archipelago, with values ranging between 0.32 and 1.18 µmol $L^{-1}$, and between 0.02 and 0.06 µmol $L^{-1}$, respectively. They were significantly higher (p < 0.01, Table S3) in the Melanesian Archipelago than in the gyre (Fig. 5a, Fig. 5b; Table 1). In the UTW, a significant attenuation of POM concentrations was observed. The mean concentrations of PON and POP remained, however, significantly higher (p < 0.01, Table S4) in the UTW of the Melanesian Archipelago than those of the gyre (Fig. 5a, Fig. 5b, Table 2). POM concentrations decreased to reach minimum and relatively constant values in the LTW. A significant relationship (p<0.001; Pearson test) was obtained between PON concentrations and $N_2$ fixation ($r^2$=0.56). The PON concentrations in excess showed a contrasted distribution in the surface waters of the studied areas. Indeed, in the Melanesian Archipelago, the highest values were observed between the surface and the top of the UTW (< σ22.5, ≈ 0-25 dbar), with a maximum of 0.37 µmol $L^{-1}$ (Fig. 5c). In the gyre, the PON concentrations in excess exhibited values around zero. Below σ23.5, the PON concentrations in excess showed a similar distribution in both areas studied, with values close to those calculated based on Redfield (Fig. 5c). A significant relationship (p<0.001; Pearson test) was obtained between PON concentrations in excess and $N_2$ fixation ($r^2$=0.86). For all the sampling stations, the PON:POP ratio was between 10.08 and 25.78 (figure not shown). The average value of the PON:POP ratio was 16.9 ± 2.4, which included the Redfield ratio (16.0). Nevertheless, a strong variability was observed between the different oceanic regions (Melanesian Archipelago *vs* gyre) and the different water masses considered. In the surface waters, the PON:POP ratio was significantly higher (p < 0.01, Table S3) in the Melanesian Archipelago than in the gyre (Table 1). In the UTW, the PON:POP ratio was not significantly different (Table S4) between the Melanesian Archipelago and the gyre**.** The distribution of PON *versus* POP concentrations followed a linear regression in both studied areas. In the gyre, a slope of 15.1 was observed, which was close to the Redfield ratio (blue line in Fig. 4d). In the Melanesian Archipelago, the slope observed deviates from the expected Redfield ratio, showing a value of 20.0 (red line in Fig. 5d).

**Section 3.4: Distribution of N*, $N_2$ fixation rates and variations of $PO_4^{3-}$ pool**

**General comment:** The title of this section has been rephrased.

Modified text: 3.4 Regional distribution of N* and $N_2$ fixation rates

**66. I had hope that now, after almost 3 pages of boring data report, some more interesting science starts to show up. However, you continue with excessive details.**

Response: We apologize for the excessive detail in this section. We have corrected all the mistakes. We have rephrased the entire section and reduced the size by approximately 45 % in the revised version of the manuscript (see the modified text below answer to comment 74).

**67. the range did not vary (since you give only one range); N\* varied here and elsewhere: try to write more simple, not overly complex (and ambigious)**

Original text (lines 313): The range of N\* concentrations varied between -2.52 and 2.15 µmol L$^{-1}$ along the transect (Fig. 10).

Response: Please see the modified text below answer to comment 68.

**68. if this is your definition of SW, give it much earlier, best in M+M; but I doubt that such a simple and formal definition is what you had in mind ...**

Original text (lines 314): A positive N\* anomaly between 0.23 and 2.15 µmol L$^{-1}$ was observed in the SW, (0-50 dbar) coupled with a strong disparity of the signal marked by values between 0.23 and 1.19 µmol L$^{-1}$ in the SW of the GY and between 0.25 and 2.15 µmol L$^{-1}$ in the SW of the MA (Fig. 10).

Response: Please see the answer to comment 54.

Modified text: At the scale of the OUTPACE cruise, N\* showed positive values ranging between 0.23 and 2.15 µmol L$^{-1}$ in the surface waters. N\* was significantly higher ($p < 0.01$, Table S3) in the Melanesian Archipelago than in the gyre (Fig. 6).

**69. see remark above, SW (0-50 dbar)**

Original text (lines 316): The UTW and LTW (100-500 dbar) …

Response: Please see the answer to comment 54.

***70. -again you repeat each number from the table ...***

   ***-see general remark on details, above***

Original text (lines 318): In the MA, the N* signal in UTW was positive (mean=1.11 ± 0.53) with values ranging between -0.12 and 2.13 μmol L$^{-1}$ and negative in the GY (mean = -0.14 ± 0.49) with values between -1.13 and 0.75 μmol L$^{-1}$ (Fig. 10b, Table 2).

Original text (lines 320-322): In the LTW, the N* concentrations ranged between -0.49 and 1.52 μmol L$^{-1}$ (mean=0.75 ± 0.54) in the waters of the MA and between -1.55 and 0.72 μmol L$^{-1}$ (mean= -0.51 ± 0.78) in GY (Fig. 10b, Table 3).

Response: Please see the modified text below answer to comment 74.

***71. a disparity marked by differences; 'In the LTW, N* was different in MA and GY waters.*'***

Original text (lines 318-319): In the LTW, a disparity in N* concentrations was observed, marked by a difference between the MA and GY waters.

Response: We agree with Reviewer#2 and have rephrased this sentence (see the modified text below answer to comment 74).

***72. -do you mean 'below'***

   ***-awk, do you mean that variability decreased with depth?***

Original text (lines 323): Above 500 dbar depth, the dispersion of the signal was attenuated, …

Response: yes, we mean 'below' and the variability decreased with depth. We apologize for the mistake and have corrected this sentence. The abbreviations have been removed (see the modified text below answer to comment 74).

***73. sounds awk: in MA you integrated over the water column (SW+upper thermocline, I guess), but in GY you integraed only over SW?***

Original text (lines 333): A sharp decrease in integrated N$_2$ fixation rates was observed in the SW of the GY.

Response: Yes, integrated $N_2$ fixation rates were calculated over the water column (surface waters +UTW) both in the Melanesian Archipelago and in the gyre. We apologize for the mistake and have added the details in the Materials and Methods section.

*74. see my comment in M+M*

Original text (lines 335): The average difference in $PO_4^{3-}$ concentrations observed between the SW of the MA and the GY ($\Delta PO_{4\ observed}^{3-}$) was -0.07 $\pm$ 0.01 µmol L$^{-1}$. From the estimation of N* in the SW, the difference between the average concentrations observed in N* ($\Delta N*$) in the SW of the MA and the GY was 1.05 $\pm$ 0.37 µmol L$^{-1}$. The variation in $PO_4^{3-}$ concentrations induced by the excess supply of nitrogen in the MA ($\Delta PO_{4\ estimated}^{3-}$) was equal to -0.07 $\pm$ 0.03 µmol L$^{-1}$.

Response: As written previously, this section was deleted in the revised manuscript.

**General comment:** In the revised text, all the corrections referred to above have been taken into account, and we propose a new version of section 3.4.

Modified text:

**3.4 Regional distribution of N* and $N_2$ fixation rates**

At the scale of the OUTPACE cruise, N* showed positive values ranging between 0.23 and 2.15 µmol L$^{-1}$ in the surface waters. N* was significantly higher (p < 0.01, Table S3) in the Melanesian Archipelago than in the gyre (Fig. 6). In the UTW and the LTW (100-500 dbar), N* showed a significant difference (p < 0.01, Table S4 and S5) between the Melanesian Archipelago and the gyre waters. In the UTW, N* mean was positive in the Melanesian Archipelago and negative in the gyre (Fig. 6b, Table 2). In the LTW, below 150 dbar depth, N* exhibited a marked decrease, with a minimum value of -1.55 µmol L$^{-1}$ at 400 dbar in the gyre. In the same layer, N* was significantly higher (p < 0.01, Table S5) in the Melanesian Archipelago than in the gyre (Fig. 6b, Table 3), with a minimum value of -0.77 µmol L$^{-1}$ at 500 dbar (Fig.6a). Below 500 dbar, N* variability decreased with depth and reached mostly negative values at 2000 dbar for all sampling stations (Fig. 6a).

Regarding the horizontal variability of N*, over the upper ($\sigma$24.7) and lower ($\sigma$26.3) thermocline layers (Fig. 7), N* showed higher values in the Melanesian Archipelago than in the gyre. In the Melanesian Archipelago (between SD1 and SD12), N* ranged between 1.05 $\pm$ 0.01 and 1.77 $\pm$ 0.02 µmol kg$^{-1}$ and between -0.02 $\pm$ 0.02 and 1.21 $\pm$ 0.02 µmol kg$^{-1}$ in the upper and lower thermocline layers, respectively. Beyond SD 12, a drastic decrease of N* was observed in both upper and lower thermocline layers

(between SD12 and SD15), reaching minimum values of -0.68 ± 0.07 µmol kg $^{-1}$ and -1.46 ± 0.02 µmol kg $^{-1}$ in the upper and lower thermocline layers, respectively (Fig. 7).

In this study, a significant relationship ($p < 0.001$; Pearson test) was obtained between N* and $N_2$ fixation ($r^2$=0.59). Vertically integrated $N_2$ fixation rates in the surface waters exhibited the same trend as N*, with higher values in the Melanesian Archipelago than in the gyre. In the Melanesian Archipelago, $N_2$ fixation rates ranged between 100 and 840 µmol N m $^{-2}$ d $^{-1}$, with an average integrated $N_2$ fixation rate of 443 ± 204 µmol N m $^{-2}$ d $^{-1}$ (red circle in Fig. 7). As for N*, a sharp decrease in integrated $N_2$ fixation rates was observed in the gyre, with values ranging between 5 and 73 µmol N m $^{-2}$ d $^{-1}$ and with an average value of 28 ± 39 µmol N m $^{-2}$ d $^{-1}$ (blue circle in Fig. 7).

**Section 3.5: Distribution of N* signal at the South Pacific scale**

**General comment:** The title of this section has been rephrased.

Modified text: 3.5 Large scale distribution of N* in the South Pacific

> *75. Here and elsewhere in this paragraph: sections are not indicated in fig 12, hence you need to point to the Fig 3 as well. But moving between Fig. 3 and 12 is very impractical, hence I suggest to move Fig. 3 to panel (c) of Fig. 12. (or make Fig. 12a, b panels of Fig. 3)*

Response: We agree with Reviewer#2, we have moved figure 3 to panel (c) of Fig.12 in the revised manuscript.

**General comment:** Like section 3.4, section 3.5 has been corrected and rewritten. We have reduced the length by approximately 40 % in the revised manuscript.

Modified text:
**3.5 Large scale distribution of N* in the South Pacific**

At the basin scale of the South Pacific, N* was minimum in the upper thermocline (σ24.7) between 5° S and 20° S, and between 80° W and 100° W (P21, P18, P19 sections in Fig. 8a). Between 5° S and 15° S, N* showed a westward increase along the northern branch of the SEC. In the WTSP, N* increased and exhibited maximum values at 155° E, with an average value of -0.27 ± 0.97 µmol kg $^{-1}$ (P11S section in Fig. 8a). South of 15° S, N* increased more rapidly than north of 15° S. In the WTSP, between 175° E and 150 °E, N* increased sharply during the westward transport of waters by the NCJ (Fig. 1b), reaching a value of 0.39 ± 0.58 µmol kg $^{-1}$ (P21 section in Fig. 8a). The maximum of N* was observed in the Coral

As observed in the upper thermocline, N* showed minimum values in the lower thermocline around the OMZ region, between 80° W and 90° W and at 18° S (P21, GT13 sections in Fig. 8b, Fig. S2), and increased westward. The maximum values of N* were observed in the Melanesian Archipelago waters, in the southern Coral and Tasman seas, exhibiting an average value of 0.52 ± 0.12 µmol kg$^{-1}$ (P14C section in Fig. 8b, Fig. S2). In the Tasman Sea and north of New Zealand, at 30° S, and between 150 °E and 180 °E, N* showed high values, with an average of 0.66 ± 0.46 µmol kg$^{-1}$ (P06 section in Fig. 8b, Fig. S2). These high N* values persist in the central part of the South Pacific, between 23° S and 32° S and 175° E and 135° W, with an average value of 0.42 ± 0.31 µmol kg$^{-1}$. In the eastern part of the South Pacific, at 110° W and between 85° W and 90° W (P18, P19 sections in Fig. 8b, Fig. S2), a drastic decrease of N* was observed. At 32° S and close to the South America coast, N* dropped sharply to reach an average value of -17.33 ± 2.63 µmol kg$^{-1}$ (P06 section in Fig. 8b, Fig. S2).

**Section 4: Discussion**

**Section 4.1: Sharp increase of N* in the surface waters of the MA**

**Section 4.1.1: Nitrogen supply and PO$_4^{3-}$ uptake by N$_2$ fixing organisms in the SW of the MA**

**General comment:** In the revised manuscript, the discussion has been rewritten in a clearer and more concise way. Section 4.1.1 has been deleted and we have focused the Discussion on the effect of intense N$_2$ fixation on the particulate organic N pools and on the N* in the WTSP, and on the N* distribution in the thermocline waters at the scale of the South Pacific Ocean.

*76. write out*

**Original text (line371 and 376):**

4.1 Sharp increase of N* in surface waters of the MA

4.1.1 Nitrogen supply and PO$_4^{3-}$ uptake by N$_2$ fixing organisms in the SW of the MA

Response: The acronyms 'MA' and 'SW' have been replaced by 'Melanesian archipelago' and "surface waters", in the text.

*77. ?? competition with ordinary phytoplankton*

Original text (lines 379): A low concentration of NO$_3^-$ characteristic of oligotrophic environment was observed in the upper water of the gyre. Deutsch et al. (2007) suggested that waters deficient in NO$_3^-$

were a favorable environment for $N_2$ fixation by probably limiting interspecific competition and favoring organisms adapted physiologically to nitrogen-deficient environments.

Response: Yes, this sentence relates to a competition with non-diazotroph phytoplankton. This part has been removed in the last part of the Discussion (see the modified text below answer to comment 123).

**78. here and elsewhere: use consistent abreviations in a ms**

Original text (lines 381): The average $PO_4^{3-}$ concentrations are higher in the SW of the GY ($PO_4^{3-}{}_{obs(GY)}$ =0.15 ± 0.02 µmol $L^{-1}$) than in the SW of the MA ($PO_4^{3-}{}_{obs(AM)}$ =0.08 ± 0.03 µmol $L^{-1}$),

Response: As stated previously, this section has been deleted from the revised manuscript and most of the abbreviations have been corrected or deleted.

**79. which is obvious from the number you gave, right delete?**

Original text (lines 382): where concentrations in $PO_4^{3-}$ are almost depleted.

Response: We agree, but the sentence has been deleted in the revised version of the paper.

**80. This sentence is unreadable!!! You seem to construct a contrast (on the other hand), but by there is no. but isn't this sentence anyway circular reasoning; again: circular reasoning! To be honest: I get headache from this text! (sorry, dear author, to say but it is really true).**

Original text (lines 382-387): On the other hand, the variability of $PO_4^{3-}$ between the surface MA's waters and of that of the GY ($\Delta PO_4^{3-}{}_{estimated}$), associated with the difference of $NO_3^-$ in excess ($\Delta N^*$), were -0.07 ± 0.03 µmol $L^{-1}$ and are of the same order of magnitude as the variations of $PO_4^{3-}$ observed between both regions ($\Delta PO_4^{3-}{}_{observed}$ = -0.07 ± 0.01 µmol $L^{-1}$). If in the SW, the $N^*$ variations are solely due to the diazotrophic activity, these results suggest that the decrease in $PO_4^{3-}$ and the associated increase in $N^*$ in the MA waters are controlled by diazotrophic organisms.

Response: We apologize for this confusing section. The whole of the paragraph above has been deleted from the revised manuscript.

**81. I dont understand the constructed contrast of this (whole) sentence**

Original text (lines 393-398): Our results support those of Deutsch et al. (2001), who initially suggested that the high N* values observed in the SW in the WTSP near Australia are due to significant $N_2$ fixation in this region (Messer et al., 2016; Bonnet et al., 2015), before proposing higher $N_2$ fixation rates in the GY (Deutsch et al., 2007). The higher DFe availability coupled with an absence of $NO_3^-$ in the SW of the MA could stimulate the diazotrophic activity, which in turn will introduce excess nitrogen, deplete the SW in $PO_4^{3-}$ and be the explanation for the positive N* anomaly in the MA waters.

Response: We agree, but the sentence has been deleted in the revised version of the paper.

**Section 4.1.2: Excess nitrogen supply via remineralization of diazotrophic organic material**

**General comment:** The title of this section has been rephrased.

Modified text: 4.1. Excess nitrogen supply *via* remineralization of diazotrophic organic material in the Western Tropical South Pacific

82. *-Typical' is enough to generalize*
    *-for such a general statement, you need more than one reference*

Original text (lines 400-403): Direct POM measurements showed significant accumulation of particles in the SW and attenuation at the top of the UTW (Fig. 7a, 7b). Despite extremely low nutrient concentrations, a significant amount of POM was measured in particular for the MA stations with PON and POP concentrations reaching 1.175 and 0.057 μmol $L^{-1}$ in the SW, respectively. Typical PON concentrations generally measured in oligotrophic regions are lower than 0.6 μmol $L^{-1}$ (Martini et al., 2013).

Response: We agree, the paragraph has been corrected, and additional references have been added (Hebel and Karl, 2001; Moutin and Raimbault, 2002).

Modified text: POM measurements showed a significant accumulation of particles in the surface waters and at the top of the UTW, followed by an attenuation in the central and lower part of the UTW (Fig. 5a, 5b). Despite the extremely low surface nutrient concentrations observed, a significant amount of POM was measured in the Melanesian Archipelago sampling stations, with PON and POP concentrations reaching 1.175 and 0.057 μmol $L^{-1}$, respectively. PON concentrations in oligotrophic regions are typically <0.6 μmol $L^{-1}$ (Hebel and Karl, 2001; Moutin and Raimbault, 2002; Martini et al., 2013).

83.  *you way want to change the direction of your argument? 'are usually larger than'*

84.  *again Martini should not be the only source, in particular for 'bloom waters'; there are many more papers*

Original text (lines 403-406): The observed values, particularly at station LD B, are only slightly less than those measured in the Low Nutrient High Chlorophyll (LNHC) zones of northern latitudes (Sarmiento and Gruber, 2006), which generally have a lower limit of 1.2 µmol L$^{-1}$ in PON (Martini et al., 2013).

Response: We agree, the paragraph has been corrected, and the missing references have been added (Gruber, 2008).

Modified text: The observed values, and particularly at station LD B, were in the same order of magnitude as those measured in subpolar waters of northern latitudes (Sarmiento and Grüber, 2006), which are usually >1.2 µmol L$^{-1}$ in PON (Martini et al., 2013; Gruber 2008).

85. *units are not comparable, rewrite*

86. *what about seasonality in that region? you never consider this, there is no? or could it be that excess PON represents taken-up NO3 in a seasonal context?*

Original text (lines 408-409): This excess of PON can be as high as 0.370 µmol L$^{-1}$ compared to RR, despite undetectable NO$_3^-$ concentrations, confirming an external nitrogen supply to the system that can support organic production.

Response: We agree with Reviewer#2, the units were not comparable. The sentence has been rephrased (see the modified text below).
Seasonal variations were the subject of a companion paper in the SI by Moutin et al. (2018). The top of the nitracline depths are mostly below the Winter mixed layer depths, which is not the case for the top of the phosphacline depths. Hence, this prevents nitrate replenishment of the upper layer during Winter mixing, but provides phosphate, which may appear as an important cause explaining the success of nitrogen fixers in the WTSP. The excess PON seems therefore more related to N$_2$ fixation than NO$_3^-$ variation in a seasonal context.

Modified text: The PON:POP ratios averaged over the entire water column were very close to the Redfield ratio, except for the most productive zones located in the Melanesian Archipelago, which had an excess of PON relative to that expected from Redfield stoichiometry, particularly close to the surface (Fig. 5d).

This excess of PON was as high as 0.37 μmol L$^{-1}$, suggesting that the POM was enriched in N, despite the undetectable NO$_3^-$ concentrations.

**87. sounds a bit like wild speculation; I do not have the Karl at al book chapter at hand? how solid was the ground on which the often-cited value of 125 rests? was this a community dominated by fixers, while yours potentially isn't? rewrite to be less speculative**

Original text (lines 412-417): This difference could be explained because PON:POP ratio of 125 observed by Karl et al. (1992) is an exceptionally high value related to an intense bloom that probably does not reflect the general relationships associated with diazotrophs growth.

Response: The N:P ratio of 125 corresponds to the chemical composition of suspended particulate organic matter collected during a *Trichodesmium* bloom (august 1989). The sentence has been rewritten more clearly in the revised manuscript (see modified text below answer to comment 92).

**88. perhaps you shoul rewrite this whole section and present the literature evidence from a more general point**

Original text (lines 414-417): Letelier and Karl (1998) observed mean ratios associated with *Trichodesmium* sp. diazotrophs between 40 and 50, which nevertheless remains much higher than the observations made during the OUTPACE cruise. Most studies reporting extremely high ratios were carried out either in the North Pacific Gyre (Karl et al., 1992), or in the Atlantic (Hansell et al., 2004), the WTSP remaining to date an under-sampled region.

Response: We agree with Reviewer#2, the whole section has been rewritten. The literature is presented from a more general point (see the modified text below answer to comment 92).

**89. -perhaps make more clear that the function is of 'inverse' type, otherwise the sentence works awk**

Original text (line 418-419): Since the cellular stoichiometry of organisms is a function of nutrient availability (Karl et al., 2002), PO$_4^{3-}$ deficiency largely higher in the MA than in the GY (Moutin et al., this issue) may explain the higher PON:POP ratios in the northern hemisphere than in the WTSP.

Response: The sentence has been rephrased in the revised version of the manuscript (see the modified text below answer to comment 92).

> **90.** *-here and elsewhere: writing style whenever possible don't come from the reference, but from the fact (and give the reference in brackets)*
> *-simplify phrase*

Original text (line 421-423): Finally, according to Mulholland et al. (2007), the more or less significant amount of nitrogen associated with diazotrophic biomass can after mineralization in the productive layer eventually be at the origin of the establishment of a particular food web.

Response: We thank Reviewer#2 for this suggestion. The sentence has been rephrased in the revised version of the manuscript (see the modified text below answer to comment 92).

> **91.** *-awk expression; this sentence would benefit from a more clear language and a clear msg*
> *- simplify sentence*
> **92.** *this whole paragraph is too much talking around; be more to the point about what you think are the major differences between your system and the NSP studied by Karl etc. I suggest to rewrite the paragraph to more integrative and not deal with the different studies in isolation.*

Original text (line 424-427): A much more intensive pattern of diatom development has been observed in MA waters (Brunet, 2016, pers. comm.). It is generally accepted that diatoms are characterized by a low PON:POP ratio (Moore et al., 2013), which could contribute to reducing the observed PON excess in comparison with the ratios measured by Karl et al. (1992) and Letelier and Karl (1998).

**General comment:** We propose a new version of the whole paragraph in the revised version of the manuscript.

Modified text:

POM measurements showed a significant accumulation of particles in the surface waters and at the top of the UTW, followed by an attenuation in the central and lower part of the UTW (Fig. 5a, 5b). Despite the extremely low surface nutrient concentrations observed, a significant amount of POM was measured in the Melanesian Archipelago sampling stations, with PON and POP concentrations reaching 1.175 and 0.057 μmol L$^{-1}$, respectively. PON concentrations in oligotrophic regions are typically <0.6 μmol L$^{-1}$ (Hebel and Karl, 2001; Moutin and Raimbault, 2002; Martini et al., 2013). The observed values, and particularly at station LD B, were in the same order of magnitude as those measured in subpolar waters of northern latitudes (Sarmiento and Grüber, 2006), which are usually >1.2 μmol L$^{-1}$ in PON (Martini et al., 2013; Gruber 2008). The PON:POP ratios averaged over the entire water column were very close to the Redfield ratio, except for the most productive zones located in the Melanesian Archipelago, which had an excess of PON relative to that expected from Redfield stoichiometry, particularly close to the

surface (Fig. 5d). This excess of PON was as high as 0.37 μmol L$^{-1}$, suggesting that the POM was enriched in N, despite the undetectable NO$_3^-$ concentrations.

Most PON:POP ratio data available in the literature corresponds to cruises in the North Pacific (Karl et al., 1992; 2001) and North Atlantic (Hansell et al., 2004), while the WTSP has so far remained an undersampled region. These studies showed that the chemical composition of the suspended particulate matter collected during *Trichodesmium* blooms was enriched in N relative to P with PON:POP ratios ranging between 40 (Letelier and Karl, 1998) and 125 (Karl et al., 1992). The stoichiometric PON:POP ratios observed during the OUTPACE cruise were much lower, with a maximum of 25.3 observed at station LD B. The differences observed in the PON:POP ratios may be explained by differences in nutrient availability, as previously suggested by Karl et al. (2002). Oligotrophic gyres in the northern hemisphere have lower concentrations of PO$_4^{3-}$ than those measured in the southern hemisphere (Moore et al., 2013), theoretically leading to higher PON:POP ratios in the North than in the South Pacific. Adding to in situ nutrient availability, new N derived from diazotrophic activity would allow the successive development of non-diazotrophic microorganisms (Caffin et al., 2018; Leblanc et al., 2018; Mulholland, 2007) with lower N:P content than diazotrophs, leading to a relatively lower PON:POP ratio.

93. *-which are your assumptions to conclude from your single cruise coverage to the annual time scale; you need to make those known to the reader; what is your underlying understanding of seasonality on an annual time scale*
*-unclear: do you consider here annual variations of N\* or is this just your 'N\* anomaly', i.e. N\**
*-awk contrast of formation of diaztrophic organic matter and fixation of N2 (in the context of reducing PO4)*

Original text (lines 428-429): On an annual time scale, the strong anomalies of surface N\* can be explained, on the one hand by the formation of diazotrophic organic matter, which together with the fixation of N$_2$ reduces the stock of PO$_4^{3-}$ and positively influences the N\* signal.

Response: We agree with Reviewer#2. From our single cruise coverage, we cannot conclude in an annual time scale variation of N\*. Here, we consider only our N\* data.
The contrast between the formation of diazotrophic organic matter and the N$_2$ fixation has been deleted in the revised manuscript.

Modified text: The high N\* values observed in the surface waters and at the top of the UTW of the Melanesian Archipelago can be explained by the remineralization of organic matter enriched in N

occurring in these two layers. Our results suggest that N supply by remineralization of diazotroph biomass and consumption of $PO_4^{3-}$ by $N_2$ fixers induces an excess of N in the WTSP.

**94. -this is against your own definition of fixed nitrogen, you mean inorganic N**

**-Appart from that the language of this sentence is overly complex, I think that you are undercomplex in the interpretation of N* (as an integrative measure) vs PON**

Original text (lines 428-429): On the other hand, part of the excess of PON which goes after remineralization towards the stock of fixed nitrogen may therefore also positively influence the N* signal.

Response: We agree with Reviewer#2, 'fixed nitrogen' has been replaced by 'inorganic nitrogen'. The interpretation of N* has been rephrased (see the modified text below answer to comment 95).

**95. awk: what do you mean here? The N* is not subject to overestimation (it has a simple equation how to compute it). The interpretation of N*, however, is subject to unaccounted (unquantified) processes that modify N* (those you mention later in this sentence**

Original text (lines 433-436): Nevertheless, Benavides and Voss (2015) point out that the N* signal could be overestimated by nitrogen inputs that are not attributable to the diazotrophic process, such as: mineralization of non-diazotrophic organisms with a high PON:POP ratio, lateral advection of DOM with a high DON:DOP ratio (Loetscher et al., 2013), and atmospheric nitrogen input.

Response: Yes, we agree with Reviewer#2, the sentence was confusing. N* could be influenced by processes other than $N_2$ fixation. This sentence has been rephrased in the revised manuscript.

Modified text: However, N* could be influenced by processes other than $N_2$ fixation, such as the non-Redfield nutrient assimilation by non-diazotrophic phytoplankton (Mills and Arrigo, 2010; Singh et al., 2017), and/or remineralization of non-diazotrophic organisms (Zamora et al., 2010; Singh et al., 2013). Furthermore, N* could also be influenced by lateral advection of DOM with a high DON:DOP ratio (Letscher et al., 2013), and atmospheric deposition of low N:P ratio aerosols (Jickells et al., 2017). The transport is probably another process which affects N* distribution in the ocean interior (Gruber and Sarmiento, 1997).

**96. N2 fixation are largely restricted in surface waters (fig xx; Deutsh 2001), have no direct impact in the thermocline N*.**

Original text (lines 438-440): As highlighted by Deutsch et al. (2001), the uptake of $PO_4^{3-}$ by diazotrophic organisms no longer occurs and therefore will have no impact on the N* signal

Response: This sentence has been rephrased in the revised version of the manuscript.

Modified text: Our results suggest that N supply by remineralization of diazotroph biomass and consumption of $PO_4^{3-}$ by $N_2$ fixers induces an excess of N in the WTSP.

**97. Can this really be said? You have the subsurface Chla max in this layer, which points to production. The gradient in AOU could just be due to transport towards the mixed SW waters which rapidly exchange with the ATM.**

Original text (lines 441-442): The results showed that UTW are a preferential remineralization zone, marked by increasing AOU values (Fig. 4b) and a significant attenuation of POM concentrations (Fig.7a, 7b).

Response: In ultra-oligotrophic regions considered as stable environments, many authors (Steele, 1964; Taylor et al., 1997; Fennel and Boss, 2003; Mignot et al., 2011) showed that the subsurface chlorophyll maximum layer is mainly due to the process of photoacclimatation, and not to the variation of biomass. So the observed subsurface chlorophyll maximum does not point to a maximum of production, and has to be considered independently from a maximum of biomass. The gradient in AOU could not just be due to transport towards the mixed surface waters. Mixed layer depth ranged between 11 and 39 dbar, and the gradient of AOU was observed deeper. But in order to clarify our discussion, the data and figures relative to Chl-*a* and AOU have been removed in the revised manuscript.

**98. awk: you just said that xx can not explain it ?? So far you did not explain what could explain the decrease of N* with depth**

Original text (lines 446-447): These results can partly explain the weak anomaly observed.

Response: We agree with Reviewer#2, in order to clarify our discussion, the sentence has been deleted in the revised version of the manuscript.

**99. awk: A concept of N* can not reveal a complexity.**

Original text (lines 447): The concept of N* reveals its full complexity in the thermocline waters.

Response: We agree with Reviewer#2, the sentence has been deleted in the revised version.

**100.** *is it really a coupling or an interference/overlapping?*

Original text (lines 448-449): Indeed, as pointed out by Deutsch et al. (2001), signal coupling between denitrification regions and $N_2$ fixation regions should be considered in the interpretation of the local N* signal, which is due to $N_2$ fixation and the mixing process undergone by a water parcel.

Response: Deutsch et al. (2001) used the term 'coupling of signal', but as suggested by Reviewer 2, the term 'overlapping of signal' seems more appropriate. This sentence has been rephrased in the revised manuscript. For more clarity, the sentence was moved to section 4.3 of the revised version (see below the answer to comment 123).

**101.** *Overall, there is no clear msg from this paragraph*

**General comment:** In order to clarify our Discussion, the whole paragraph (line 438-449 of the original manuscript) has been deleted.

**Section 4.2: The WTSP, a source of excess nitrogen for the thermocline waters of the South Pacific**
**Section 4.2.1: Westward transport of the negative N* signal**

**General comment:** In the revised version of the manuscript, we chose to focus on the Eastward transport of the positive N* produced in the Melanesian Archipelago to the south Pacific gyre. The whole section 4.2.1 has deleted.

**102.** *Superfluous, in particular in combination with the next sentence.*

Original text (lines 451): From a biogeochemical point of view, the thermocline waters are of a crucial importance.

Response: We agree with Reviewer#2 and the sentence has been deleted in the revised manuscript.

**103.** *awk expression*

Original text (lines 458-459): This negative N* anomaly increased with water subduction centered on σ24.7 in the central Pacific (18°-20° S, 100°-150° W, Fig. 12a, Fig. S1).

Response: We agree, this sentence has been rewritten. For more clarity, this sentence has been moved to section 4.3 of the revised manuscript.

Modified text: In this study, we observed that the surface eastern central Pacific waters with low N\* (high P\*) sink on isopycnal σ24.7 (18°-20° S, 100°-150° W, Fig. S1a).

**104.     *where is this, somewhere at the equator? (central wr to N-S) ?***

Original text (lines 457-459): Tomczak and Hao, (1989) and Solokov and Rintoul, (2000) showed that SLW (or SPEW) was transported eastward from the central part of the Pacific and had a strong influence on the water masses in the Coral Sea and the Solomon Sea.

Response: The location has been added to the revised version of the manuscript. For more clarity, this sentence has been moved to section 4.3 of the revised version.

Modified text: Tomczak and Hao, (1989) and Solokov and Rintoul, (2000), showed that SPEW is transported westward from the central part of the South Pacific (12°-25° S; 100°- 150° W), and has a strong influence on the water masses in the Coral Sea and the Solomon Sea.

**105.     *Awk***

Original text (lines 478-480): It has been shown that the thermocline waters and more particularly those located at the depth of the σ26.3 (Fig. 12b) will be also the seat of variations of N\* signal induced by the preferential transport through the SEC of the strongly negative N\* anomaly from the OMZ in the ETSP.

Response: The sentence was a repetition and has therefore been deleted.

**106.     *awk phrase; either 'nitrogen remineralization' or 'nitrogen supply from remineralization'***

Original text (lines 483): remineralization nitrogen supply

Response: We thank Reviewer#2 for this comment. The sentence was also a repetition and has therefore been deleted.

**Section 4.2.2: Eastward transport of the positive N\* signal**

**General comment:** The title of this section has been rephrased.

Modified text: **4.2. Eastward transport of the positive N\* produced in the Melanesian Archipelago to the South Pacific gyre**

*107.       awk expression*

Original text (lines 488): Bonnet et al. (2017) combined the $N_2$ fixation measurements from 6 cruises in the WTSP (including OUTPACE cuise) and concluded for a hot spot of $N_2$ fixation in this region.

Response: The sentence has been rephrased in the revised version.

Modified text: South of 15° S in the WTSP, the positive N\* observed in both the upper ($\sigma$24.7) and lower ($\sigma$26.3) thermoclines (Fig. 8a, 8b, Fig. S2) can be explained by the high levels of $N_2$ fixation observed in this region (Bonnet et al., 2017).

*108.       awk expression*

Original text (lines 494): The northern SLW outcroped in Winter in the central Tasman Sea and were converted into at the dense southern variety of SLW (Solokov and Rintoul, 2000).

Response: The sentence has been rephrased in the revised version.

Modified text: The northern Subtropical Lower Waters outcropped in Winter in the central Tasman Sea and are converted to the dense southern variety of Subtropical Lower Water (Solokov and Rintoul, 2000).

*109.       TKG (fig 2b), right ?*

Original text (lines 506)

Response: We thank Reviewer#2 for this comment. The correct term is 'TKR = Tonga Kermadec Ridge'. The term has been corrected in the modified text and figure.

**110.    what is this?, awk expression**

Original text (lines 507-508): These results suggest, on the one hand, that the denitrified waters of the southern part of the OMZ (30°-32° S, 80° W) remained confined in the eastern part of the Pacific.

Modified text: These results suggest, on the one hand, that negative N* in the southern part of the OMZ (30°-32° S, 80° W) remained confined to the eastern part of the Pacific.

**General comment:**   In the modified text, all the corrections referred to above have been taken into account, and we propose a new version of section 4.2.

Modified text:

South of 15° S in the WTSP, the positive N* observed in both the upper (σ24.7) and lower (σ26.3) thermoclines (Fig. 8a, 8b, Fig. S2) can be explained by the high levels of $N_2$ fixation observed in this region (Bonnet et al., 2017). South of 15° S in the Melanesian Archipelago, the thermocline waters are referred to as the Northern component of Subtropical Lower Water (Wyrtki, 1962a; Solokov and Rintoul, 2000). In the south of the Coral and in the Tasman Seas, between Australia and New Zealand and north of latitude 40° S, N* was positive in the lower thermocline water (Fig. 8b). Solokov and Rintoul, (2000) showed that south of 25° S on the P11 section (Fig. 8c), the lower thermocline water was supplied by the lighter, northern variety of Subtropical Lower Waters described above, which were characterized by an excess of $NO_3^-$ on σ24.7. The northern Subtropical Lower Waters outcropped in Winter in the central Tasman Sea, and are converted to the dense southern variety of Subtropical Lower Water (Solokov and Rintoul, 2000). In the Tasman Sea, we observed an eastward flow in a band between 22° S and 27° S (Wyrtki, 1962). The EAC (see fig.1b), separated from the coast near 30° S into a series of filament-like eastward flows in the Tasman Sea (Ridgway and Dunn, 2003) and transport 30 Sv of Subtropical Lower Water (Solokov and Rintoul, 2000).We hypothesize that the positive N* observed in the thermocline waters of the Tasman Sea could reflect the southern and then eastern transport *via* the EAC recirculation (see fig.1b) (Stramma et al., 1995; Wijffels et al., 2001) of the positive N* observed in the Melanesian Archipelago waters and in the Coral Sea (Fig. 8a, 8b). Our results showed that N* was positive in the thermocline waters of the entire South Pacific between latitude 23° S and 32° S and up to longitude 100° W. On P06-2009 section (32° S, 150° E-80° W, Fig. 8c), N* remained relatively high and stable at σ26.3. Solokov and Rintoul, (2000) suggested that ventilated Subtropical Lower Water carried eastward by the EAC spread around the subtropical gyre of the South Pacific. From a linear inverse model combined with P06 cruise data (Fig. 8c), Wijffels et al. (2001) showed that 7 Sv of thermocline waters recirculate west to the Tonga-Kermadec-Ridge (Fig. 1b), and that the thermocline flow is predominantly zonal between 177° W and 125° W. In the eastern part of the basin, at longitude 80°-90° W, a sharp decrease of N* was

observed (Fig. 8b), which could be linked to denitrification in this area. These trends are also observed in sections P06 - 1992 and P06 - 2003 (Fig. S5e, Fig. S5f). These results suggest, on the one hand, that negative N* in the southern part of the OMZ (30°-32° S, 80° W) remained confined to the eastern part of the Pacific. On the other hand, the southern branch of the subtropical gyre (see Fig. 1b) was the main vector of excess N transport in the thermocline waters of the South Pacific. Using cruise datasets that have occurred at different seasons and years, and thus not considering seasonal and interannually variability in N*, our interpretation is questionable. Nevertheless, we found the same patterns at crossover stations (Table S1, Fig. S4, Fig. S5). A strong negative N* was observed on σ24.7 and σ26.3 roughly between latitude 5° S and 20°-23° S, and a positive N* was observed in the WTSP and in the southern part of the basin, roughly between latitude 23° S and 32° S (Fig. S4, Fig. S5).

**Section 4.2.3: Spatial decoupling between N sources and sinks in the south Pacific?**

**General comment:** The title of this section has been rephrased.

Modified text: 4.3 Regarding the spatial coupling/decoupling between N sources and sinks in the South Pacific

> ***111.*** ***This is clearly a wrong interpretation of the Gruber 2016 paper!! Please read and understand a paper before citing it. Gruber (2016) is a commentary on the paper by Knapp et al. from the same PNAS issue. Gruber does not provide any new results, but just introduces the uncertainties related to both the Deutsch (inverse model) study and the Knapp observational study. The results of Knapp should be cited here, which are in line with Bonnet et al. 2017.***

Original text (lines 514-518): Gruber (2016) still confirms using an inverse modeling approach the hypothesis of Deutsh et al. (2007) that the coupling between denitrification and $N_2$ fixation would be spatially narrow, and concludes that $N_2$ fixation rates of the order of 500 µmol N m$^{-2}$ d$^{-1}$ are present near the Chilean coast (80°-100° W). The model results also showed $N_2$ fixation rates of the order of 280 to 400 µmol N m$^{-2}$ d$^{-1}$ in the subtropical gyre between 100° W and 160° W despite the fact that recent fixation measurements in the central and eastern part of the subtropical Pacific showed very low levels of $N_2$ fixation (Bonnet et al., 2017).

Response: We agree with Reviewer#2 and we apologize for the wrong interpretation of the Gruber 2016 paper. All comments have been taken into account and we propose a new version of the whole section in the revised version of the manuscript (see the modified text below answer to comment 123).

**112.      *between 20S and ...?or north of 20S ? south of 20S***

Original text (lines 520): In the ETSP and the South Pacific gyre, between longitude 80° W and 100° W and latitude 20 °S,

Response: Just at latitude 20°S. But the sentence has been deleted in the revised paragraph (see the modified text below answer to comment 123).

**113.      *use consistent units throughout your paper***

Original text (lines 522): Moutin et al. (2008) also report extremely low $N_2$ fixation rates ($\approx 0.12$ nmol N $L^{-1}$ $d^{-1}$) in the subtropical gyre waters

Response: We agree with Reviewer#2, but the sentence has been deleted in the revised paragraph (see the modified text below answer to comment 123).

**114.      *This is a vague term here, you likely mean N\*, right? N\* itself does not provide any evidence for rates, you need an age estimate in addition.***

Original text (lines 525): and in situ data and geochemical calculation

Response: We agree with Reviewer#2, the sentence was confusing. N\* itself does not provide any evidence for rates, and furthermore the age of water mass is not known to date in the WTSP. The sentence has been removed in the revised paragraph (see the modified text below answer to comment 123).

**115.      *unclear, what do you mean? try to be a little more specific***

Original text (lines 526): Indeed, according to the first results, it seems that the spatial coupling between the strong denitrification zones and the strong $N_2$ fixation zones could exist.

Response:  We agree, thank you for the comment. The sentence has been rewritten more clearly in the revised manuscript (see the modified text below answer to comment 123).

***116.*** ***The coupling is one way, i.e. high P\* waters transported to the East to stimulate N2-fix there. A transport of high N\* waters in the other direction does not matter for the coupling.replace by: and is promoted by the transport of elevated P\* waters from West to East.***

Original text (lines 18): through the westwards and eastward transport of the SEC, the EAC and finally of the SPC between a High Iron/High Nitrogen/High Phosphate (HI-HN-HP) zone in the eastern part and a High Iron/Low Nitrogen/Low Phosphate (HI-LN-LP) zone in the western part of the subtropical Pacific

Response: We thank Reviewer#2 for this comment. The sentence has been rephrased in the revised manuscript (see the modified text below answer to comment 123).
.

***117.*** ***If you only repeat the conclusions/hypthesis from companion papers, you question the significance of your own work.***

Original text (lines 531-533): As hypothesized by Bonnet et al. (2017) and Moutin et al. (this issue), the spatial coupling could thus be spatially more distant than predicted by Deutsch et al. (2007) and Gruber (2016).

Response: The debate about the close or large spatial coupling between areas of denitrification and $N_2$ fixation is indeed an old debate, and Deutsch himself initially proposed a large coupling (Deutsch et al., 2001) and later a closer spatial coupling (Deutsch et al., 2007), first contradicted by Moutin et al. (2008), and later by Knapp et al. (2016) and Bonnet et al. (2017). In the present paper, we give a reasonable analysis of the water masses circulation in the whole South Pacific basin, coupled with a new basin scale analysis of all available data allowing a N\* calculation. Our work confirms the influence of low N\* (high P\*) water originating from the east toward the north of the WTSP, and additionally shows for the first time the large-scale influence of $N_2$ fixation in the WTSP in the south of the South Pacific gyre. The sentence has been rewritten (see the modified text below answer to comment 123).

***118.*** ***'elusive'?***

Original text (lines 534): If the $N_2$ fixation was elusive in the GY,

Response: We agree with Reviewer#2, the term was confusing and has been deleted in the revised paragraph (see the modified text below answer to comment 123).

**119.** *Simplify sentence.*

Original text (lines 533-534): If the $N_2$ fixation was elusive in the GY, how will it be possible to explain the increase in N* signal of the thermocline waters during their westward transport by the nothern branch of the SEC between latitude 5° S and 23° S.

Response: The sentence has been deleted in the revised manuscript (see the modified text below answer to comment 123).

**120.** *You should not only refer to work of your co-authors. See Mills and Arigo, 2010, NGEO for a similar suggestion*

Original text (lines 535) : Moutin et al. (2008) argued that $N_2$ fixation is not the only process driving the increase in N* (or decrease in P*), and hypothesize that the export of material with a N:P ratio lower the RR could be at the origine of N* increase.

Response: The sentence has been removed in the revised paragraph (see the modified text below answer to comment 123).
The reference to Mills and Arrigo (2010) and Singh et al. (2017) has been added in section 4.1 of the revised manuscript.

**121.** *TKG? compare Fig. 2b*

Original text (lines 538): Wijffels et al. (2001) suggested that on P06 transect west of TKR

Response: Here, the term is correct (TKR = Tonga-Kermadec-Ridge), but the full name has been added.

**122.** *what do you mean?'at the origin' (providing just the initial value) or 'the origin' (explaining the westward increase?*

Original text (lines 541): We hypothesize that the nitrogen excess observed on σ26.3 in the WTSP is advected from the Tasman sea first eastward and then northward in the circulation of the gyre and could influence positively the thermocline waters of the South Pacific being thus at the origin of the westward increase of the strongly negative N* signal transported by the SEC.

Response: In order to clarify our Discussion, this part has been removed from the revised version of the manuscript.

**123.** *(the update was published in 2016, I doupt that the database was updated in the year of publication)*

Original text (lines 545): This N* study is based on all cruises data available in the GLODAPv2 database updated in 2016.

Response: We thank Reviewer#2 for this comment. We apologize for this mistake and the term 'updated in 2016' has been deleted in the revised version.

**General comment:** We thank Reviewer#2 for all these remarks and corrections. All comments have been taken into account in the revised paragraphs. We propose a new version of the whole section in the revised version of the manuscript.

Modified text:

Coupling a biogeochemical model with ocean circulation, Deutsch et al. (2007) hypothesized that denitrification and $N_2$ fixation sites are geographically coupled in the ETSP. These authors suggested the low N* (high P*) waters originating from the ETSP and reaching the South Pacific gyre create a favorable niche diazotroph development. However, *in situ* direct measurements have shown low $N_2$ fixation rates in the waters adjacent to the ETSP (Moutin et al., 2008; Knapp et al., 2016), questioning such spatial coupling. The coupling was recently hypothesized to be at larger spatial scale, considering that iron availability plays a central role, allowing (in the WTSP) or preventing (in the gyre) the process of $N_2$ fixation (Bonnet et al., 2017) of the waters transported by the global surface circulation (SEC). In this study, we observed that the surface eastern central Pacific waters with low N* (high P*) sink on isopycnal σ24.7 (18°-20° S, 100°-150° W, Fig. S1a). These surface and subsurface waters sink with a subduction rate of 6-7 Sv in the South Pacific (12°-25° S; 100°- 170° W) (Donguy and Henin 1977; O'Connor et al. 2002; Fiedler and Talley, 2006) under the lighter and less salty surface waters produced by an excess of precipitation over evaporation. This water mass is named the South Pacific Equatorial Water (Donguy and Henin, 1977; Emery and Meincke, 1986; Tomczak and Hao, 1989). Tomczak and Hao, (1989) and Solokov and Rintoul, (2000), showed that South Pacific Equatorial Water is transported westward from the central part of the South Pacific (12°-25° S; 100°- 150° W), and has a strong influence on the water masses in the Coral Sea and the Solomon Sea. From OUTPACE data and the GLODAPv2 database, we showed that in the upper thermocline of the gyre, the high salinity on σ24.7 (Fig. 2a) is associated with a low N* (high P*) (Fig. 8a). The low N* (high P*) waters originating from the ETSP reach the upper

thermocline waters of the iron rich WTSP and could create a favorable environment for diazotrophs. Nevertheless, the upper thermocline waters were detected between 100-250 m depth, i.e. below the depth where most of the $N_2$ fixation was observed (30 m; Fig. 3). Moreover, the Winter mixed layer depths allowing the replenishment of surface waters in excess P never exceed 100 m depth in the WTSP (Moutin et al., 2018). Because the influence of waters originating from the ETSP is below the maximum mixing depth estimated in the WTSP, the link between N sink in the east and N source in the west implies longer timescales than previously thought, and even calls into question the close coupling between denitrification and $N_2$ fixation.

**Conclusion**

*124.        deficit? compared to what?*

Original text (lines 555): On the one hand, the waters located in the Melanesian Archipelago (160° E-170° W) have a deficit of $NO_3^-$ and $PO_4^{3-}$ in the productive layer (0-100 dbar),

Response: Thank you for these comments, it was indeed a mistake in the formulation of the sentence. This sentence has been deleted in the revised manuscript (see the modified text below answer to comment 130).

*125.        you mean 'an excess of PON relative to the Redfield equivalent POP concentration'*

Original text (lines 555-556): an excess of PON relative to POP associated in part with diazotrophic POM formation

Response: The sentence has been corrected in the revised version of the manuscript.

Modified text: a PON in excess relative to that expected from Redfield stoichiometry

*126.        'nitrate in excess over the Redfield equivalent of PO4'*

Original text (lines 557): This nitrogen excess over $PO_4^{3-}$ measured in the waters of the MA

Response: The sentence has been rephrased in the revised version.

Modified text: Positive N* in the surface waters and in the upper and lower thermocline waters of the Melanesian Archipelago was observed in parallel with record-high rates of $N_2$ fixation.

**127.** *your usage of abbreviations is excessive! why shall a reader learn the many abbreviations you introduce in your paper? use abbreviations with care! At the minimum you have to provide a table with all your abbreviations. Better; replace those rarely used (< 5 times) by full text.*

Original text (lines 559): are characterized by $NO_3^-<QL$,

Response: To clarify our paper, many abbreviations have been removed in the revised version of the manuscript, and those rarely used have been replaced by the full text. A table with the mean abbreviations has been added in the Supplementary Materials of the revised manuscript.

**128.** *excess of what? either you define PONexcess carefully early in the paper, or you write out here*

Response: PON excess was defined earlier in the revised version of the manuscript (see section 2.4 of the revised version).

**129.** *what is a N\* anomaly?, isn't N\* already an anomaly, hence 'N\* anomaly' is a tautology*

Original text (lines 559): significant $PO_4^{3-}$ concentrations, a lack of PON in excess, and a N\* anomaly close to zero or negative

Response: Yes, this term has been corrected here and everywhere in the revised version of the paper.

**130.** *it is already a hypothesis*

Original text (lines 565-566): We hypothesize that the southern branch of the subtropical gyre is probably the main vector of excess nitrogen transport in the thermocline waters of the South Pacific.

Response: Yes, it is already a hypothesis. The term has been deleted and the sentence has been rephrased in the revised manuscript (see the modified text below).

**General comment:** In the modified text, all the corrections referred to above have been taken into account, and we propose a new version of the Conclusion.

Modified text:

As part of the OUTPACE cruise in the western tropical South Pacific (160° E-160° W, 18°-20° S), we have analyzed the effect of the local $N_2$ fixation process on the dissolved and particulate organic N pools. We have revealed the presence of two distinct oceanic areas. The waters located in the Melanesian Archipelago (160° E-170° W) were characterized by significant $N_2$ fixation rates, high PON concentrations and a PON in excess relative to that expected from Redfield stoichiometry in the surface waters and in the top of the upper thermocline waters. In this area, we showed a significant relationship between $N_2$ fixation and PON in excess. Positive N* in the surface waters and in the upper and lower thermocline waters of the Melanesian Archipelago was observed in parallel with record-high rates of $N_2$ fixation. In contrast, the surface waters and the top of the upper thermocline waters of the South Pacific Gyre (170° W-160° W) are characterized by low $N_2$ fixation rates and low PON concentrations. In agreement with the low diazotrophic activity observed, negative or close to zero N* values were observed in the South Pacific Gyre.

The intense $N_2$ fixation observed in the WTSP significantly influences the dissolved inorganic (N excess) and particulate organic (PON excess) N pools of the surface and thermocline waters in the WTSP. The analyses of the most recent available nutrient data in the South Pacific Ocean enabled us to show a high positive N* in the thermocline waters of the whole South gyre. We therefore hypothesized that the local high N* observed in the Melanesian Archipelago is transported eastward *via* the circulation, and consequently that $N_2$ fixation in the WTSP may influence the nutrient distribution at the scale of the South Pacific basin.